# Optogenetic stimulation of vagal nerves for enhanced glucose-stimulated insulin secretion and β cell proliferation

Yohei Kawana[1], Junta Imai [1]✉, Yosuke M. Morizawa [2], Yoko Ikoma[2], Masato Kohata[1], Hiroshi Komamura[1], Toshihiro Sato[1], Tomohito Izumi [1], Junpei Yamamoto[1], Akira Endo[1], Hiroto Sugawara[1], Haremaru Kubo[1], Shinichiro Hosaka, Yuichiro Munakata[1], Yoichiro Asai[1], Shinjiro Kodama[1], Kei Takahashi [1], Keizo Kaneko[1], Shojiro Sawada[3], Tetsuya Yamada[4], Akira Ito[5], Kuniyasu Niizuma [5,6,7], Teiji Tominaga[5], Akihiro Yamanaka [8], Ko Matsui [2] & Hideki Katagiri [1]

The enhancement of insulin secretion and of the proliferation of pancreatic β cells are promising therapeutic options for diabetes. Signals from the vagal nerve regulate both processes, yet the effectiveness of stimulating the nerve is unclear, owing to a lack of techniques for doing it so selectively and prolongedly. Here we report two optogenetic methods for vagal-nerve stimulation that led to enhanced glucose-stimulated insulin secretion and to β cell proliferation in mice expressing choline acetyltransferase-channelrhodopsin 2. One method involves subdiaphragmatic implantation of an optical fibre for the photostimulation of cholinergic neurons expressing a blue-light-sensitive opsin. The other method, which suppressed streptozotocin-induced hyperglycaemia in the mice, involves the selective activation of vagal fibres by placing blue-light-emitting lanthanide microparticles in the pancreatic ducts of opsin-expressing mice, followed by near-infrared illumination. The two methods show that signals from the vagal nerve, especially from nerve fibres innervating the pancreas, are sufficient to regulate insulin secretion and β cell proliferation.

Pancreatic β cells have been shown to maintain substantial proliferative activity in vivo despite being terminally differentiated[1–4]. This feature contributes to preventing hyperglycaemia, which develops in insulin-resistant states, by increasing β cell mass and enhancing insulin secretion in response to increased insulin demand. It is worth noting that the volume of β cells decreases not only in type-1 diabetes but also in advanced type-2 diabetes[5]. Thus, modulating β cell proliferation is one of the major goals of diabetes research, because such strategies

[1]Department of Metabolism and Diabetes, Tohoku University Graduate School of Medicine, Sendai, Japan. [2]Super-network Brain Physiology, Tohoku University Graduate School of Life Sciences, Sendai, Japan. [3]Division of Metabolism and Diabetes, Faculty of Medicine, Tohoku Medical and Pharmaceutical University, Sendai, Japan. [4]Department of Molecular Endocrinology and Metabolism, Graduate School of Medical and Dental Sciences, Tokyo Medical and Dental University, Tokyo, Japan. [5]Department of Neurosurgery, Tohoku University Graduate School of Medicine, Sendai, Japan. [6]Department of Neurosurgical Engineering and Translational Neuroscience, Tohoku University Graduate School of Medicine, Sendai, Japan. [7]Department of Neurosurgical Engineering and Translational Neuroscience, Graduate School of Biomedical Engineering, Tohoku University, Sendai, Japan. [8]Department of Neuroscience II, Research Institute of Environmental Medicine, Nagoya University, Nagoya, Japan. ✉e-mail: imai@med.tohoku.ac.jp

**Fig. 1 | ChR2–YFP was expressed in vagal nerves of ChAT–ChR2 mice.**
**a**, Graphical abstract of two oVNS methods. Left: ChAT–ChR2 mice implanted with a small optical fibre in the subdiaphragmatic oesophagus This fibre delivers light stimulation to the vagal nerves that are connected to the pancreas. Right: blue light-emitting LMPs in the pancreatic ducts of the ChAT–ChR2 mice. Mice are exposed to NIR light, which stimulates the vagal nerves connected to the pancreas without the need for fibre implantation. **b**, YFP-positive vagal trunk running along the oesophageal wall. Each white arrowhead denotes vagal trunk. **c**, YFP-positive parasympathetic ganglia adjacent to a pancreatic islet, counterstained with anti-Tuj1 antibody (general neuronal marker). More than two ChAT–ChR2 mice were tested. Each white arrowhead denotes parasympathetic ganglia. DAPI, 4′,6-diamidino-2-phenylindole. **d**, YFP-positive parasympathetic fibres in contact with each of the islet cell types. More than two ChAT–ChR2 mice were tested. Each white arrowhead indicates YFP-positive nerves that make contact with CD31-positive vascular endothelial cells in islets. Each 3D islet image is reconstructed as Z-stacks of 70 images with a pitch of 0.3 μm. Ins, insulin-positive β cells; Gcg, glucagon-positive α cells; Sst, somatostatin-positive δ cells. **e**, Contact density between cholinergic fibres and islet cells, expressed as the proportion of the total volume of the shared areas between YFP-positive nerve fibres and islet cells/total volume of islet cells ($n$ = 3 mice, 5 islets per mouse). Data are presented as mean ± s.e.m. Scale bars denote 100 μm (**b**), 50 μm (**c**) and 30 μm (**d**).

for increasing β cell mass would potentially be effective therapies for a substantial portion of the diabetic population.

Vagal nerve-derived signals are reportedly involved in regulating both insulin secretion[6–8] and β cell mass[8–13]. A majority of parasympathetic ganglia in the pancreas were shown to be located in the vicinity of pancreatic islets[12,14,15]. This anatomical structure may allow the vagus to selectively stimulate islet cells. As for glucose-stimulated insulin secretion (GSIS), stimulatory effects of cholinergic agonists in vitro were previously reported[16]. As for β cell mass, a previous report[9] showed that lesioning of ventromedial hypothalamus, a sympathetic nerve

centre, enhanced β cell proliferation. β cell proliferation in ventromedial hypothalamus-lesioned rats was blocked by subdiaphragmatic vagotomy, suggesting the involvement of vagal signals in the β cell proliferative effects. In addition, vagal nerve signals reportedly play an important role in compensatory β cell proliferation during obesity development[8,10–12]. Furthermore, we previously clarified that a neuronal relay, consisting of splanchnic afferents from the liver and vagal efferents to the pancreas, is involved in β cell proliferation[8,12]. In these reports, however, vagal-nerve involvement was revealed by experimental interruptions of the neuronal network, including dissection of the vagal nerves innervating the pancreas. Thus, these previous results highlighted the necessity of vagal nerve signals for promoting GSIS as well as eliciting β cell proliferation in several physiological and experimental settings. However, whether activation of the efferent vagal nerves is sufficient to promote GSIS and/or cause β cell proliferation is unclear. A major obstacle to showing sufficiency has been the absence of techniques allowing stable, prolonged and selective in vivo activation of the vagal nerves in awake animals.

Optogenetic activation and de-activation have emerged as powerful tools for investigating neuronal functions[17]. Most of the previous optogenetic studies focused on the central nervous system[18]. In addition, optogenetic activation and de-activation approaches have recently been applied to peripheral nerves, such as the sciatic[19] and sympathetic[20–22] nerves. However, optogenetic strategies have not been applied to vagal nerves by employing peripheral approaches. Furthermore, it has been difficult to optogenetically modulate peripheral nerves innervating intended tissues and organs selectively and chronically in freely moving animals. In this Article, to clarify whether vagal nerve stimulation is sufficient to promote GSIS and/or β cell proliferation, we developed two optogenetic vagal-nerve-stimulation methods. First, we generated choline acetyltransferase (ChAT)–channelrhodopsin 2 (ChR2) mice in which ChR2 was expressed selectively in cholinergic neurons and developed an optical fibre-implanting surgical technique allowing photostimulation of the subdiaphragmatic vagal nerves, selectively and chronically, in freely moving ChAT–ChR2 mice. Next, to exclude the extra-pancreatic effects of optogenetic subdiaphragmatic vagal nerve stimulation and to selectively stimulate the vagal nerves innervating the pancreas, we developed a near-infrared (NIR) light-mediated optogenetic nerve stimulation method by placing blue light-emitting lanthanide microparticles (LMPs) in the pancreatic ducts of ChAT–ChR2 mice, followed by illumination with NIR light, applied externally to the bodies of freely moving ChAT–ChR2 mice (Fig. 1a). As the body is permeable to NIR, this method has enabled us to activate vagal nerves innervating the pancreas, as intended without fibre implantation being required. Employing these optogenetic strategies, we herein show that selective activation of vagal nerves innervating the pancreas is sufficient to both enhance GSIS and induce marked β cell proliferation, thereby increasing functional β cell mass and suppressing hyperglycaemia induced by insulin deficiency.

## Results

### ChR2 is expressed in vagal nerves of ChAT–ChR2 mice

In an effort to achieve stable vagal nerve activation in vivo, we employed the optogenetic system. Rosa–CAG–LSL–ChR2 (H134R)–enhanced yellow fluorescent protein (EYFP) mice are widely used for optogenetic research[23]. Therefore, we generated ChAT–ChR2 mice by crossing ChAT–IRES–Cre mice, in which Cre recombinase was expressed specifically in cholinergic neurons[24,25], with Rosa–CAG–LSL–ChR2 (H134R)–EYFP mice. Consequently, in ChAT–ChR2 mice, ChR2 (H134R)–EYFP fusion protein was expressed under the control of the CAG promoter in cholinergic neurons after Cre recombination[26]. These mice were expected to express ChR2–EYFP fusion protein in all cholinergic neuronal cells and fibres, including vagal nerves innervating pancreatic islets. First, we histologically analysed the expressions of YFP in several intra-abdominal sections. We detected a YFP-positive vagal trunk running along the oesophageal wall (Fig. 1b). In addition, YFP-positive cells and fibres were detected in the stomach, the duodenum, the jejunum and the ileum, but not in the liver or around the portal vein (Extended Data Fig. 1a). Importantly, and consistently with our previous report[12] showing many of the pancreatic parasympathetic ganglia to be located in the vicinity of pancreatic islets, the clumps of YFP-positive cells, which were also positive for β-tubulin 3, a neuron cell-specific marker, were observed to be adjacent to pancreatic islets (Fig. 1c). These data indicate that ChR2 protein is indeed expressed in vagal nerves, including those in the pancreas, of ChAT–ChR2 mice.

Using these mice, we investigated whether vagal nerve fibres were in contact with islet cells as well as with vascular endothelial cells in islets by estimating the contact density. Consistent with a previous report[27], YFP-positive nerve fibres made contact with both β and α cells, which are positive for insulin and glucagon, respectively (Fig. 1d,e). In addition, YFP-positive nerves made contact with CD31-positive vascular endothelial cells in islets, and, to a lesser extent, with somatostatin-positive δ cells (Fig. 1d,e). These findings suggest that vagal nerve fibres directly regulate both endocrine function and blood flow in pancreatic islets.

### Optical stimulation activates vagus of ChAT–ChR2 mice

We next examined whether photostimulation functionally activates intra-abdominal vagal nerves in ChAT–ChR2 mice. Blue light stimuli were loaded onto the subdiaphragmatic anterior vagal trunk of anaesthetized ChAT–ChR2 mice, and photostimulation parameters as well as electrophysiological responses were recorded using hooked paired bipolar electrodes (Fig. 2a). Simple near-biphasic and small amplitude responses to blue light (470 nm) were evoked by the 0.5 ms stimulus, whereas larger complex compound responses were evoked by increasing the optical stimulus duration to 10 ms (Fig. 2b). No electrophysiological responses were observed in the absence of the blue light illumination. The response amplitude appeared to approach almost the maximum, and was then maintained at this level, with the 25 ms stimulus duration (Fig. 2b). Importantly, the amplitude of vagal nerve activity elicited by the 25 ms duration of stimulation was similar to that observed for physiological vagal nerve activity[28].

Next, long trains of multiple 25-ms pulses (20 Hz) were applied to vagal nerves (Fig. 2c), and we observed stable responses to these illuminations. Then, the same sequences were delivered every 10 s 30 times (Fig. 2d). The responses appeared to be stable and no failures were observed. On the other hand, a longer illumination duration, for 5 s, produced only an initial compound response, with no apparent further responses being observed after this initial response (Fig. 2e). These data showed that vagal nerves of ChAT–ChR2 mice do indeed respond to the blue light stimuli and that photostimulation, delivered by frequent pulses of 25 ms duration, rather than a single pulse of longer duration, activates the vagal nerves of ChAT–ChR2 mice.

### Optogenetic stimulation elicits vagal activation in vivo

To photostimulate subdiaphragmatic vagal nerves of freely moving ChAT–ChR2 mice selectively and chronically, we developed the optical fibre-implanting surgical procedure illustrated in Fig. 3a. We elaborately devised a silicon cuff attached to the tip of an optical fibre (Extended Data Fig. 1b) and a needle to penetrate the back wall of the abdominal cavity (Extended Data Fig. 1c). Then, we set the optical fibre near the posterior vagal trunk of anaesthetized ChAT–ChR2 mice by wrapping the silicon cuff around the subdiaphragmatic oesophagus. The other edge of the optical fibre was passed through a needle penetrating the back wall of the abdominal cavity and connected to the light source and driver systems (Fig. 3b,c). After awaking from anaesthesia, mice which had been operated on moved freely and showed no apparent disturbance in daily activities (Fig. 3b and Supplementary Video 1). Then, these mice were maintained with or without optogenetic vagal nerve stimulation (oVNS) by blue light stimuli evoked by a light-emitting

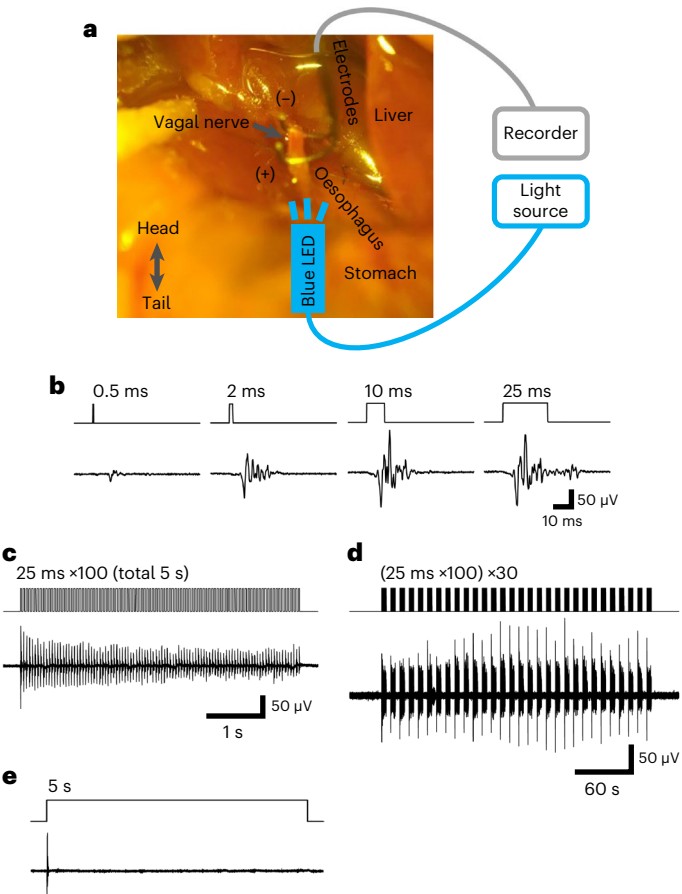

**Fig. 2 | Optical stimulation activated vagal nerves of ChAT–ChR2 mice.**
**a**, The setting for electrophysiological measurement of ChAT–ChR2 mice subdiaphragmatic vagal nerve response to blue light stimuli (470 nm) under anaesthesia. **b**, Vagal nerve responses evoked by various pulse-width patterns. **c**,**d**, Persistent vagal nerve responses evoked by a train of pulses (25 ms, 20 Hz, 100 pulses, total 5 s) (**c**) and its repetition of 30 trains (10 s interval, total 5 min) (**d**). **e**, Only an initial short compound response, which did not persist, was produced by illumination for 5 s.

diode (LED) (LED-oVNS). Chronic LED-oVNS loading was continued for 2 weeks according to the schedule shown in Fig. 3d. We adopted a pulse frequency of 5 Hz for photostimulation, because this pulse frequency, as well as the amplitude, is within the range of observed physiological vagal nerve activities[28] and previous electrical VNS studies showed that low stimulation frequencies (1–10 Hz), rather than high frequencies (20–30 Hz), more efficiently stimulate efferent nerves[29,30]. The light intensity of LED was in the range of 5–10 mW. Electrophysiological responses of intrapancreatic vagal nerves were clearly evoked by 5 Hz of blue light stimulation (Fig. 3e).

Next, we examined body weights of ChAT–ChR2 mice with (CC-LED-mice) or without LED-oVNS (CC-ctrl-mice). Body weights did not differ between CC-LED-mice and CC-ctrl-mice, in which optical fibres had been implanted but without loading of blue light stimuli, during the experimental period (Extended Data Fig. 1d). Neither diarrhoea nor muscle convulsion was observed in CC-LED-mice. Therefore, adverse effects of LED-oVNS on general conditions of ChAT–ChR2 mice were unlikely. The parasympathetic nervous system is known to promote vasodilatation in several organs, such as the brain[31] and the salivary glands[32]. Therefore, we evaluated blood flow in the pancreas in vivo in CC-LED-mice employing a laser speckle tissue blood flow imager in accordance with a previous report[33]. Consistent with the results that YFP-positive cells and fibres not being detected in the livers

of ChAT–ChR2 mice (Extended Data Fig. 1a) as well as with previous reports indicating vagal innervation of the liver to be very sparse[34], hepatic blood flow was unaffected. On the other hand, LED-oVNS significantly increased blood flows in both the pancreas and the duodenum in CC-LED-mice as compared with CC-ctrl-mice (Extended Data Fig. 1e). These findings suggest that, while stimulation of vagal nerves innervating the pancreas by LED-oVNS was feasible, the LED-oVNS system may also affect vagal nerves innervating organs in the abdominal cavity other than the pancreas.

We then performed glucose tolerance tests (GTTs) concomitantly with LED-oVNS. Plasma levels of glucagon in the fasting state and at 15, 30 and 60 min after glucose loading, and those of somatostatin and amylase in the fasting state and at 15 min after glucose loading in CC-LED-mice were similar to those in control mice (Extended Data Fig. 2a,b). In addition, amylase concentrations in the pancreatic bile juice did not differ after versus before the 30-min LED stimulation (Extended Data Fig. 2c), indicating that LED-oVNS exerted minimal effects on amylase secretion into the duodenum. On the other hand, plasma insulin levels after glucose loading were significantly higher in CC-LED-mice than in CC-ctrl-mice (Fig. 3f), while blood glucose levels did not differ between the two groups (Fig. 3g). Insulin tolerance testing performed concomitantly with LED-oVNS revealed that insulin sensitivity was similar in the CC-LED-mice and CC-ctrl-mice (Extended Data Fig. 2d). Thus, optogenetic activation of subdiaphragmatic vagal nerves selectively enhances insulin secretion, especially after glucose stimulation without altering insulin sensitivity. These findings confirmed that the LED-oVNS system activates vagal nerves in freely moving mice, leading to functional modulation of β cells.

**LED-oVNS induces β cell proliferation**
Our next goal was to examine whether chronic LED-oVNS increases β cell mass. We evaluated β cell mass after 2 weeks of stimulation. Electrophysiological responses of the oesophageal vagal nerves to LED stimulation were clearly detected 2 weeks after starting LED-oVNS (Extended Data Fig. 3a). Notably, β cell mass was significantly increased, by 1.6-fold, in CC-LED-mice as compared with CC-ctrl-mice and WT-LED-mice, in which optical fibres had been implanted in wild-type mice followed by loading of LED blue light stimuli, (Fig. 3h). Furthermore, bromodeoxyuridine (BrdU)-positive β cells were markedly increased in CC-LED-mice as compared with CC-ctrl-mice and WT-LED-mice after the 2-week stimulation (Fig. 3i). Insulin tolerance testing with or without concomitant LED-oVNS revealed that insulin sensitivity did not differ between the CC-LED-mice and CC-ctrl-mice, and it was true under both experimental conditions, after LED stimulation for 2 weeks (Extended Data Fig. 3b) at the timepoint when BrdU-positive β cells were markedly increased (Fig. 3i). Thus, vagal nerve activation, rather than altered systemic insulin sensitivity, is critical for promoting β cell proliferation in CC-LED-mice. Ratios of terminal deoxynucleotidyl transferase dUTP nick end labelling (TUNEL)-positive β cells in CC-LED-mice were similar to those in CC-ctrl-mice after the 1-week stimulation (Extended Data Fig. 3c). Proliferations of α, δ and exocrine cells were not enhanced in CC-LED-mice (Extended Data Fig. 3d). The sizes of α and δ cells were similar in CC-LED-mice and CC-ctrl-mice (Extended Data Fig. 3e). These results indicate that chronic activation of subdiaphragmatic vagal nerves selectively induces β cell proliferation, thereby substantially increasing β cell mass.

However, food intake amounts were increased by about 1.2-fold, a significant difference, in CC-LED-mice as compared with CC-ctrl-mice throughout the experimental period (Extended Data Fig. 4a). Subdiaphragmatic vagal nerves innervate not only the pancreas but also other organs in the abdominal cavity, including the gastrointestinal tract[35]. In fact, we observed that LED-oVNS significantly enhanced gastrointestinal tract motility (Extended Data Fig. 4b and Supplementary Videos 2 and 3) and duodenum blood flows (Extended Data Fig. 1e).

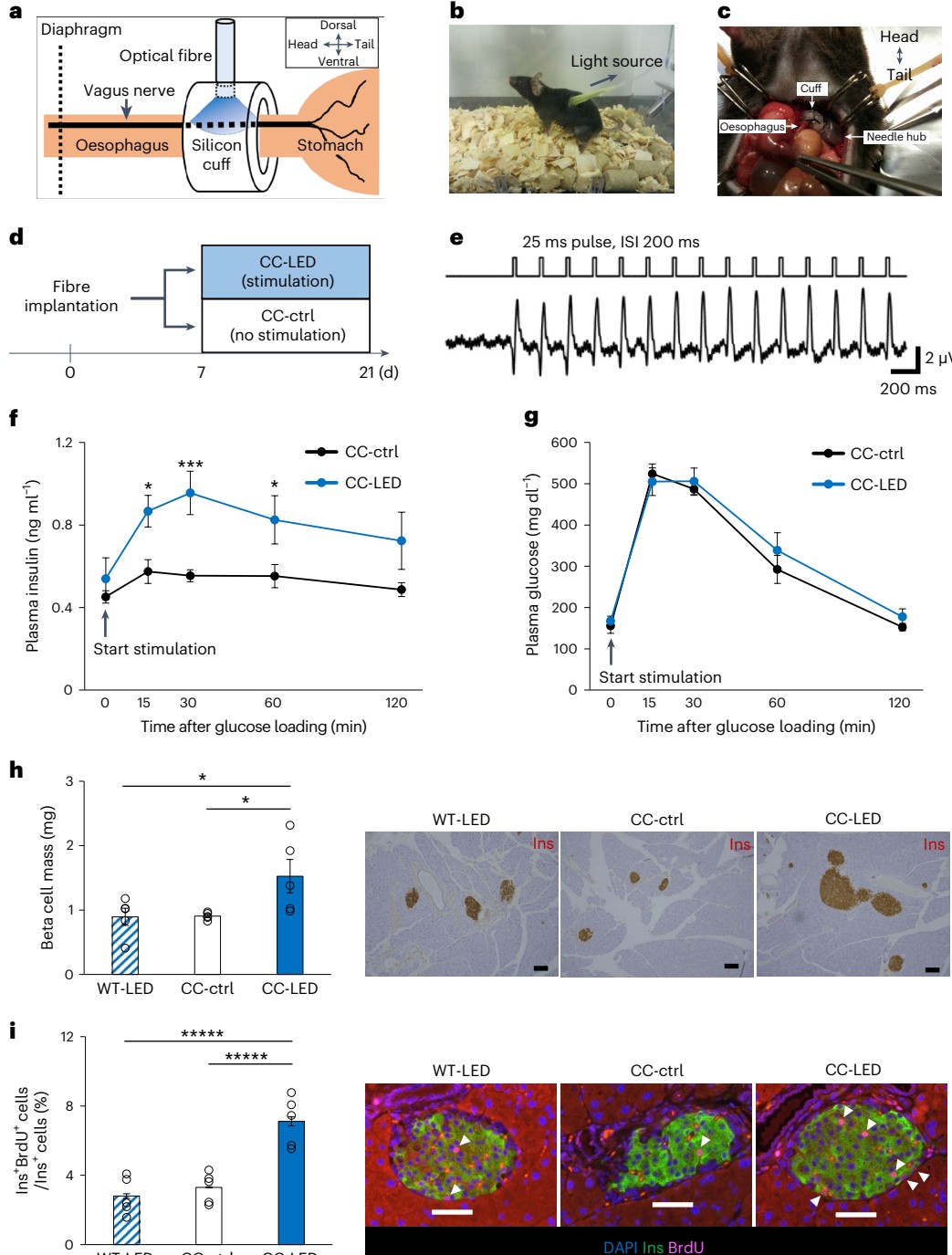

**Fig. 3 | LED-oVNS elicited acute and chronic vagal nerve activation in vivo and induced pancreatic β cell proliferation. a**, Illustration of optical fibre implantation surgery. **b**, An optical fibre-implanted mouse. **c**, A silicon cuff was wrapped around the subdiaphragmatic oesophagus. The other end of the fibre was passed through the back using a needle. **d**, The schedule of LED-oVNS experiments. **e**, Intrapancreatic vagal nerve responses evoked by blue light stimulation. ISI, inter-stimulus intervals. **f,g**, Plasma insulin (**f**) and glucose (**g**) levels of ChAT–ChR2 mice, which had undergone acute LED-oVNS, during GTTs (**f**: two-way repeated measures ANOVA followed by Bonferroni post hoc test; CC-ctrl versus CC-LED at 15 min, *P = 0.0190; CC-ctrl versus CC-LED at 30 min, ***P = 0.0019; CC-ctrl versus CC-LED at 60 min, *P = 0.0280, n = 5). **h**, β cell masses of ChAT–ChR2 mice after 2 weeks of chronic LED-oVNS (one-way ANOVA followed by Ryan's method as a post hoc test; CC-ctrl versus CC-LED, *P = 0.0239; WT-LED versus CC-LED, *P = 0.0217, n = 5); representative images are shown in the right three panels. Scale bars denote 100 μm. Ins, insulin-positive areas. **i**, The ratio of BrdU-positive β cells to all β cells in the islets of ChAT–ChR2 mice after 2 weeks of chronic LED-oVNS (one-way ANOVA followed by Ryan's method as a post hoc test; CC-ctrl versus CC-LED, *****P = 2.121 × 10⁻⁴; WT-LED versus CC-LED, *****P = 7.08 × 10⁻⁵, n = 5); representative images are shown in the right three panels. Each arrowhead denotes a BrdU-positive β cell. Scale bars denote 50 μm. Data are presented as mean ± s.e.m. DAPI, 4′,6-diamidino-2-phenylindole.

Thus, glucose absorption from the peritoneal membrane might be enhanced, resulting in minimizing suppression of post-load blood glucose elevation after glucose loads, despite promotion of GSIS.

Therefore, a method of activating the vagal nerves innervating the pancreas more selectively is required to prove the direct involvement of vagal nerves in enhancing GSIS and β cell proliferation.

## Development of the NIR light-mediated oVNS method

To exclude any extra-pancreatic effects of oVNS and to achieve more selective vagal nerve activation, we endeavoured to utilize a recently developed NIR light-mediated optogenetic nerve stimulation system[36,37]. Upconversion microparticles absorb tissue-penetrating NIR light and locally emit wavelength-specific visible light[38]. Therefore, to achieve selective stimulation of the vagal nerves innervating the pancreas, we placed blue light-emitting LMPs in the pancreatic ducts of ChAT−ChR2 mice (CC-blue-NIR-mice), followed by NIR light illumination from outside of the bodies, with no fibre implantation (NIR-oVNS) (Fig. 4a). To negate non-specific effects, including heat production, of NIR-illuminated LMPs, we used ChAT−ChR2 mice in which red light-emitting LMPs had been placed in the pancreatic ducts as controls (CC-red-NIR-mice). After injecting the LMPs into the pancreatic ducts and NIR light illumination, mice that had undergone these procedures moved freely and showed no apparent disturbances in daily activities. Importantly, NIR light illumination from 20 cm above the isolated pancreas revealed that LMPs injected into the pancreatic ducts spread throughout the pancreas and were retained in abundance for at least 4 weeks after the injection (Fig. 4b). In addition, NIR light illumination evoked substantial blue light emission from pooled blue light-emitting LMPs in the pancreases of ChAT−ChR2 mice (Fig. 4b). Histological analysis showed that injected LMPs resided throughout the pancreas mainly in exocrine tissues (Extended Data Fig. 5a). To examine whether NIR illumination delivered from outside of the body, that is, from 20 cm above the mouse body, actually reached the pancreas inside the abdominal cavity, we measured NIR intensity in many portions of the pancreas, from head to tail, by inserting light intensity sensors into the abdominal cavities of the anaesthetized mice under the same experimental conditions. The NIR light intensities were approximately 0.5 mW mm$^{-2}$ on average and the standard error value was very small (Extended Data Fig. 5b), indicating that NIR reached the pancreas efficiently and uniformly.

Then, we measured electrophysiological responses of vagal nerves innervating the pancreas during NIR illumination by applying hooked paired bipolar electrodes to the pancreas, and found that simple near-biphasic and small amplitude responses were clearly evoked in response to NIR illumination (Fig. 4c). On the other hand, electrophysiological responses were not observed in either NIR light-illuminated wild-type mice loaded with blue light-emitting LMPs (WT-blue-NIR-mice), or light-illuminated (with another wavelength: 850 nm) ChAT−ChR2 mice loaded with blue light-emitting LMPs (Extended Data Fig. 5c). In addition, back-propagating responses were not detected at vagal nerves running along the oesophageal wall during NIR-oVNS (Extended Data Fig. 5d). Furthermore, ratios of c-Fos-positive cells in YFP-positive cells in the pancreas, that is, parasympathetic ganglionic cells, of CC-blue-NIR-mice after 1 week of NIR-oVNS were significantly increased as compared with those in CC-red-NIR-mice (Fig. 4d). Collectively, these findings electrophysiologically and histologically demonstrated that blue light derived from injected LMPs does indeed functionally activate vagal nerves innervating the pancreas. Neither diarrhoea nor muscle convulsions were observed in CC-blue-NIR-mice. Plasma amylase levels were not elevated in CC-blue-NIR-mice (Extended Data Fig. 6a). Additionally, body weights did not differ among CC-blue-NIR-mice, CC-red-NIR-mice and ChAT−ChR2 mice in which vehicle had been placed in the pancreatic ducts, followed by NIR light illumination (CC-veh-NIR-mice) (Extended Data Fig. 6b). Of note, in this model, food intakes of CC-blue-NIR-mice were similar to those of CC-red-NIR-mice and CC-veh-NIR-mice throughout the experimental periods (Extended Data Fig. 6c). Furthermore, body weights and food intakes were similar in CC-blue-NIR-mice and CC-red-NIR-mice even after 2 months of NIR illumination (Extended Data Fig. 6d,e), and body weights and food intakes were maintained at levels similar to those after 2 weeks of NIR illumination (Extended Data Fig. 6b,c), suggesting that LMP-induced chronic blue light emission had minimal adverse effects on the general conditions of ChAT−ChR2 mice. NIR-oVNS significantly increased pancreatic blood flows, whereas blood flows in the duodenum and the liver were unaffected (Extended Data Fig. 6f). In addition, in contrast to LED-oVNS (Extended Data Fig. 4b and Supplementary Videos 2 and 3), NIR-oVNS did not affect gastrointestinal tract motility (Extended Data Fig. 6g and Supplementary Videos 4 and 5). Therefore, the effects of blue lights from LMPs pooled in the pancreas are probably restricted to those exerted on the pancreas.

To examine whether selective activation of vagal nerves innervating the pancreas acutely stimulates insulin secretion, we performed GTTs. We applied NIR illumination with 973 nm NIR pulses (25 ms, 5 Hz) at a power of 35 W to freely moving LMP-loaded ChAT−ChR2 mice for 3 s per cage every 36 s (Fig. 4e and Supplementary Video 6) during the tests. Mouse body surface temperature was not elevated by 2 days of NIR illumination (Extended Data Fig. 6h). Notably, the blood insulin levels after glucose loading were markedly elevated in CC-blue-NIR-mice as compared with CC-red-NIR-mice (Fig. 5a). In addition, we confirmed that blood insulin levels after glucose loading were markedly elevated in CC-blue-NIR-mice as compared with the other two types of control: WT-blue-NIR-mice and ChAT−ChR2 mice loaded with blue light-emitting LMPs, followed by being maintained in the dark (CC-blue-dark-mice) (Supplementary Fig. 1a). Accordingly, blood glucose levels after glucose loading were lower in CC-blue-NIR-mice than in the three types of control, that is, CC-red-NIR-, WT-blue-NIR- and CC-blue-dark-mice, without eliciting hypoglycaemia (Fig. 5b and Supplementary Fig. 1b). Plasma levels of glucagon, somatostatin and amylase in CC-blue-NIR-mice were similar to those in CC-red-NIR-mice in both the fasting state and at 15 min after glucose loading (Extended Data Fig. 7a). In addition, amylase secretion into the duodenum was not altered by NIR-oVNS (Extended Data Fig. 7b). Furthermore, subdiaphragmatic vagotomy markedly blunted the enhancement of GSIS and reductions of blood glucose levels in CC-blue-NIR-mice (Fig. 5c). These findings indicate that stimulation of vagal nerves innervating the pancreas selectively enhances GSIS.

Acetylcholine, a major neurotransmitter secreted from vagal nerves, reportedly enhances GSIS from β cells through a mechanism dependent on the M3 muscarinic acetylcholine receptor[39]. Therefore, to examine whether M3 muscarinic signals are involved in the GSIS enhancement in CC-blue-NIR-mice, we performed GTTs on CC-blue-NIR-mice which had been pre-treated with 4-diphenylacetoxy-N-methylpiperidine methobromide (4-DAMP), a selective M3 receptor antagonist. Notably, 4-DAMP treatment markedly blunted the enhancement of GSIS in CC-blue-NIR-mice, and blood glucose level reductions in CC-blue-NIR-mice were reversed by 4-DAMP treatment (Fig. 5d). Collectively, these results indicate that selective activation of vagal nerves innervating the pancreas enhances GSIS through an M3 receptor-dependent mechanism in vivo.

## NIR light-mediated oVNS induces β cell proliferation

We next examined β cell masses in LMP-loaded ChAT−ChR2 mice after chronic NIR light-mediated selective stimulation of vagal nerves innervating the pancreas. Freely moving LMP-loaded ChAT−ChR2 mice were illuminated with NIR pulses under the same conditions as described above (Fig. 4e and Supplementary Video 6) for 2 weeks. After this chronic NIR illumination, pancreatic islets throughout the entire pancreas showed dramatic expansion (Fig. 6a), and β cell mass was markedly increased in CC-blue-NIR-mice to double that in CC-red-NIR-, WT-blue-NIR- and CC-blue-dark-mice (Fig. 6b and Supplementary Fig. 2a). Furthermore, BrdU-positive β cells were also markedly increased in CC-blue-NIR-mice as compared with those in CC-red-NIR-, WT-blue-NIR- and CC-blue-dark-mice (Fig. 6c and Supplementary Fig. 2b). Notably, β cell proliferative effects of NIR-oVNS were still present at 2 months (Extended Data Fig. 8a), and consequently, β cell mass of CC-blue-NIR-mice was markedly larger than that of CC-red-NIR-mice 2 months after starting oVNS (Extended

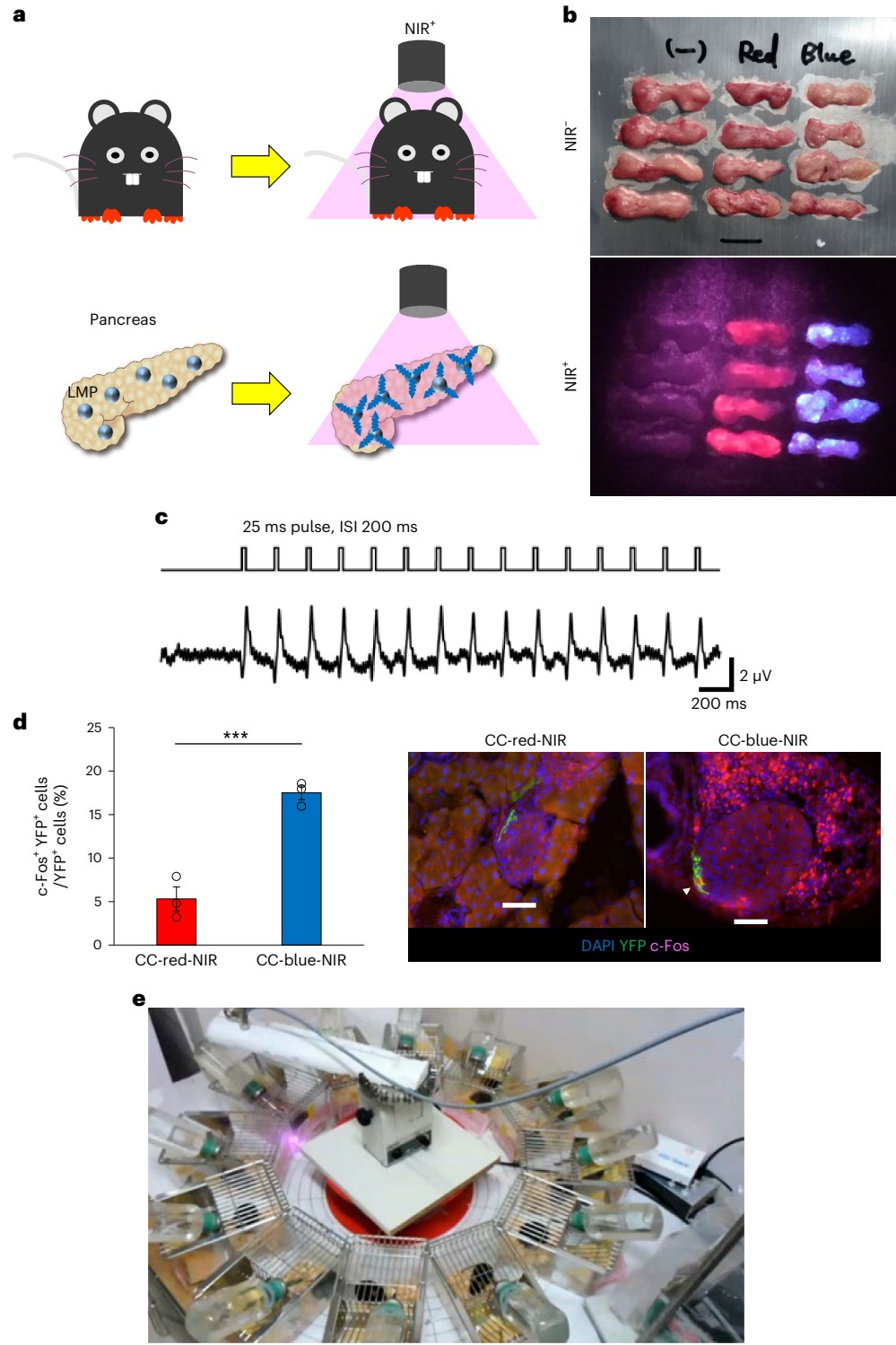

**Fig. 4 | Development of the NIR light-mediated oVNS method. a**, Illustration of the NIR-oVNS method, wherein blue light-emitting LMPs are injected into the pancreatic ducts of ChAT–ChR2 mice (referred to as CC-blue-NIR-mice). Following the injection, NIR light is externally applied from 20 cm above the mouse body to illuminate the bodies of freely moving mice. **b**, Excised pancreas collected 4 weeks after injection, containing red or blue LMPs, illuminated with NIR light. **c**, Intrapancreatic vagal nerve responses evoked by NIR illumination.

**d**, The ratio of c-Fos-positive ganglion cells to all YFP positive ganglion cells adjacent to the islets of ChAT–ChR2 mice after 1 week of chronic NIR-oVNS (two-tailed unpaired $t$-test (CC-red-NIR versus CC-blue-NIR), ***$P = 0.0016$, $n = 3$); representative images are shown in the right two panels. An arrowhead denotes a c-Fos-positive ganglion cell. Scale bars denote 50 µm. DAPI, 4′,6-diamidino-2-phenylindole. **e**, NIR illumination using a rotator. The rotational speed was 3 s per cage every 36 s.

Data Fig. 8b). As for α or δ cells, neither proliferations (Extended Data Fig. 9a) nor cellular sizes (Extended Data Fig. 9b) were increased in CC-blue-mice, while BrdU-positive exocrine cells were increased in

these mice (Extended Data Fig. 9a). In addition, insulin tolerance testing with or without concomitant NIR-oVNS revealed that insulin sensitivity of CC-blue-NIR-mice was similar to that of CC-red-NIR-mice after

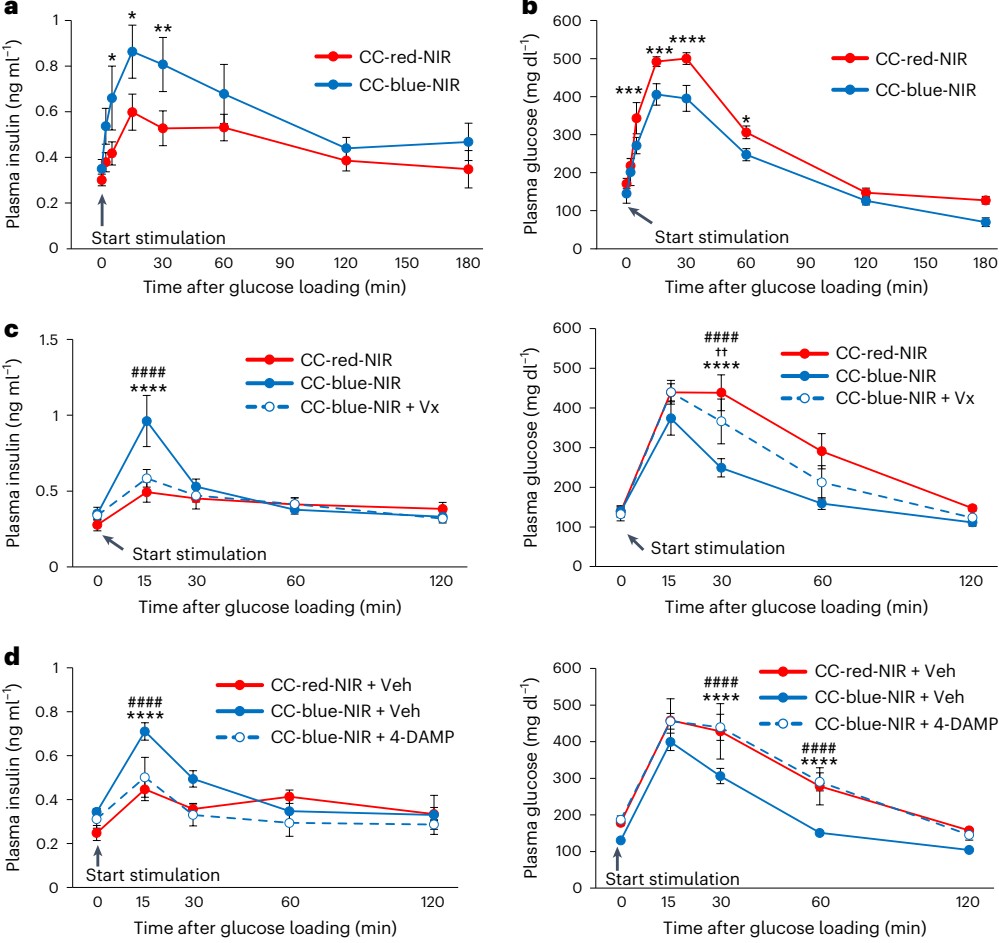

**Fig. 5 | NIR-oVNS elicited acute vagal nerve activation in vivo. a,b**, Plasma insulin (**a**) and glucose (**b**) levels of ChAT–ChR2 mice, which had undergone acute NIR-oVNS, during GTTs (**a**: two-way repeated measures ANOVA (CC-red-NIR versus CC-blue-NIR), *$P$ = 0.0262 at 5 min, *$P$ = 0.0152 at 15 min and **$P$ = 0.0092 at 30 min; **b**: two-way repeated measures ANOVA (CC-red-NIR versus CC-blue-NIR), ***$P$ = 0.0030 at 5 min, ***$P$ = 0.0029 at 15 min, ****$P$ = 0.000649 at 30 min and *$P$ = 0.0475 at 60 min, $n$ = 4 of CC-red-NIR, 5 of CC-blue-NIR). **c**, Plasma insulin (left) and glucose (right) levels of ChAT–ChR2 mice, which had undergone acute NIR-oVNS with or without subdiaphragmatic vagotomy (Vx), during GTTs (**c**: two-way repeated measures ANOVA followed by Bonferroni post hoc test; CC-red-NIR versus CC-blue-NIR plasma insulin at 15 min, ****$P$ = 4.434 × 10$^{-17}$; CC-blue-NIR versus CC-blue-NIR + Vx plasma insulin at 15 min, ####$P$ = 3.329 × 10$^{-15}$; CC-red-NIR versus CC-blue-NIR plasma glucose at 30 min, ****$P$ = 4.994 × 10$^{-10}$; CC-red-NIR versus CC-blue-NIR + Vx plasma glucose at 30 min, ††$P$ = 0.00495; CC-blue-NIR

versus CC-blue-NIR + Vx plasma glucose at 30 min, ####$P$ = 0.00000147, $n$ = 5 of CC-red-NIR, 7 of CC-blue-NIR, 8 of CC-blue-NIR + Vx). **d**, Plasma insulin (left) and glucose (right) levels of ChAT–ChR2 mice, which had undergone acute NIR-oVNS with or without administration of the muscarinic acetylcholine receptor M3 antagonist 4-DAMP, during GTTs (**d**: two-way repeated measures ANOVA followed by Bonferroni post hoc test; CC-red-NIR + Veh versus CC-blue-NIR + Veh plasma insulin at 15 min, ****$P$ = 6.461 × 10$^{-10}$; CC-blue-NIR + Veh versus CC-blue-NIR + 4-DAMP plasma insulin at 15 min, ####$P$ = 1.658 × 10$^{-7}$; CC-red-NIR + Veh versus CC-blue-NIR + Veh plasma glucose at 30 min, ****$P$ = 0.0000147; CC-blue-NIR + Veh versus CC-blue-NIR + 4-DAMP plasma glucose at 30 min, ####$P$ = 0.00000425; CC-red-NIR + Veh versus CC-blue-NIR + Veh plasma glucose at 60 min, ****$P$ = 0.00000833; CC-blue-NIR + Veh versus CC-blue-NIR + 4-DAMP plasma glucose at 60 min, ####$P$ = 0.00000219, $n$ = 4). Data are presented as mean ± s.e.m.

stimulation for 2 weeks (Extended Data Fig. 9c) at the timepoint when BrdU-positive β cells were markedly increased (Fig. 6c). Furthermore, ratios of TUNEL-positive β cells in CC-blue-NIR-mice were similar to those in CC-red-NIR-mice after the 1-week NIR stimulation (Extended Data Fig. 9d). Thus, neither alterations in systemic insulin sensitivity nor reduced β cell apoptosis was likely to be involved in the β cell mass increase produced by NIR-oVNS.

To further examine whether non-specific effects of LMP implantation and NIR illumination are exerted on β cell proliferation, we placed blue or red light-emitting LMPs in the pancreatic ducts of mice carrying only the Rosa–CAG–LSL–ChR2 (H134R)–EYFP cassette without the *Cre* gene, and applied NIR illumination to these mice for 2 weeks followed by evaluating their β cell masses. In contrast to ChAT–ChR2 mice, β cell masses were similar in mice with blue and red light-emitting LMPs after 2 weeks of NIR illumination (Extended Data Fig. 9e), suggesting that the impacts of either LMP implantation or NIR illumination alone

on β cell proliferation were minimal. Collectively, these results clearly demonstrate that signals from vagal nerves innervating the pancreas are sufficient to promote β cell proliferation.

Next, we explored the mechanisms by which vagal nerve activation promotes β cell proliferation. First, subdiaphragmatic vagotomy markedly blunted the increases in β cell mass in CC-blue-NIR-mice (Fig. 6d), confirming involvement of vagal nerves in promoting β cell proliferation in CC-blue-NIR-mice. We previously reported that vagal nerve-derived factors, including acetylcholine, enhance β cell proliferation through a FoxM1-dependent mechanism[12]. Therefore, we next examined gene expressions of *FoxM1* and its downstream target *Cdk1* in islets isolated from CC-blue-NIR-mice. Expressions of these genes were significantly increased in islet cells from CC-blue-NIR-mice (Fig. 6e), suggesting that a vagal nerve-derived factor(s) activates the β cell FoxM1 pathway and thereby enhances β cell proliferation in CC-blue-NIR-mice. Then, we treated CC-blue-NIR-mice with 4-DAMP

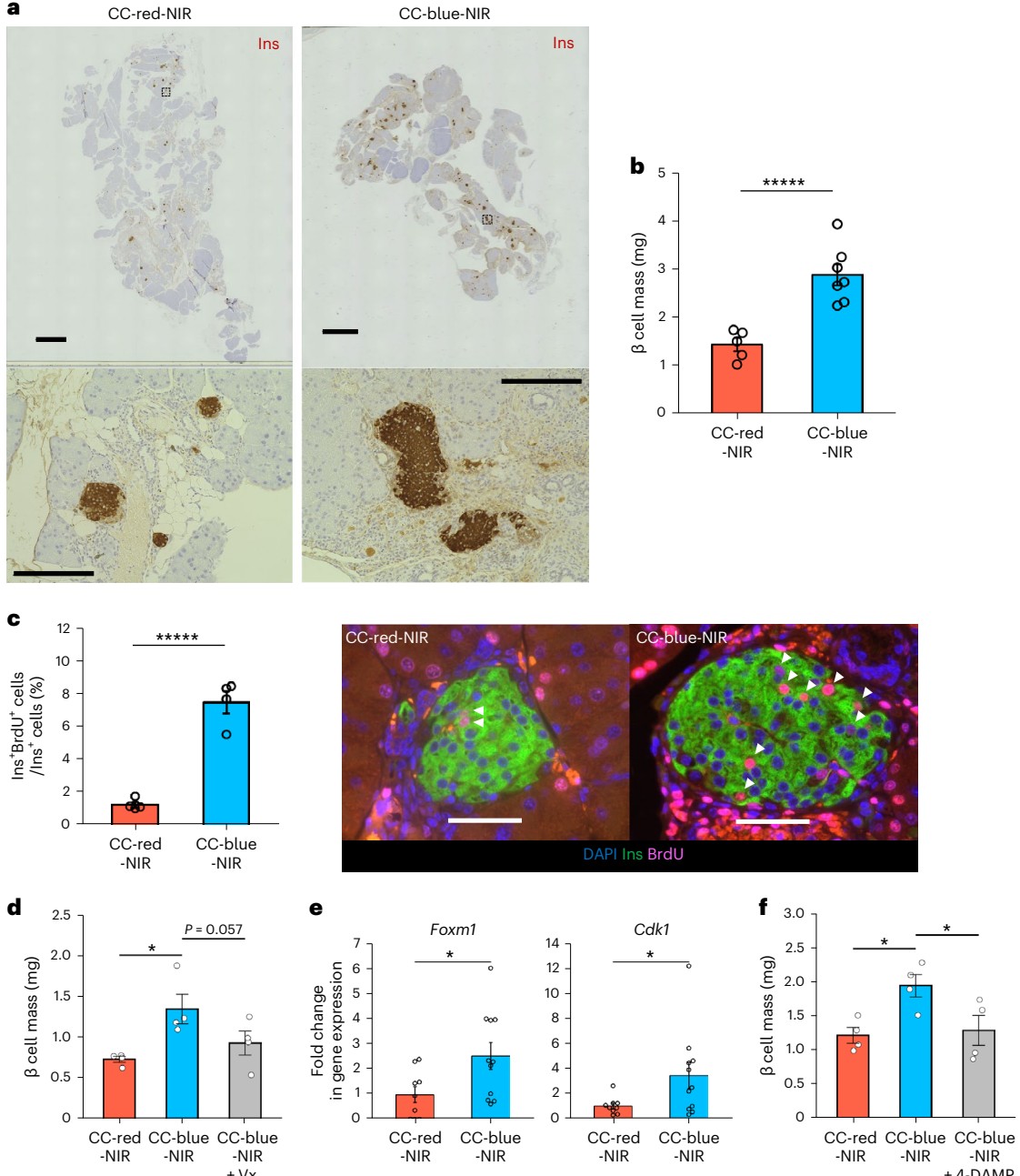

**Fig. 6 | Chronic NIR-oVNS induced pancreatic β cell proliferation. a,** Insulin immunostaining (brown) of pancreatic sections from ChAT−ChR2 mice after 2 weeks of chronic NIR-oVNS. Bottom two panels are magnified images of the inset. More than 12 ChAT−ChR2 mice were tested. Scale bars denote 2 mm (top two panels) and 200 μm (bottom two panels). Ins, insulin-positive areas. **b,** β cell masses of ChAT−ChR2 mice after 2 weeks of chronic NIR-oVNS (**b:** two-tailed unpaired *t*-test (CC-red-NIR versus CC-blue-NIR), *****P = 0.0005371, n = 5 of CC-red-NIR, 7 of CC-blue-NIR). **c,** The ratio of BrdU-positive β cells to all β cells in the islets of ChAT−ChR2 mice after 2 weeks of chronic NIR-oVNS (**c:** two-tailed unpaired *t*-test (CC-red-NIR versus CC-blue-NIR), *****P = 0.0001129, n = 4); representative images are shown in the right two panels. Each arrowhead denotes a BrdU-positive β cell. Scale bars denote 50 μm. Ins, insulin-positive β cells; DAPI, 4′,6-diamidino-2-phenylindole.

**d,** β cell masses of ChAT−ChR2 mice after 2 weeks of chronic NIR-oVNS with or without subdiaphragmatic vagotomy (**d:** one-way ANOVA followed by Ryan's method as a post hoc test; CC-red-NIR versus CC-blue-NIR, *P = 0.0104; CC-blue-NIR versus CC-blue-NIR + Vx, P = 0.0572, n = 4). **e,** Changes in cell cycle-related gene expressions in the islets from ChAT−ChR2 mice after 2 weeks of chronic NIR-oVNS (**e:** two-tailed unpaired *t*-test (CC-red-NIR versus CC-blue-NIR), for *Foxm1*, *P = 0.0313; for *Cdk1*, *P = 0.0498, n = 9 of CC-red-NIR, 11 of CC-blue-NIR). **f,** β cell masses of ChAT−ChR2 mice after 2 weeks of chronic NIR-oVNS with or without the administration of muscarinic acetylcholine receptor M3 antagonist 4-DAMP (**f:** one-way ANOVA followed by Ryan's method as a post hoc test, CC-red-NIR versus CC-blue-NIR, *P = 0.0152; CC-blue-NIR versus CC-blue-NIR + 4-DAMP, *P = 0.0242, n = 4). Data are presented as mean ± s.e.m.

throughout the NIR stimulation period. Notably, 4-DAMP treatment markedly blunted the increases in β cell mass in CC-blue-NIR-mice (Fig. 6f). Taking these findings together, NIR-oVNS promotes β cell proliferation involving M3 receptor-dependent cholinergic signalling.

## NIR-oVNS suppresses streptozotocin-induced hyperglycaemia

To explore whether the β cells augmented by NIR-oVNS are functional, we administered streptozotocin (STZ) to ChAT−ChR2 mice loaded with blue or red light-emitting LMPs, accompanied by NIR illumination, and

monitored the blood glucose levels of these mice (Fig. 7a). One week after the STZ administration, β cell masses were reduced, to similar extents, in both STZ-treated CC-red-NIR-mice and CC-blue-NIR-mice (Extended Data Fig. 10a). β cell apoptosis and macrophage infiltration of the islets during oVNS were similar in STZ-treated CC-blue-NIR-mice and CC-red-NIR-mice 1 week after starting oVNS (Extended Data Fig. 10b,c). Blood insulin levels were significantly increased in STZ-treated CC-blue-NIR-mice as compared with STZ-treated CC-red-NIR-mice after starting oVNS (Fig. 7b). Blood glucose levels of STZ-treated CC-blue-NIR-mice were significantly lower than those in STZ-treated CC-red-NIR-mice after starting oVNS (Fig. 7c), suggesting that blood glucose reductions during the early phase after starting oVNS are probably due to enhancement of insulin secretion from residual β cells by oVNS. Notably, blood glucose elevations were markedly suppressed in STZ-treated CC-blue-NIR-mice for the 2-month period of NIR stimulation (Fig. 7c). In addition, BrdU-positive β cells were markedly increased in STZ-treated CC-blue-NIR-mice as compared with STZ-treated CC-red-NIR-mice at both 3 weeks and 2 months after starting oVNS (Fig. 7d). As a result, β cell masses of STZ-treated CC-blue-NIR-mice 3 weeks after starting oVNS were significantly greater than those of STZ-treated CC-red-NIR-mice (Fig. 7e). Furthermore, β cell masses of STZ-treated CC-blue-NIR-mice were still increased 2 months after starting oVNS, while β cell masses of STZ-treated CC-red-NIR-mice were decreased 2 months after starting oVNS as compared with those measured 3 weeks after starting oVNS (Fig. 7e). Body weights of STZ-treated CC-blue-NIR-mice were maintained for 2 months, whereas those of STZ-treated CC-red-NIR-mice gradually decreased probably due to insulin deficiency (Extended Data Fig. 10d). These results indicate that NIR-oVNS augments functional β cell mass by inducing β cell proliferation, thereby suppressing STZ-induced hyperglycaemia.

## Discussion

Earlier studies[9–11], including our previous investigations[8,12], showed the necessity of vagal signals for the induction of β cell proliferation in several physiological and experimental settings. While this manuscript was undergoing revision, two additional studies were published. One demonstrated that optogenetic vagal-nerve activation enhanced GSIS in anaesthetized mice[40], the other that selective activation of vagal nerves innervating the pancreas by a chemogenetic approach enhanced GSIS in freely-moving mice[41]. In the present study, we showed that selective activation of the vagal nerves innervating the pancreas by NIR light-mediated oVNS enhanced GSIS, results which are consistent with those presented in previous reports. Importantly, we also show that selective vagal nerve activation alone is sufficient to induce marked pancreatic β cell proliferation, thereby increasing the mass of functional β cells substantially. We generated ChAT–ChR2 mice in which photosensitive ChR2–EYFP fusion protein is specifically expressed in cholinergic neurons and confirmed that vagal nerves of these mice precisely responded to blue light stimuli. Moreover, we developed two optogenetic experimental systems enabling us to apply selective and chronic vagal nerve stimulation in freely moving mice. Notably, taking advantage of blue light-emitting LMP, NIR illumination selectively activated vagal nerves innervating the pancreas. Considering the experimental strategy, NIR-oVNS probably activates both pre- and post-ganglionic vagal nerves in this system. In addition, the anatomical structure of the pancreatic ducts made LMPs spread throughout the pancreas, thereby probably achieving substantial oVNS effects in the entire pancreas. Thus, the NIR-oVNS system enabled us to clarify that selective stimulation of vagal nerves innervating the pancreas is sufficient to promote GSIS and to substantially increase β cell mass. These results suggest that, via the elaborate network of vagal nerves innervating the pancreas, vagal signals appear to play active roles in β cell physiology, probably via the integrative regulation of both the function and the volume of many islets within the entire pancreas, rather than by simply mediating modifications of GSIS.

We previously showed that vagal nerve-derived factors, including acetylcholine, enhance β cell proliferation[12]. In addition, vagal factor-mediated β cell proliferation was abolished in β cells of inducible β cell-specific FoxM1 knockout mice[12]. In CC-blue-NIR-mice in the present study as well, the FoxM1 pathway was activated in islets isolated from CC-blue-NIR-mice. Furthermore, treatment with a selective M3 receptor antagonist markedly blunted both enhancement of GSIS and increases in β cell mass in CC-blue-NIR-mice. As vagal factor-mediated β cell proliferation was abolished in β cells of inducible β cell-specific FoxM1 knockout mice[12], vagal factors enhance β cell proliferation through a FoxM1-dependent mechanism. These findings indicate that oVNS enhanced β cell proliferation via muscarinic M3 signal-mediated upregulation of the β cell FoxM1 pathway. A previous report showed that α cells secrete acetylcholine to promote GSIS in human islets[42]. Involvement of paracrine effects from α cells in vagal signal-mediated regulation of β cells in vivo remains elusive.

We have shown that vagal nerves innervating a peripheral organ are selectively activated. Since vagal nerves innervate a variety of organs and tissues in both thoracic and abdominal sites, optogenetic stimulation of dorsal motor neurons of the vagus in the brain stem[43] or electrical stimulation of the vagal trunk[44] may affect the fibres innervating many organs. Procedures for studying the significance of vagal actions on individual organs in vivo have as yet been unavailable. We confirmed the specificity of NIR-oVNS on vagal nerve fibres innervating the pancreas employing several approaches. First, we evaluated blood flows in abdominal organs and showed a selective increase in the pancreas. Second, we evaluated gastrointestinal tract movements in both CC-LED-mice and CC-blue-NIR-mice. LED-oVNS significantly enhanced gastrointestinal tract motility as compared with that in control mice. Contrary to LED-oVNS, NIR-oVNS did not affect gastrointestinal tract motility. Food intake does not increase in CC-blue-NIR-mice, unlike in CC-LED-mice, apparently due to the absence of enhancing gastrointestinal tract motility in CC-blue-NIR-mice. Collectively, these results suggest that selective activation of vagal nerves innervating the pancreas was achieved by the NIR-oVNS system. In this sense, the NIR-oVNS system is considered to be useful for achieving selective stimulation of the vagal fibres innervating an intended organ at any intended timepoints.

Whether vagal nerve signals are involved in basal β cell proliferation, that is, in the maintenance of β cell mass, is worthwhile investigating. Taking advantage of this system, the vagal nerves innervating the pancreas can be selectively activated. In addition, by employing mice in which archaerhodopsin or halorhodopsin is expressed[26] in vagal nerves, the vagal nerve-regulating system used in this study will allow suppression of vagal activity at any intended timepoints as well. It is difficult to achieve chronic inhibition of vagal nerve signals by vagotomy, because efferent vagal nerves undergo regrowth and reconnection after the vagotomy within a few months[45]. In this context, functional inhibition of the vagus at any intended timepoints and/or in a chronic manner is important for understanding physiological regulation of β cell mass as well as influences induced by pathological states, such as diabetes, obesity and ageing. Thus, the optogenetic systems that we developed may be useful for unravelling the role of the vagal nerve system in the long-term maintenance of whole-body homeostasis.

Unlike the situation in mice, sparse innervation of islets cells by the vagal fibres, in contrast to the rich innervations of sympathetic fibres, have been reported in humans[27,42,46,47]. On the other hand, rich vagal innervation of pancreatic islets in humans has also been shown[15]. Thus, the density of autonomic nerve innervation on pancreatic islets in humans remains elusive. Further research aimed at examining the effects of vagal nerve activation on β cell proliferation in humans is required for the application of our results in clinical settings. It is, however, noteworthy that vagal nerve activation actually increased β cells located in their physiological positions, within pancreatic islets, at least in mice. Regenerated β cells can benefit from the inherent innervation

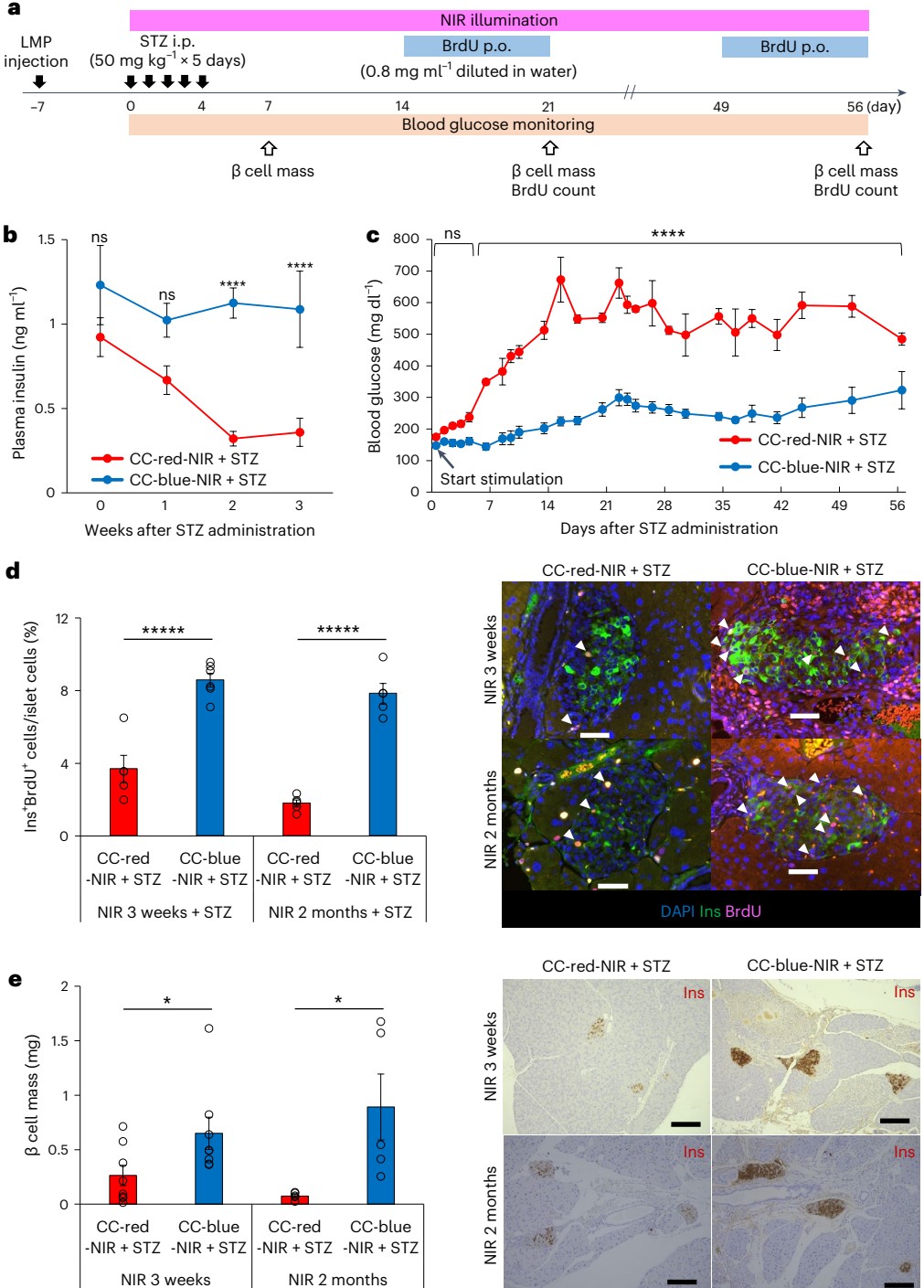

Fig. 7 | **Chronic NIR-oVNS suppressed STZ-induced hyperglycaemia. a**, Time course of experiment. i.p., intraperitoneal injection; p.o., oral administration. **b**, Plasma insulin levels of ChAT−ChR2 mice after starting STZ administration (50 mg kg[−1] i.p., 5 consecutive days) accompanied by 3 weeks of chronic NIR-oVNS (two-way repeated measures ANOVA, CC-red-NIR + STZ versus CC-blue-NIR + STZ at week 2, ****$P$ = 0.00006483; at week 3, ****$P$ = 0.0002235, $n$ = 5). **c**, Plasma glucose levels of ChAT−ChR2 mice after starting STZ administration (50 mg kg[−1] i.p., 5 consecutive days) accompanied by 2 months of chronic NIR-oVNS (two-way repeated measures ANOVA, CC-red-NIR + STZ versus CC-blue-NIR + STZ from day 6 to 56, ****$P$ = 5.039 × 10[−6], 2.345 × 10[−6], 1.933 × 10[−8], 3.061 × 10[−8], 4.940 × 10[−11], 1.893 × 10[−18], 1.335 × 10[−11], 5.361 × 10[−10], 8.275 × 10[−14], 1.886 × 10[−10], 8.108 × 10[−11], 5.111 × 10[−12], 3.974 × 10[−8], 5.265 × 10[−8], 2.367 × 10[−11], 2.371 × 10[−9], 1.458 × 10[−10], 1.272 × 10[−8], 1.026 × 10[−11], 2.273 × 10[−10], 2.467 × 10[−4] at days 6, 8, 9, 10, 13, 15, 17, 20, 22, 23, 24, 26, 28, 30, 34, 36, 38, 41, 44, 50 and 56, respectively, $n$ = 5). **d**, The ratio of BrdU-positive β cells to all islet cells of ChAT−ChR2 mice after starting STZ administration (as stated above) accompanied by 3 weeks or 2 months of chronic NIR-oVNS (two-tailed unpaired $t$-test, CC-red-NIR + STZ versus CC-blue-NIR + STZ at 3 weeks, *****$P$ = 0.0001868; CC-red-NIR + STZ versus CC-blue-NIR + STZ at 2 months, *****$P$ = 7.858 × 10[−6], $n$ = 5, 6, 5 and 5, respectively). Representative images are shown in the right two panels. Each arrowhead denotes a BrdU-positive β cell. Scale bars denote 50 μm. DAPI, 4′,6-diamidino-2-phenylindole. **e**, β cell masses of ChAT−ChR2 mice after starting STZ administration (as above stated) accompanied by 3 weeks or 2 months of chronic NIR-oVNS (two-tailed unpaired $t$-test, CC-red-NIR + STZ versus CC-blue-NIR + STZ at 3 weeks, *$P$ = 0.0444; CC-red-NIR + STZ versus CC-blue-NIR + STZ at 2 months, *$P$ = 0.0273, $n$ = 8, 8, 5 and 5, respectively). Representative images are shown in the right two panels. Scale bars denote 100 μm. Data are presented as mean ± s.e.m. NS, not significant.

and blood supply of the pancreas. Furthermore, insulin secreted from the regenerated β cells directly enters the portal vein. Thus, increasing β cell mass by vagal nerve activation may be a more physiological and advantageous therapeutic option than transplantation of pancreatic islets or stem cell-derived β cells into tissues besides the pancreas.

Cervically implanted electric vagal-nerve-stimulation devices have recently been applied in clinical situations. Stimulation of afferent vagal nerves is applied for epilepsy and depression, whereas efferent vagal stimulation is utilized for rheumatoid arthritis, inflammatory bowel disease[29] and arrhythmias[48]. In more recent years, a subdiaphragmatically implanted electric device, which de-activates efferent vagal nerves, was shown to reduce body weight by suppressing gastrointestinal tract motility in patients with obesity[35]. In addition, taking advantage of anatomical structure, the vagal nerve probably transmits signals directly to pancreatic β cells, thereby overcoming drug-delivery issues. Taken together with the present results, activation of vagal nerves is a potentially practical and promising therapeutic option for insulin-deficient diabetes. Impairment of limb regeneration[49,50] and retardation of liver regeneration[51] due to surgical denervation are widely recognized. We have also shown that vagal signals can be involved in acute liver regeneration after liver damage[52]. Therefore, the concept of autonomic nerve modulation being an option for regenerative therapy might expand to tissues and organs other than pancreatic islets and pancreatic tissues.

## Methods

### Ethics statement
All experiments in this study were conducted in accordance with the Tohoku University institutional guidelines. Ethics approval was obtained from the Institutional Animal Care and Use Committee of the Tohoku University Environmental & Safety Committee.

### Animals
Male ChAT-IRES-Cre mice[24] (Jax stock number 018957) and LSL-ChR2(H134R)-EYFP mice[26] (Jax stock number 024109) were purchased from Jackson Laboratory. To generate ChAT–ChR2 mice that are heterozygous for ChAT–IRES–Cre and homozygous for LSL–ChR2–EYFP, ChAT–IRES–Cre mice were crossed with LSL–ChR2–EYFP mice. Male wild-type mice (C57BL/6J) were purchased from CLEA Japan. All mice were housed individually in a controlled environment (room temperature 25 °C with 50% humidity) with a 12-h light–dark cycle, and received standard chow and drinking water ad libitum. Insulin-deficient diabetes model mice were created by intraperitoneal infusion of 50 mg kg$^{-1}$ body weight STZ (Sigma) for 5 consecutive days. STZ was dissolved in 0.05 M citrate sodium buffer (pH 4.5) and injected into the mice[53]. ChAT–ChR2 mice at 10–20 weeks of age were used in the experiments. Body temperature was measured using a BAT-12 (Physitemp Instruments). Animal studies were conducted in accordance with Tohoku University institutional guidelines.

### Immunohistochemistry
Excised pancreata were fixed in 10% formalin, embedded in paraffin and sectioned. Sections were stained with haematoxylin and eosin or incubated with primary antibodies: Tuj1 (Tubulin β-3) (1:500, catalogue no. 802001, BioLegend), insulin (1:2,000, no. I2018, Sigma), glucagon (1:400, no. 8233, Cell Signaling Technology), somatostatin (1:100, no. MAB354, Millipore), amylase (1:100, no. A8273, Sigma) and CD31 (1:100, no. 550274, BD Bioscience, San Jose). Immunoreactivity was visualized by incubation with a substrate solution containing 3,3′-diaminobenzidine. Alexa Fluor 488 goat anti-mouse IgG (1:500, no. ab150117, abcam), Alexa Fluor 488 goat anti-rabbit IgG (1:500, no. 4412, Cell Signaling Technology), Alexa Fluor 488 goat anti-rat IgG (1:500, no. A11006, Molecular Probes), Alexa Fluor 488 goat anti-rabbit IgG (1:500, no. A11008, Molecular Probes), Alexa Fluor 488 donkey anti-mouse IgG (1:200, no. 715-545-151, Jackson ImmunoResearch), Alexa Fluor 546 goat anti-rabbit IgG (1:500, no. A11010, Invitrogen) or

Alexa Fluor 594 donkey anti-mouse IgG (1:200, no. 715-585-151, Jackson ImmunoResearch) was used as the fluorescent secondary antibody. For detection of TUNEL-positive cells, pancreatic sections were stained with anti-TUNEL antibody (1:50, no. G3250, Promega). For detection of macrophages, pancreatic sections were stained with anti-Iba1 antibody (1:500, no. 019-19741, Wako) and Alexa Fluor 594 donkey anti-rabbit IgG (1:200, no. A21207, Life Technologies). For BrdU in situ detection, mice were administered BrdU (BD Bioscience) diluted in drinking water (0.8 mg ml$^{-1}$). The labelled cells were immunostained with anti-BrdU antibody (1:20, no. 551321, BD Bioscience) and streptavidin Alexa Fluor 594 (1:100, no. S32356, Invitrogen). BrdU-positive nuclei in pancreatic islets were counted and divided by all nuclei within islets. We counted the number of BrdU-positive nuclei in 2,000 islet cells per pancreatic specimen. For detection of c-Fos-positive parasympathetic ganglion cells, pancreatic sections were stained with anti-c-Fos antibody (1:1,000, no. ab222699, abcam) and Alexa Fluor 546 goat anti-rabbit IgG (1:500, no. A11010, Invitrogen). c-Fos-positive cells in YFP-positive parasympathetic ganglia of ChAT–ChR2 mice were counted and divided by all YFP-positive parasympathetic ganglion cells. We counted the number of c-Fos-positive cells in ~60–120 ganglion cells per pancreatic specimen. Tissue images were obtained using a BIOREVO BZ-X710, BIOREVO BZ-X700 Viewer (version 1.4.0.1) and BZ-X700 analyzer (version 1.4.1.1) (Keyence).

### Quantification of contacts between nerve and islet cells
We stained pancreatic specimens 20 μm in thickness from ChR2-EYFP mice using insulin, glucagon, somatostatin and CD31 as specific markers for β, α, δ and vascular endothelial cells, respectively, and estimated the proximity of these cells and YFP-positive nerve fibres. We obtained 3D images of the specimens using the sectioning function of the BIOREVO BZ-X700 Viewer and BZ-X700 analyzer (Keyence) and evaluated contact density by means of Imaris software's surface function (Bitplane, Belfast, UK) (version 9.6.0) as reported previously[27]. The frequency of contacts between nerve and islet cells was denoted as the 'contact density' of a particular cell type, calculated as the proportion of the total volume of the shared areas between nerve fibres and islet cells/total volume of islet cells.

### Islet cell size measurement
Average islet cell size was determined by calculating the quotient of the total areas immunostained with insulin, glucagon and somatostatin and the number of cells included in these areas by means of ImageJ (version 1.53c).

### Optical stimulation
Blue light (470 nm) was generated by the LED light source (Prizmatix or Doric Lenses). The timing of illumination was controlled via transistor–transistor–logic signals created with Master-8 (AMPI) or LED driver (Doric Lenses). Light was delivered through plastic optical fibres, and the final output power at the tip was ~10 mW. For electrophysiological experiments using anaesthetized mice, several patterns of photostimulation parameters were investigated. For chronic LED-oVNS experiments using free-moving mice, the following photostimulation parameters were adopted: 5 Hz, 25 ms, ON 10 s, OFF 50 s, 9:00–21:00 (12 h). Photostimulation with parameters of 5 Hz, 25 ms, ON 5 s and OFF 5 s, for 2 h during GTT without prior stimulation, was defined as 'acute' stimulation. Photostimulation with parameters of 5 Hz, 25 ms, ON 10 s and OFF 50 s, for 2 weeks, was defined as 'chronic' stimulation.

### Electrophysiology
The anterior subdiaphragmatic vagal trunk running along the oesophagus was exposed and placed on a pair of hook-shaped tungsten bipolar recording electrodes (Unique Medical) with ~1.25 mm separation. A ground electrode was placed on the ear. Nerve activities were amplified using DAM80 (WPI) and digitized using Micro1401-3 and Spike2

(CED) at a sampling frequency of 10 kHz. The strength of responses was calculated from the integration of variance from the baseline.

## Optical fibre implantation

Operations were performed on 14–20-week-old male ChAT–ChR2 mice, anaesthetized with an intraperitoneal injection containing a mixture of medetomidine (0.3 mg kg$^{-1}$), midazolam (4 mg kg$^{-1}$) and butorphanol tartrate (5 mg kg$^{-1}$). We made an abdominal incision on the ventral midline. A silicon cuff (inside/outside diameter 3/5 mm) attached to an optical fibre was wrapped around the subdiaphragmatic oesophagus along which the posterior vagal trunk ran. The other edge of the fibre was passed through the back wall of the peritoneal cavity using a needle. The edge was connected to the LED light source when starting the photostimulation experiment. On completion of the surgical procedures, the abdominal muscles and skin were sutured layer by layer with 6–0 silk.

## Pancreatic intraductal injections of LMPs and subdiaphragmatic vagotomy

ChAT–ChR2 mice were anaesthetized in the same way as described above. An abdominal incision was made on the ventral midline. The common bile duct was exposed and clamped near the liver using a clip. The injection was performed through the ampulla of Vater into the pancreatic duct from the duodenum using a 1-ml syringe. Blue- or red-emitting LMP (Shanghai Keyan Phosphor Technology) mixed with 500 μl of saline (20 mg ml$^{-1}$) was injected. After removing the clamp, the hole on the ampulla was sealed with 3M Vetbond Tissue Adhesive (3M). Subdiaphragmatic vagotomy was performed concomitantly with LMP implantation. When subdiaphragmatic vagotomy was concomitantly performed, the ventral and dorsal vagal trunks were both separated from the subdiaphragmatic oesophagus and transected. The abdominal muscles and skin were sutured layer by layer with 6–0 silk. At 7 days after these operations, GTTs were performed.

## NIR stimulation

NIR light (973 nm) was generated employing a LuOcean Mini (Lumics GmbH). Control light (850 nm) was generated by fibre-coupled LED M850F3 (Thorlabs). The timing of stimulation was controlled via transistor–transistor–logic signals created with AWG-100 (ELMOS). Light was delivered through a fibre cable (200 μm/0.22 numerical aperture), and the final output power at the tip was ~35 W. Using free-moving mice, the tip was set downward at a height of 20 cm above the animals and rotated employing a test tube mixer at 36 s per revolution. Each mouse cage was loaded with 3 s of NIR illumination every 36 s. NIR illumination was carried out, with parameters of 5 Hz, 25 ms, for 3 h during GTT without prior illumination or continuously for 2–8 weeks

## Blood analysis

Blood samples were collected from the tail veins of mice. Blood glucose was measured with Glutest Mint (Sanwa Kagaku)[54]. Plasma concentrations of insulin and glucagon were measured with enzyme-linked immunosorbent assay kits (insulin, Morinaga Institute of Biological Science; glucagon, Mercodia), plasma somatostatin was measured using Enzyme Immunoassay kits (Phoenix Pharmaceuticals), and plasma α-amylase activity was determined with an Enzychrom α-amylase Assay Kit (BioAssay Systems).

## GTT

ChAT–ChR2 mice were intraperitoneally injected with 2 mg kg$^{-1}$ glucose solution after a 10-h fast[55]. The mice underwent LED-oVNS or NIR-oVNS immediately after glucose loading. Photostimulation protocols were described above.

## Insulin tolerance test

Mice were intraperitoneally injected with 0.25 U kg$^{-1}$ of insulin solution diluted with saline in a fed state after 2 weeks of LED-oVNS and NIR-oVNS

had been conducted. Blood glucose levels were measured every 15 min. During ITT, neither LED-oVNS nor NIR-oVNS was conducted.

## β cell mass quantification

Pancreatic specimens were excised and fixed with 10% formalin, and embedded in paraffin. After insulin immunostaining, total pancreatic and insulin-positive areas of each section were measured using BIOREVO BZ-X710 and BZ-X Analyzer (Keyence)[56]. We calculated the average ratio of the total insulin-positive area to the total pancreatic area of all sections, and multiplied this ratio by total pancreatic weight to determine the β cell mass.

## 4-DAMP administration

As reported previously[57], 4-diphenylacetoxy-N-methylpiperidine methobromide (4-DAMP) (abcam, #ab120144) was mixed into saline and intraperitoneally administered at a concentration of 0.21 μmol kg$^{-1}$ body weight 10 min before GTT. In chronic NIR-oVNS experiments, 4-DAMP was administered twice a day during the experimental period.

## Laser microdissection

Pancreatic frozen sections 8 μm in thickness were placed on polyethylene naphthalate-coated slides (Leica Microsystems). Laser microdissections of the islets were performed on a Leica LMD7000 system (Leica Microsystems). Immediately after microdissection, total RNA was purified using an RNeasy Micro Kit (Qiagen) and then processed for quantitative polymerase chain reaction (PCR) analysis[58]. PCR protocol and the sequences of primers are described in our previous report[12]. PCR primer sequences were as follows: 'GCTCCATAGAAATGTGACATC' (mouse *FoxM1*-forward), 'AACCTTCACTGAGGGCTGTAAC' (mouse *FoxM1*-reverse), 'AAGAACCTGGACGAGAACG' (mouse *Cdk1*-forward), 'GTCATCAAAGTACGGGTGCT' (mouse *Cdk1*-reverse), 'GATGCCCTGAGGCTCTT' (mouse *Actb*-forward) and 'TGTGTTGGCATAGAGGTCTTTAC' (mouse *Actb*-reverse).

## Pancreatic bile juice collection

ChAT–ChR2 mice were anaesthetized as described above. We made an abdominal incision on the ventral midline. The common bile duct was exposed. PE-10 tubing was retained in the common bile duct, and pancreatic bile juice was collected for an hour.

## Intestinal motility measurement

ChAT–ChR2 mice were anaesthetized as described above. We made an abdominal incision on the ventral midline and exposed the intestines, which were marked with a blue tissue dye. We recorded videos of intestinal movements with a Nikon SMZ1270 microscope (Nikon) and tracked the trajectory of the dye mark using video motion analysis software Kinovea ver. 0.8.15 (https://www.kinovea.org).

## Organ blood flow measurement

ChAT–ChR2 mice were anaesthetized with 1.5% isoflurane and maintained with inhalation of a mixture of 66% nitric oxide plus 33% oxygen as a carrier gas through a face mask. We made an abdominal incision on the ventral midline. The pancreatic, hepatic and duodenal tissues were exposed, and the blood flow images of these organs were obtained and analysed using OMEGAZONE, Laser Speckle Blood Flow Imager (version 1.03) and Laser Image Analyzer (version 1.07.3) (Omegawave).

## NIR light intensity measurement

ChAT–ChR2 mice were anaesthetized in the same way as described above. An abdominal incision was made on the ventral midline. Mice were positioned in the abdominal position. An optical sensor connected to laser power meter 8230E (ADCMT) was inserted into the abdominal cavities. Then, NIR light intensities in many portions of the pancreas, from head to tail, were measured under the same NIR illumination conditions (see 'NIR stimulation').

## Statistical analysis

The investigators did not apply randomization and were not blinded to group assignment during the experiments or to the outcome assessments. All data were expressed as mean ± standard error of the mean (s.e.m.). The statistical significance of differences between two or among three groups was assessed using the unpaired $t$-test or one-way ANOVA followed by Ryan's method as a post hoc test. For the longitudinal analysis comparisons, two-way repeated measures ANOVA followed by the Bonferroni post hoc test was used. Statistical analysis was performed using BellCurve for Excel (version 4.02) and R (version 3.5.1).

## Reporting summary

Further information on research design is available in the Nature Portfolio Reporting Summary linked to this article.

## Data availability

The main data supporting the results in this study are available within the paper and its Supplementary Information. The raw and analysed datasets generated during the study are available for research purposes from the corresponding author on reasonable request. Source data are provided with this paper.

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

## Acknowledgements

This research was supported by the Japan Science and Technology Agency, JST, Moonshot R&D (grant number JPMJMS2023) (to H.K.), as well as by the Japan Agency for Medical Research and Development, AMED, under grant numbers JP21gm5010002h0005 (to H.K.) and 21gm6210002h0004 (AMED-PRIME) (to J.I.).This work was also supported by Grants-in-Aid for Scientific Research to J.I. (22H03124 and 22K19303), T.I. (20K17525) and H.K. (20H05694) from the Japan Society for the Promotion of Science. We thank T. Takasugi, K. Watanabe, S. Goto, Y. Yoshizawa, M. Iwama, K. Takahashi and H. Yokoyama (all of whom belong to the Department of Metabolism and Diabetes, Tohoku University Graduate School of Medicine) for technical support.

## Author contributions

Y.K. and J.I. conducted the research, obtained the data and contributed to discussions and to writing, reviewing and editing the manuscript. Y.M.M., Y.I. and K.M. obtained the data regarding electrophysiological responses of vagal nerves, and contributed to discussions. A.I., K.N. and T.T. obtained the data regarding organ blood flows and contributed to discussions. A.Y. offered LMPs and contributed to discussions. M.K., H. Komamura, T.S., T.I., J.Y., A.E., H.S., H. Kubo, S.H., Y.M., Y.A., S.K., K.T., K.K., S.S. and T.Y. contributed to discussions. H. Katagiri contributed to discussions, and to writing reviewing and editing the manuscript.

## Competing interests

The authors have no competing interests to declare.

## Additional information

**Extended data** is available for this paper at https://doi.org/10.1038/s41551-023-01113-2.

**Correspondence and requests for materials** should be addressed to Junta Imai.

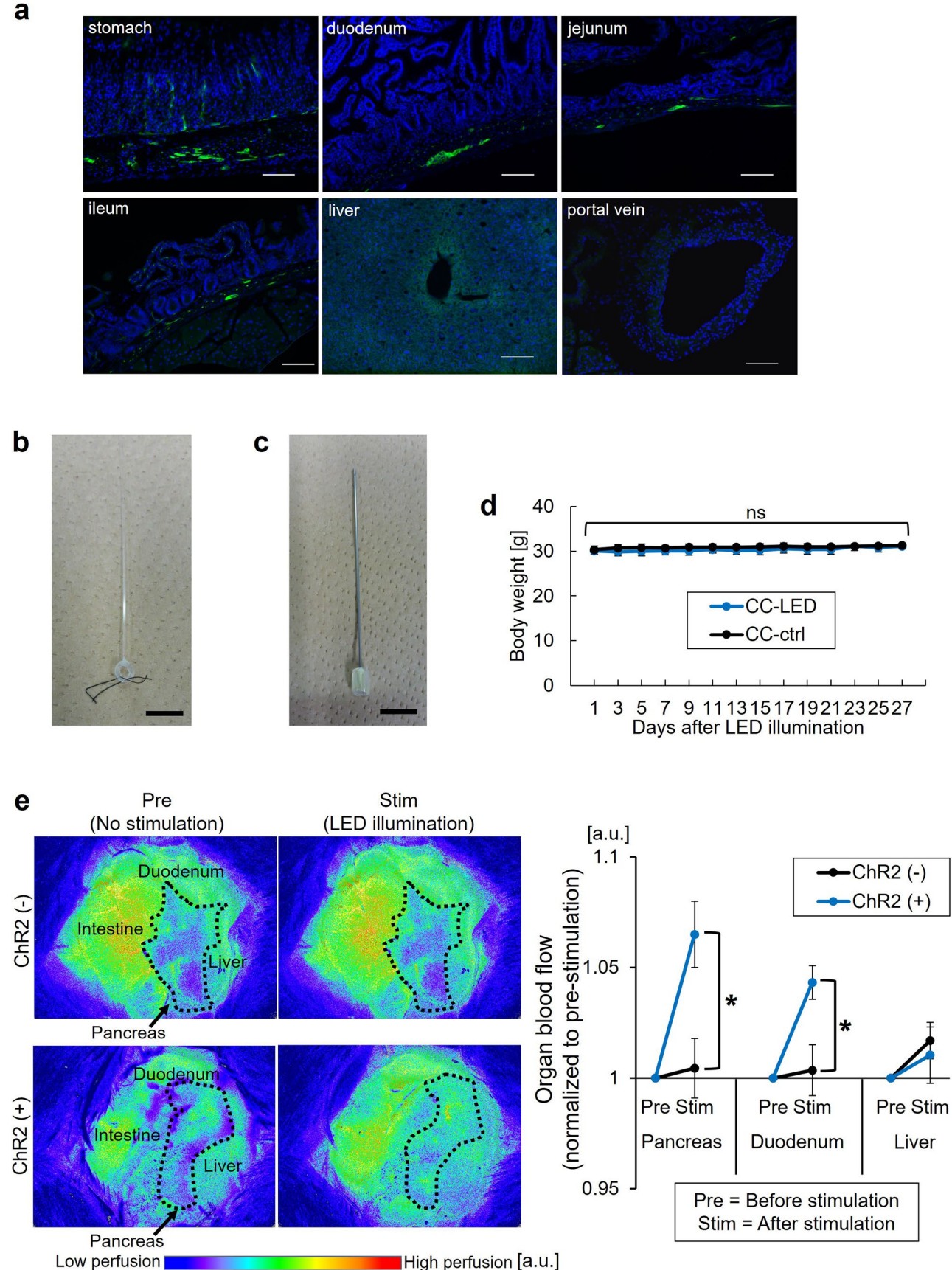

**Extended Data Fig. 1 | See next page for caption.**

**Extended Data Fig. 1 | Development of the LED-oVNS system. a**, YFP-positive parasympathetic ganglia and fibres were detected in the stomach, duodenum, jejunum and ileum, but not in the liver or the portal vein. More than two ChAT-ChR2 mice were tested. Scale bars denote 100 µm. **b**, A plastic optical fibre (500 µm diameter) was attached to a silicon cuff. The scale bar indicates 1 cm. **c**, A needle was passed through the back wall of the abdominal cavity. Its hub was wrapped with a silicon tube. The scale bar indicates 1 cm. **d**, Body weights of ChAT-ChR2 mice during 4 weeks of chronic LED-oVNS (n = 5). **e**, Pancreatic blood flow change after LED-oVNS. (Left) representative organ blood flow images from a non-ChAT-ChR2 (ChAT-Cre(+); LSL-ChR2(−)) or a ChAT-ChR2 mouse before ('Pre') and during ('Stim') LED-oVNS, obtained by laser speckle flowmetry. Each area enclosed by the dotted line denotes the pancreas. We captured one image per second during each period (10 sec respectively). (Right) quantification of abdominal organ blood flow. Averaged blood flow levels were calculated from 10 images per period and normalised to the Pre period (two-tailed unpaired t-test, ChR2(−)-pancreas vs ChR2(+)-pancreas during the Stim period, *P = 0.0219; ChR2(−)-duodenum vs ChR2(+)-duodenum during the Stim period, *P = 0.0322, n = 5). a.u.: arbitrary units. Data are presented as means ± s.e.m. ns: not significant.

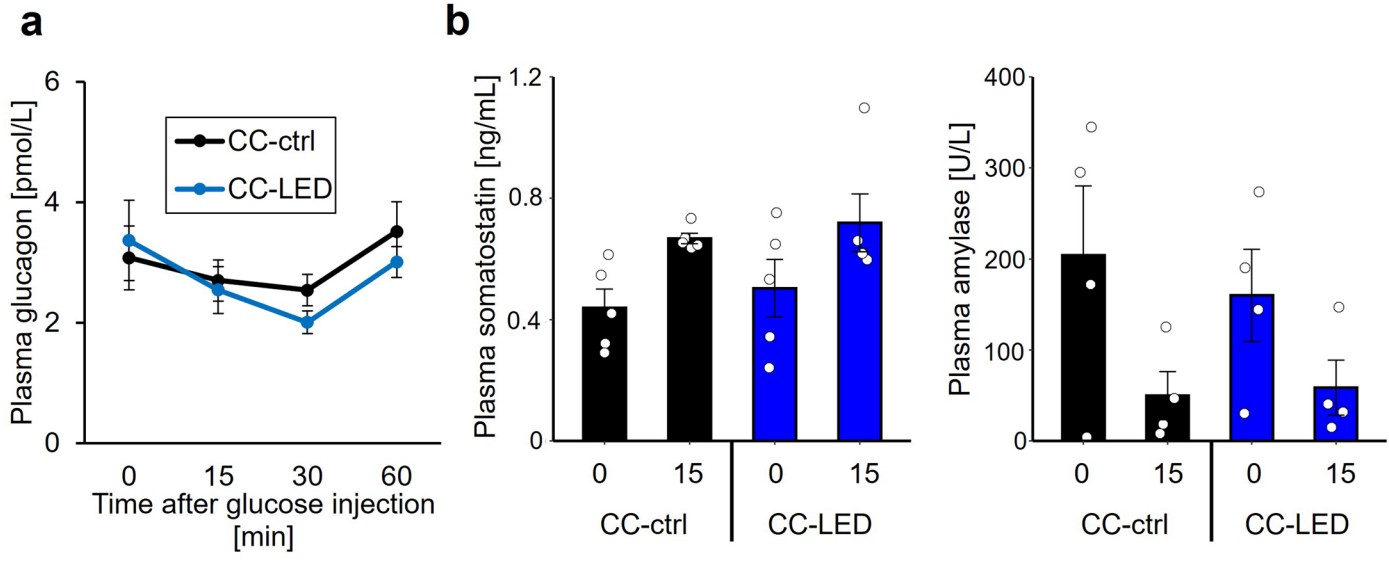

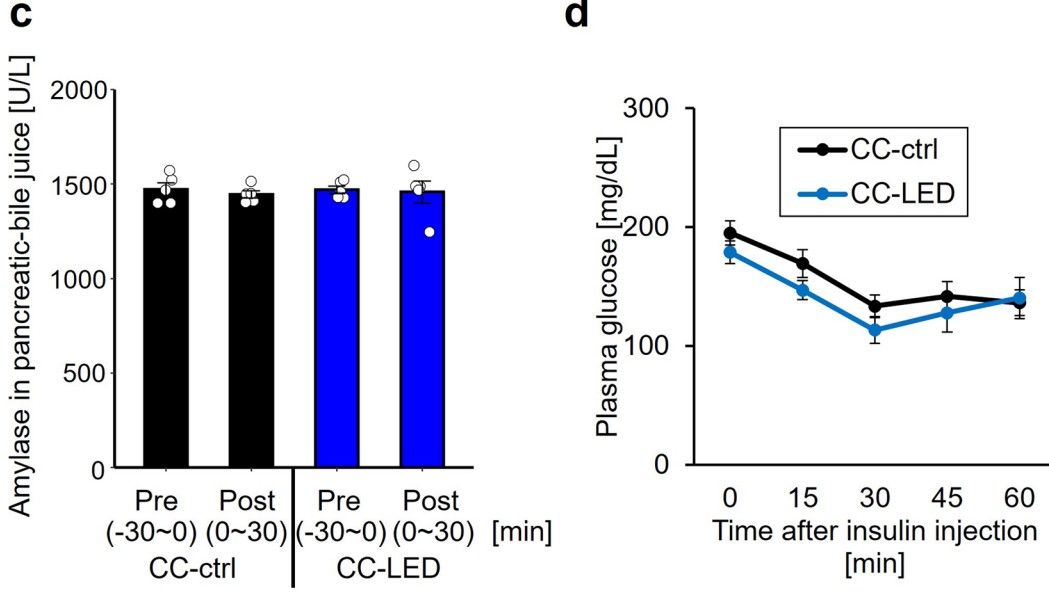

**Extended Data Fig. 2 | Other endocrine and exocrine hormone levels in LED-oVNS mice. a**, Plasma glucagon levels of ChAT-ChR2 mice, which had undergone acute LED-oVNS, during glucose tolerance tests (n = 5 of CC-ctrl, 6 of CC-LED). **b**, Plasma somatostatin and amylase levels of ChAT-ChR2 mice, which had undergone acute LED-oVNS, during glucose tolerance tests. Blood samples were collected at 0 and 15 min after starting LED-oVNS (n = 5 of somatostatin, 4 of amylase). **c**, Amylase concentrations in pancreatic-bile juice of ChAT-ChR2 mice before ('Pre' period) and after ('Post' period) starting LED-oVNS (n = 5). Pancreatic-bile juice was collected for 30 min during each period. **d**, Plasma glucose levels of ChAT-ChR2 mice, which had undergone acute LED-oVNS, during insulin tolerance tests (n = 5 of CC-ctrl, 6 of CC-LED). Data are presented as means ± s.e.m.

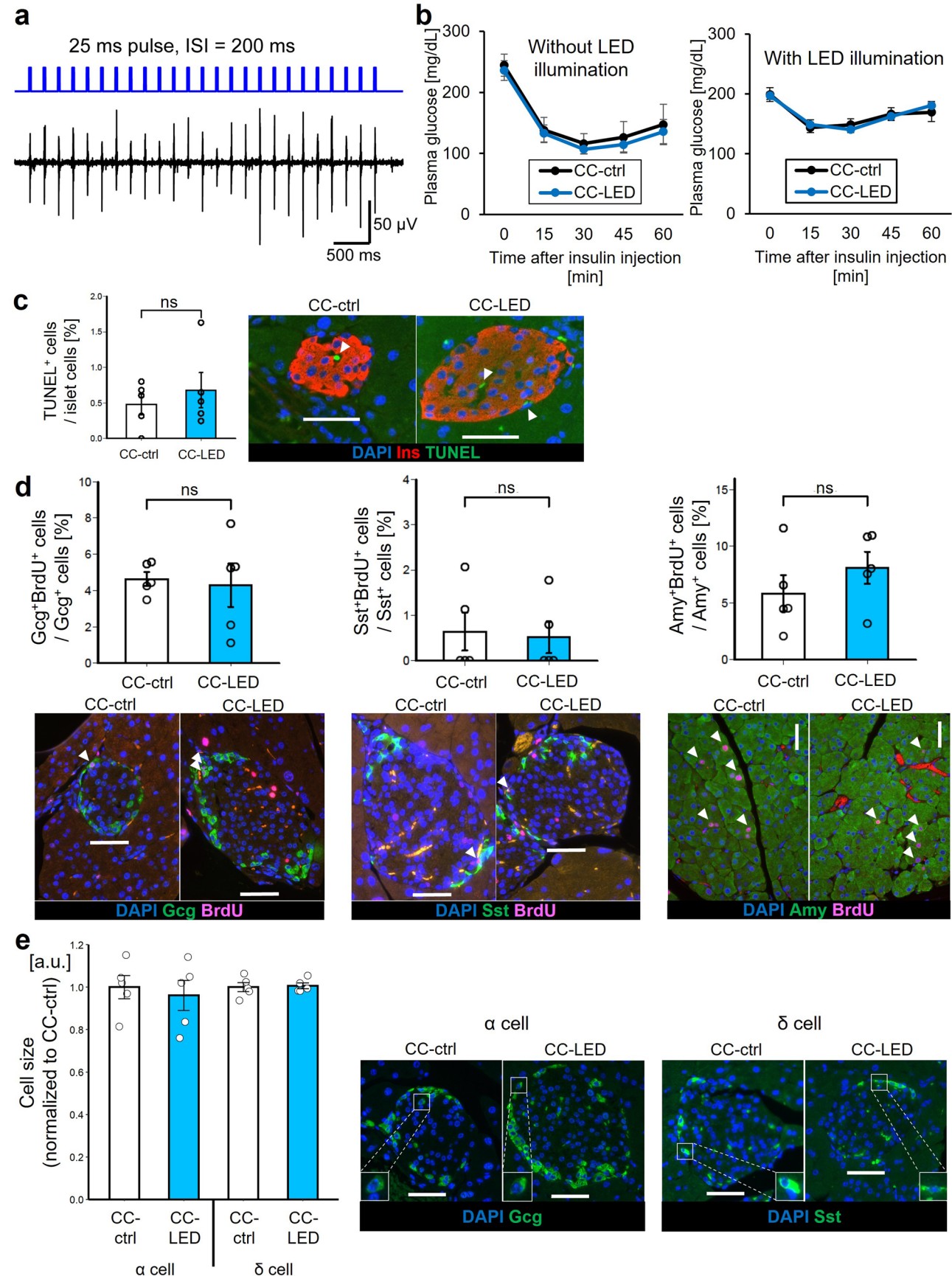

**Extended Data Fig. 3 | See next page for caption.**

**Extended Data Fig. 3 | Chronic LED-oVNS did not affect oesophageal vagal nerve responses, insulin sensitivity, apoptosis of beta-cells, non-beta-cell proliferation or non-beta-cell size. a**, Vagal nerve responses to blue LED pulses were still stably detected after 2 weeks of LED-oVNS. **b**, Plasma glucose levels of ChAT-ChR2 mice, which had undergone 2 weeks of chronic LED-oVNS, during insulin tolerance tests concomitantly with or without LED-oVNS (n = 6 of CC-ctrl without LED illumination, 6 of CC-LED without LED illumination, 5 of CC-ctrl with LED illumination, and 6 of CC-LED with LED illumination). **c**, The ratio of TUNEL positive beta-cells to all cells in the islets of ChAT-ChR2 mice after 2 weeks of chronic LED-oVNS (n = 5, two-tailed unpaired t-test, CC-ctrl vs CC-LED); representative images are shown in the right two panels. Each arrowhead denotes a TUNEL-positive beta-cell. Scale bars denote 50 μm. **d**, The ratio of BrdU positive non-beta-cells (α, δ and exocrine acinar cells) to the total number of each cell type in the islets or exocrine tissues of ChAT-ChR2 mice after 2 weeks of chronic LED-oVNS (n = 5, two-tailed unpaired t-test, CC-ctrl vs CC-LED); representative images are shown in the bottom two panels. Each arrowhead denotes a BrdU-positive non-beta-cell. Scale bars denote 50 μm. **e**, The average sizes of non-beta-cells (α and δ cells) of ChAT-ChR2 mice after 2 weeks of chronic LED-oVNS (n = 5); respective representative images are shown in the right two panels (white boxes indicate α and δ cells). Cell size is expressed as the ratio of the hormone-positive area to cell number. Scale bars denote 50 μm. Data are presented as means ± s.e.m. ns: not significant. Ins: insulin-positive beta cells, Gcg: glucagon-positive α cells, and Sst: somatostatin-positive δ cells.

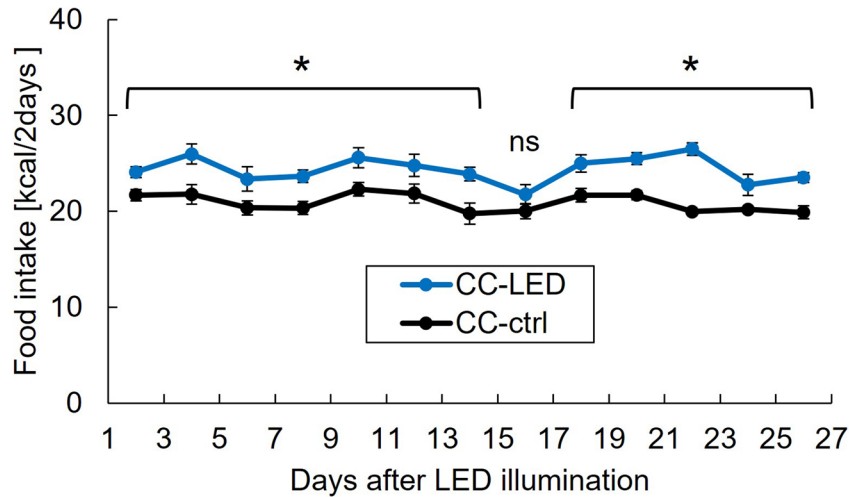

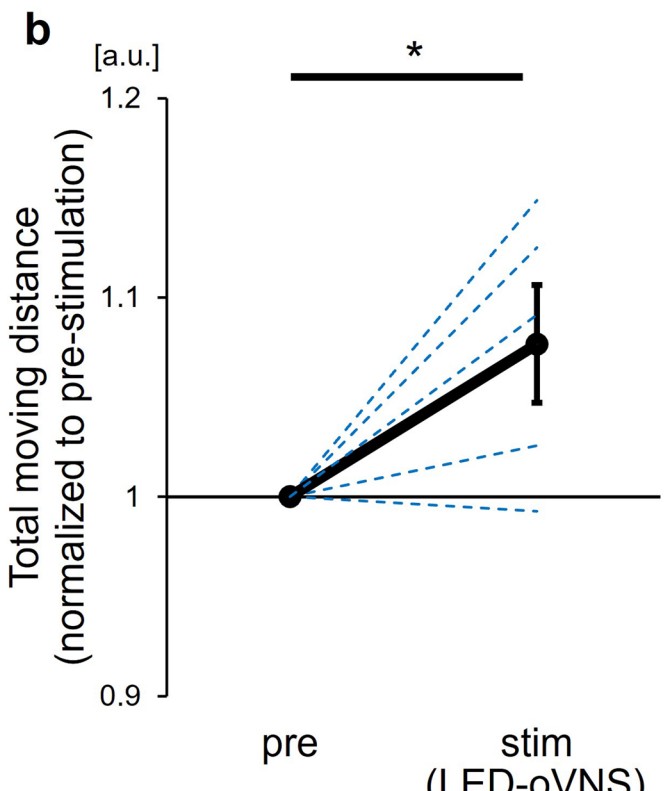

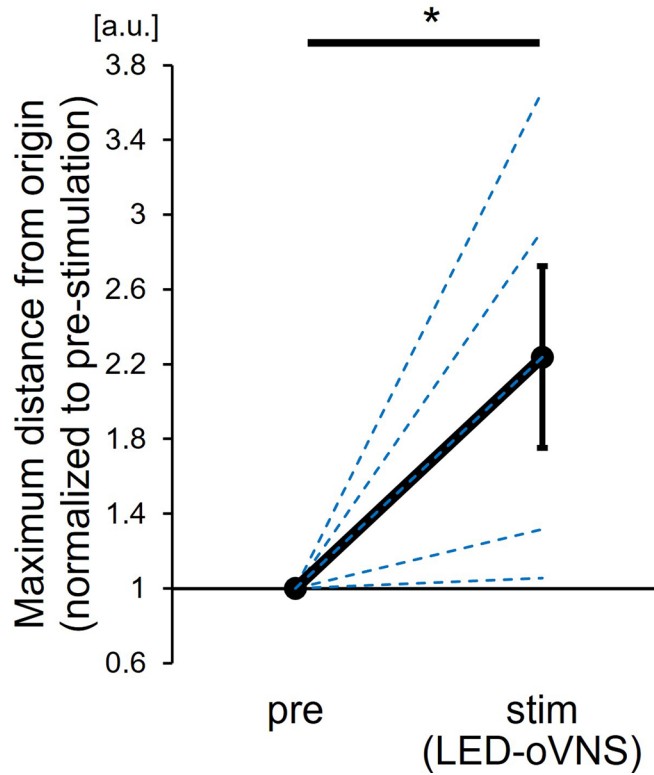

**Extended Data Fig. 4 | LED-oVNS promoted food intake and intestinal motility. a**, Food intake amounts of ChAT-ChR2 mice during 4 weeks of chronic LED-oVNS (two-way repeated measures ANOVA, CC-ctrl vs CC-LED, *P = 0.04781, 0.0009853, 0.01422, 0.007765, 0.007832, 0.01724, 0.001291, 0.007501, 0.00257, 0.000002757, 0.03505, 0.003765, at day 1-3, 3-5, 5-7, 7-9, 9-11, 11-13, 13-15, 17-19, 19-21, 21-23, 23-25, 25-27, respectively, n = 5). **b**, Intestinal motility change after LED-oVNS. As shown in Supplementary Video 2a, b, by tracing the trajectory of the mark on the intestine, total distance moved and maximum distance from the origin were measured for 10 sec before ('Pre') and after ('Stim') starting LED-oVNS (b: two-tailed unpaired t-test (Pre vs Stim), for the total moving distance, *P = 0.0313; for the maximum distance from the origin, *P = 0.0345, n = 5). Solid lines and dotted lines indicate average and individual values, respectively. Data are presented as means ± s.e.m.

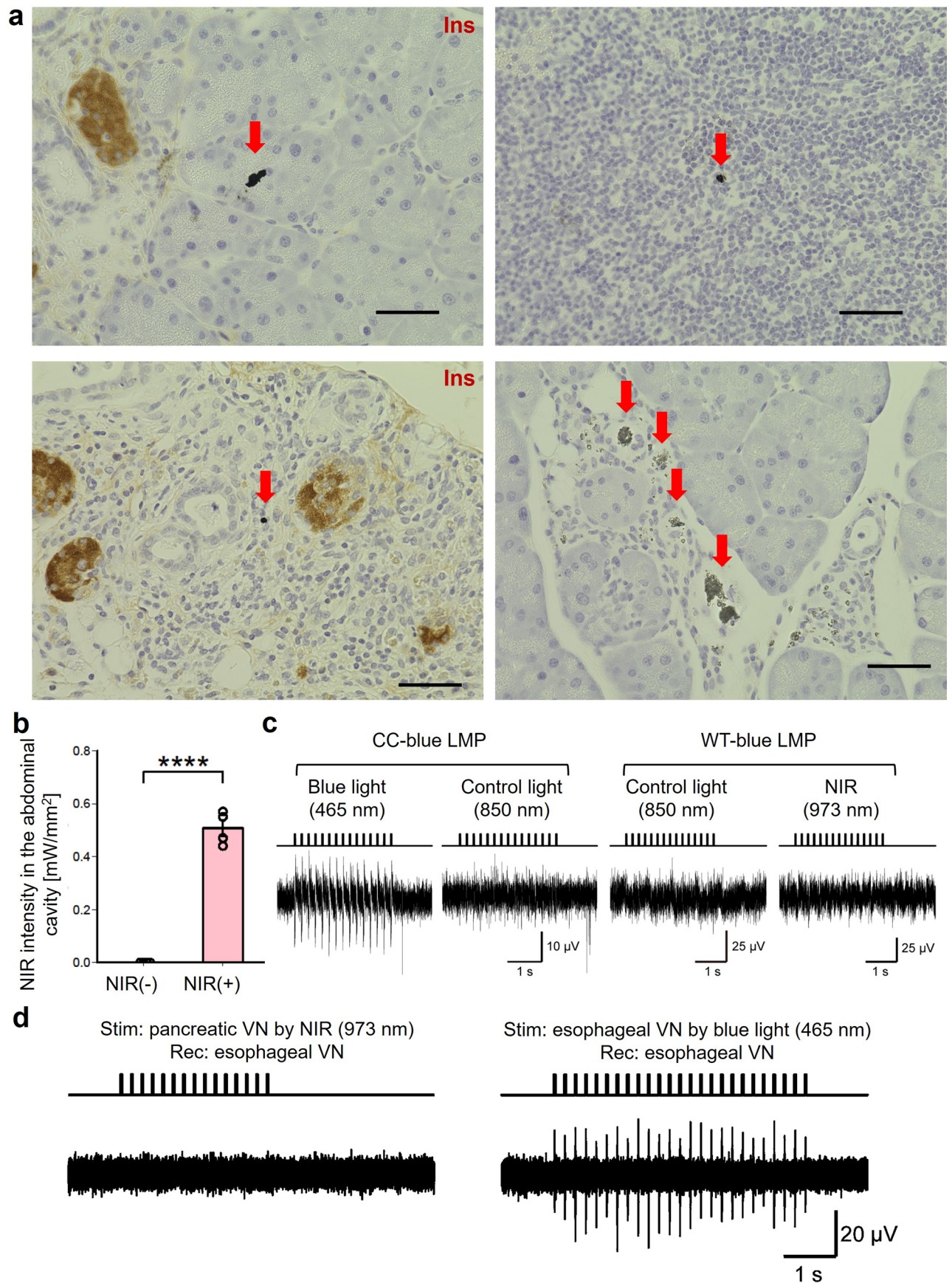

**Extended Data Fig. 5 | See next page for caption.**

**Extended Data Fig. 5 | Development of the NIR-oVNS system. a**, Distribution of the LMPs injected into the pancreatic duct. LMPs were found in exocrine tissues (left 2 panels), lymphatic tissues (top right), and the pancreatic ducts (bottom right). More than two ChAT-ChR2 mice were tested. Each of the red arrows indicates LMPs. Scale bars denote 50 μm. **b**, NIR light intensities in the pancreas, from head to tail (b: two-tailed unpaired t-test (NIR(−) vs NIR(+)), ****P = 0.0003431, n = 4). **c**, Intrapancreatic vagal nerve responses of ChAT-ChR2 and WT mice, in which blue light-emitting LMPs had been implanted, to various wavelengths of light. **d**, Left: lack of oesophageal vagal nerve responses of ChAT-ChR2 mouse in which intrapancreatic vagal nerves had been activated by NIR-oVNS. Right: evident oesophageal vagal nerve responses of the same mouse in which oesophageal vagal nerves had been activated by LED-oVNS. Rec: recording site. Data are presented as means ± s.e.m.

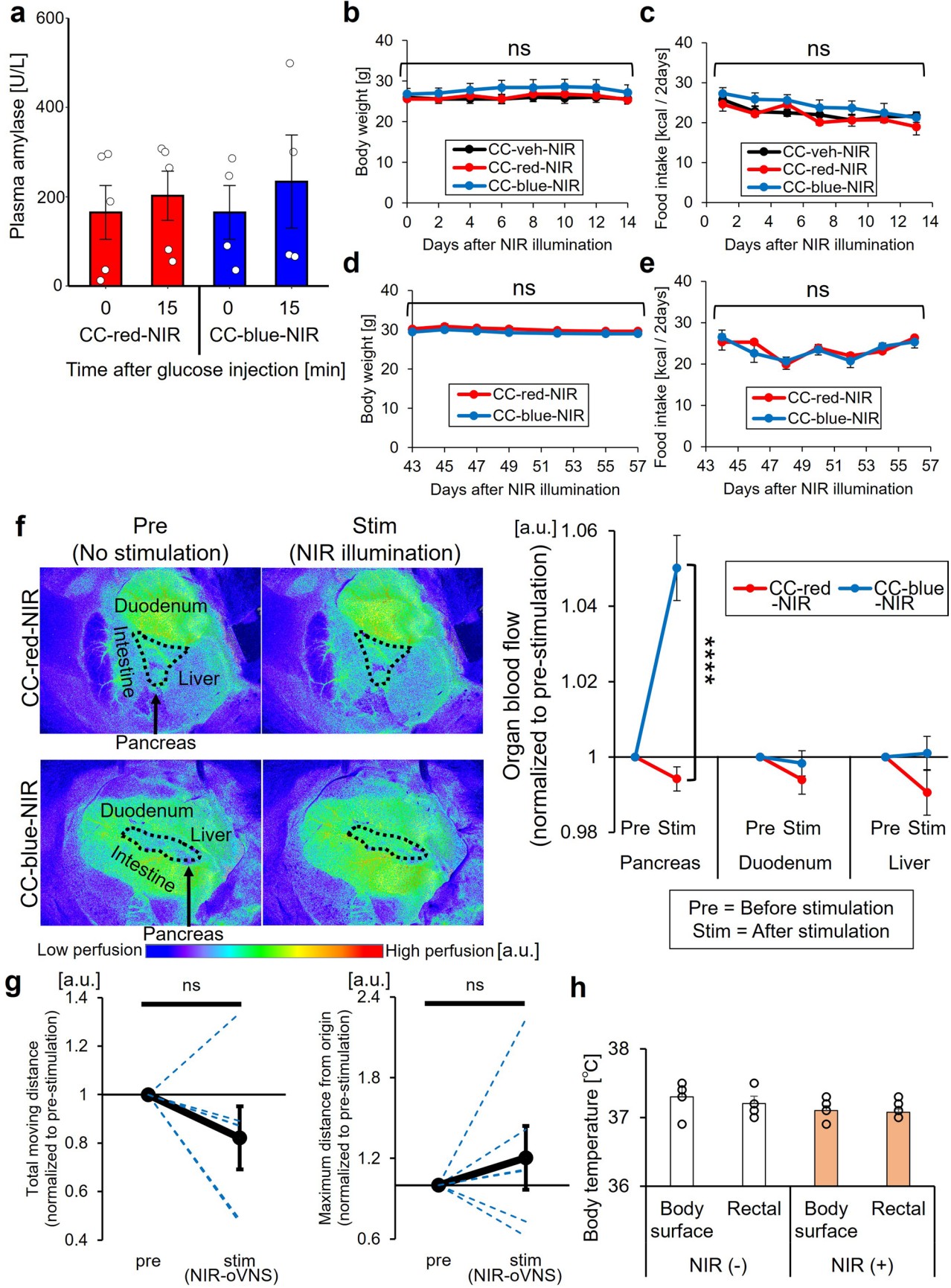

**Extended Data Fig. 6 | See next page for caption.**

**Extended Data Fig. 6 | Characterization of NIR-oVNS mice. a**, Plasma amylase levels of ChAT-ChR2 mice, which had undergone acute NIR-oVNS, during glucose tolerance tests. Blood samples were collected at 0 and 15 min after starting NIR-oVNS (n = 5 of CC-red-NIR, 4 of CC-blue-NIR). **b–e**, Body weights (b,d) and food intake amounts (c,e) of ChAT-ChR2 mice during 2 weeks of NIR-oVNS or the last 2 weeks of 2-month-NIR-oVNS (n = 5, two-way repeated measures ANOVA followed by Bonferroni post hoc test). **f**, Abdominal organ blood flow change after NIR-oVNS. (Left) Representative organ blood flow images of red or blue LMP-loaded ChAT-ChR2 mice before ('Pre') and after 3 seconds of NIR-oVNS ('Stim'), values were obtained by laser speckle flowmetry. Area enclosed by the dotted line denotes the pancreas. We captured one image per second during each period (10 sec, respectively). (Right) Quantification of blood flow change in each organ. Averaged blood flow levels were calculated from 10 images per period and normalised to that of the Pre period (f: two-tailed unpaired t-test (CC-red-NIR pancreas vs CC-blue-NIR pancreas during Stim period), ****P = 0.000934, n = 4 of CC-red-NIR, 5 of CC-blue-NIR). a.u.: arbitrary units. **g**, Intestinal motility change after NIR-oVNS. As shown in Supplementary Video 3a, b, by tracing the trajectory of the mark on the intestine, total distance moved and maximum distance from the origin were measured for 10 sec before ('Pre') and after ('Stim') starting NIR-oVNS. Solid lines and dotted lines indicate average and individual values, respectively (n = 6, two-tailed unpaired t-test, Pre vs Stim). **h**, Body temperature of mouse after 2-day-NIR illumination (n = 4). Data are presented as means ± s.e.m. ns: not significant.

## a

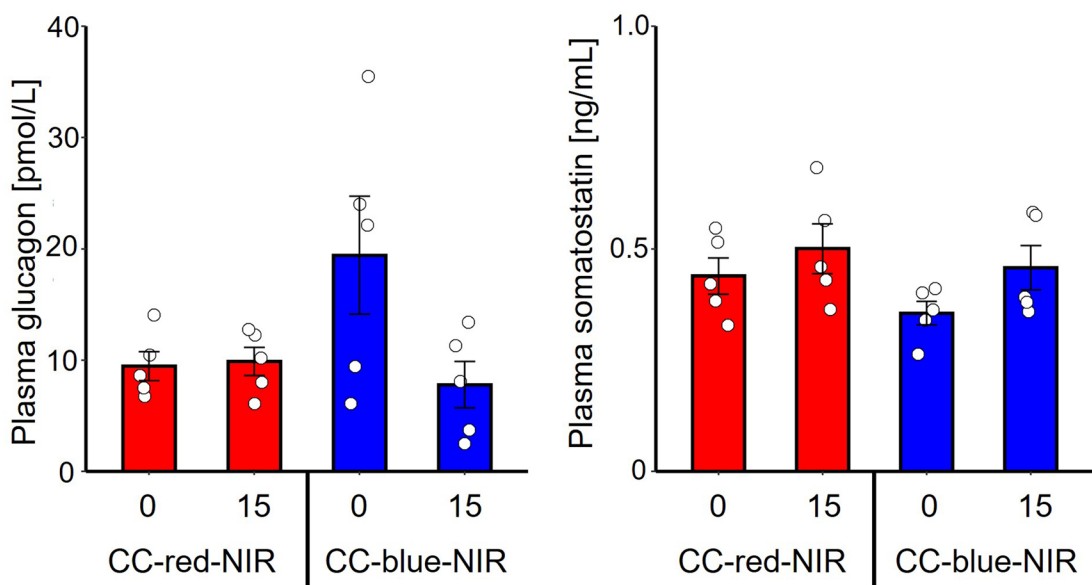

Time after glucose injection [min]

## b

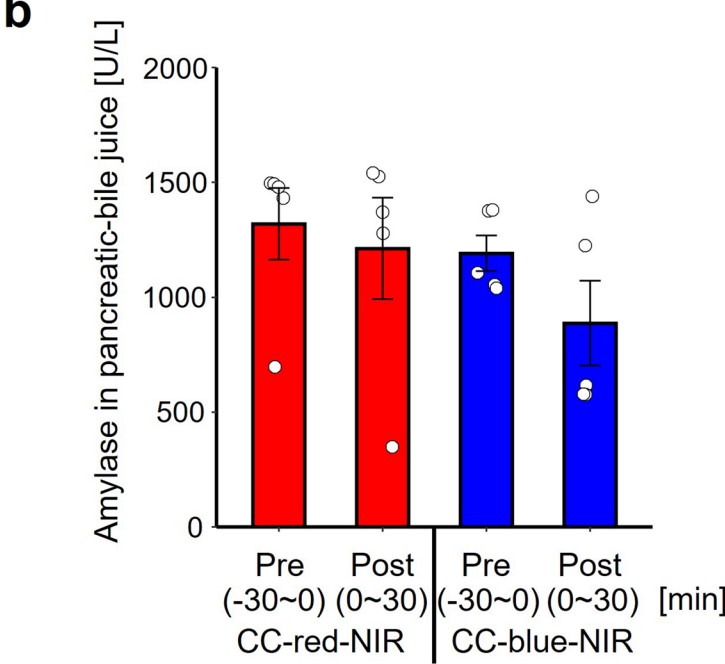

**Extended Data Fig. 7 | Other endocrine and exocrine hormone levels of NIR-oVNS mice. a**, Plasma glucagon and somatostatin levels of ChAT-ChR2 mice, which had undergone acute NIR-oVNS, during glucose tolerance tests. Blood samples were collected at 0 and 15 min after starting NIR-oVNS (n = 5). **b**, Amylase concentrations in pancreatic bile juice from ChAT-ChR2 mice before ('Pre' period) and after ('Post' period) starting NIR-oVNS (n = 5). Pancreatic-bile juice was collected for 30 min during each period. Data are presented as means ± s.e.m.

**a**

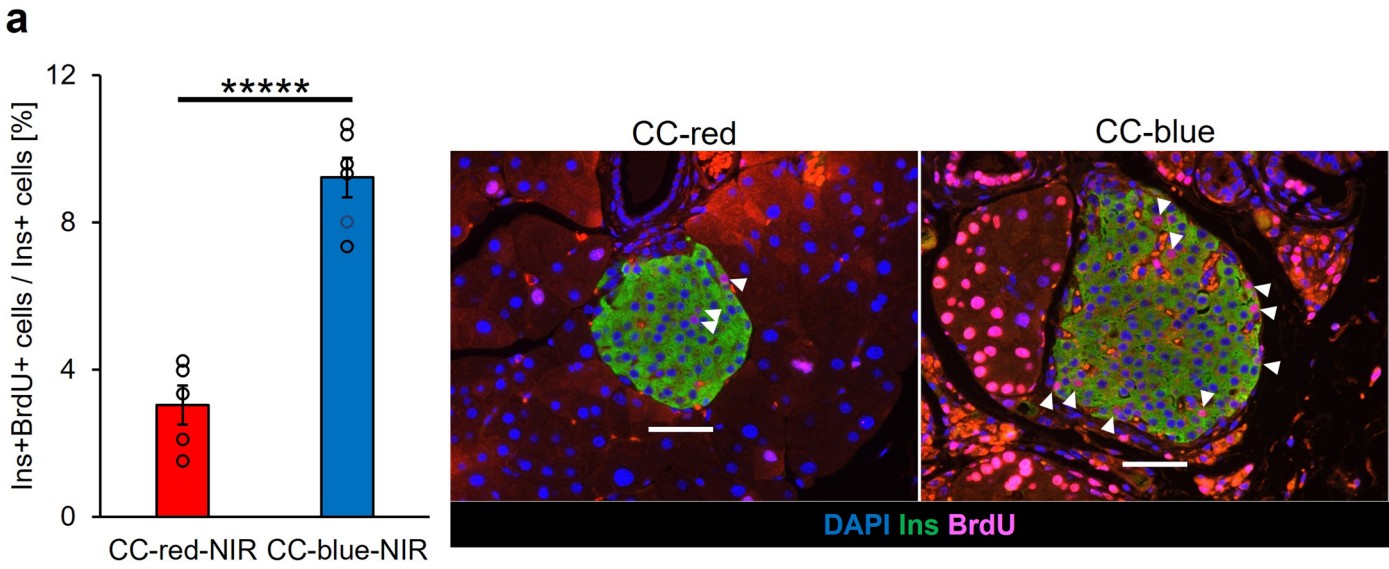

**b**

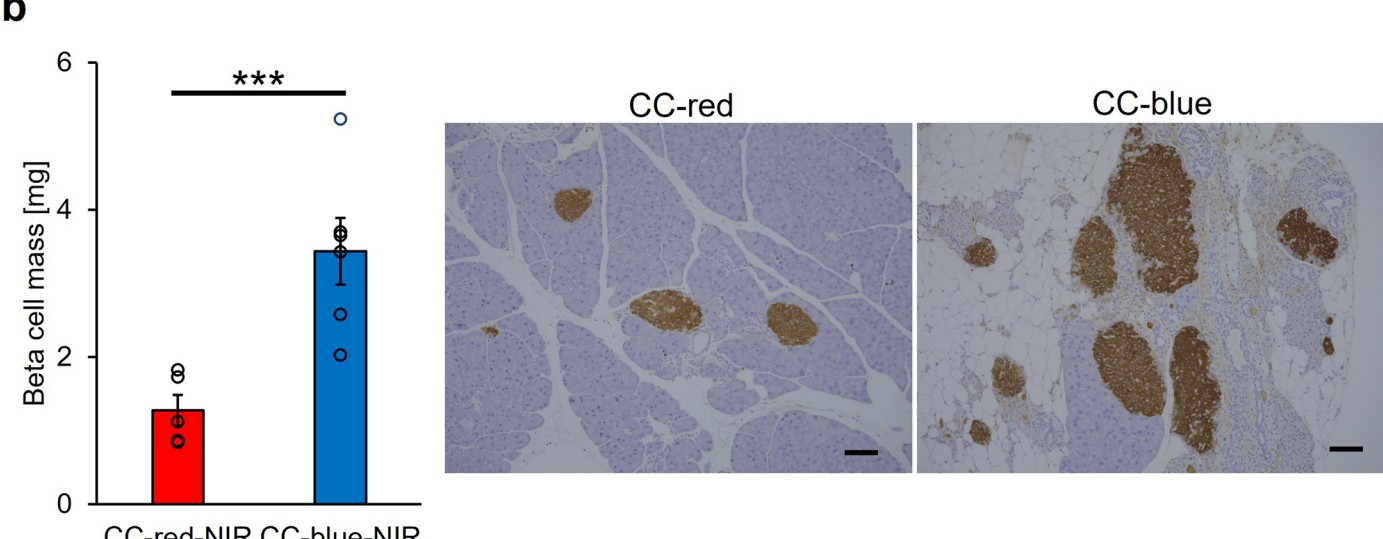

**Extended Data Fig. 8 | beta-cell proliferative effects of NIR-oVNS were maintained for 2 months. a**, The ratio of BrdU positive beta-cells to all beta-cells in the islets of ChAT-ChR2 mice after 2 weeks of chronic NIR-oVNS (two-tailed unpaired t-test (CC-red-NIR vs CC-blue-NIR), *****P = 0.00001875, n = 5 of CC-red-NIR, 6 of CC-blue-NIR); representative images are shown in the right two panels. Each arrowhead denotes a BrdU-positive beta-cell. Scale bars denote 50 μm. **b**, beta-cell masses of ChAT-ChR2 mice after 2 months of chronic NIR-oVNS (two-tailed unpaired t-test (CC-red-NIR vs CC-blue-NIR), ***P = 0.0029, n = 5 of CC-red-NIR, 6 of CC-blue-NIR); representative images are shown in the right two panels. Scale bars denote 100 μm.

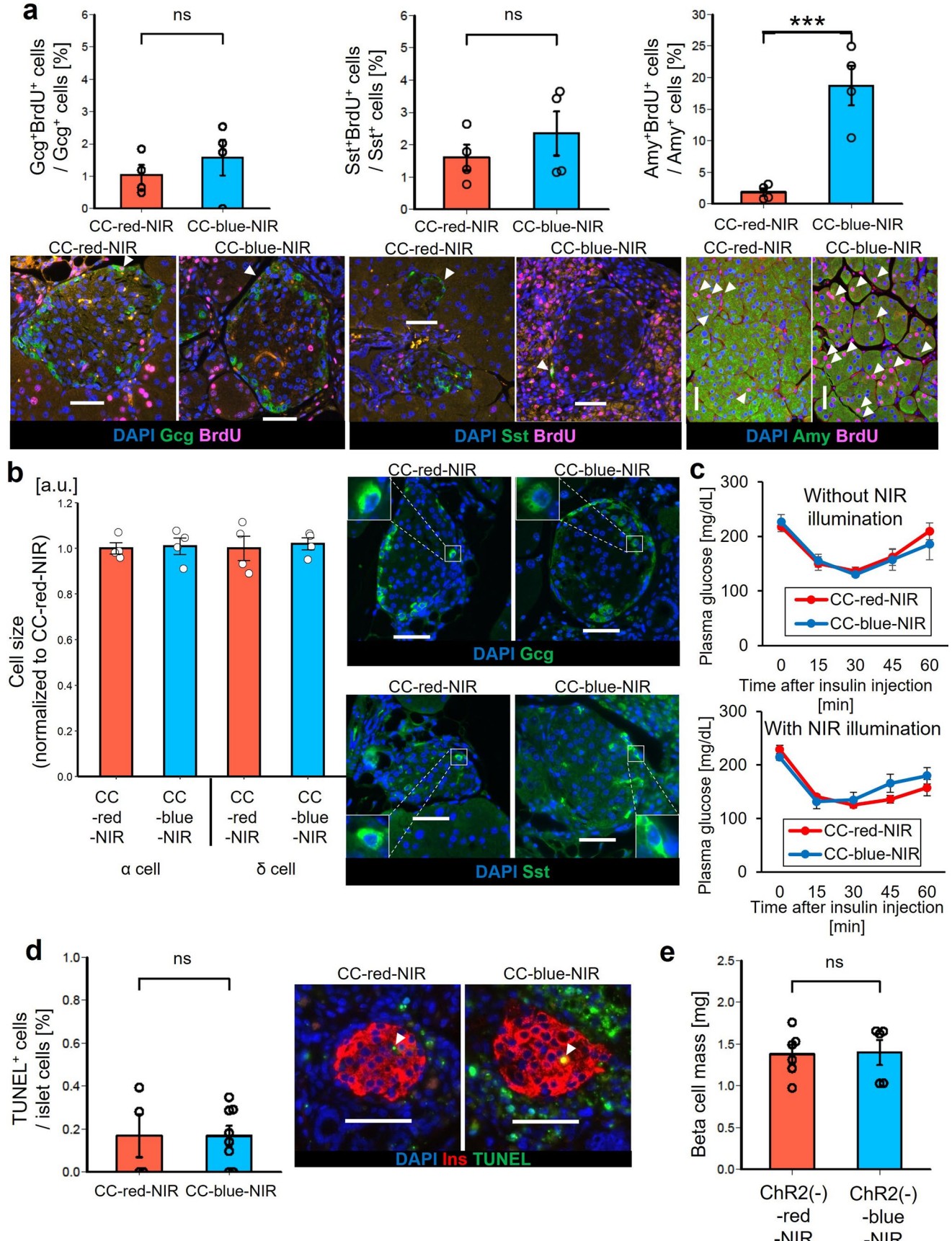

**Extended Data Fig. 9 | See next page for caption.**

**Extended Data Fig. 9 | Effects of chronic NIR-oVNS on non-beta-cell proliferation, non-beta-cell size, insulin sensitivity and beta-cell apoptosis.**
**a**, The ratio of BrdU-positive non-beta-cells to the total number of each cell type in the islets or exocrine tissues of ChAT-ChR2 mice after 2 weeks of chronic NIR-oVNS (a: two-tailed unpaired t-test (CC-red-NIR vs CC-blue-NIR for Amy+ BrdU+ cells), ***P = 0.0018, n = 4); representative images of Gcg stained cells (left, 2 panels), Sst stained cells (middle, 2 panels) and Amy stained cells (right, 2 panels). Each arrowhead denotes a BrdU-positive non-beta-cell. Scale bars denote 50 μm.
**b**, The average size of non-beta-cells (α and δ cells) in ChAT-ChR2 mice after 2 weeks of chronic NIR-oVNS (n = 4); respective representative images of Gcg stained cells (top panels), Sst stained cells (lower panels). White boxes indicate α and δ cells. Cell size was expressed as the ratio of the hormone-positive area to cell number. Scale bars denote 50 μm. **c**, Plasma glucose levels of ChAT-ChR2 mice,

which had undergone 2 weeks of chronic NIR-oVNS, during insulin tolerance tests conducted concomitantly with or without NIR-oVNS (n = 5 of CC-red-NIR without NIR illumination, 5 of CC-blue-NIR without NIR illumination, 5 of CC-red-NIR with NIR illumination, and 4 of CC-blue-NIR with NIR illumination). **d**, The ratio of TUNEL positive beta-cells to all cells in the islets of ChAT-ChR2 mice after 2 weeks of chronic NIR-oVNS (n = 4 of CC-red-NIR, 8 of CC-blue-NIR, two-tailed unpaired t-test, CC-red-NIR vs CC-blue-NIR); representative images are shown in the right two panels. Each arrowhead denotes a TUNEL-positive beta-cell. Scale bars denote 50 μm. **e**, beta-cell masses of non-ChAT-ChR2 (ChAT-Cre(+); LSL-ChR2(-)) mice after 2 weeks of chronic NIR illumination (two-tailed unpaired t-test, CC-red-NIR vs CC-blue-NIR, n = 6 of ChR2(−)-red-NIR, 5 of ChR2(−)-blue-NIR). Data are presented as means ± s.e.m. ns: not significant.

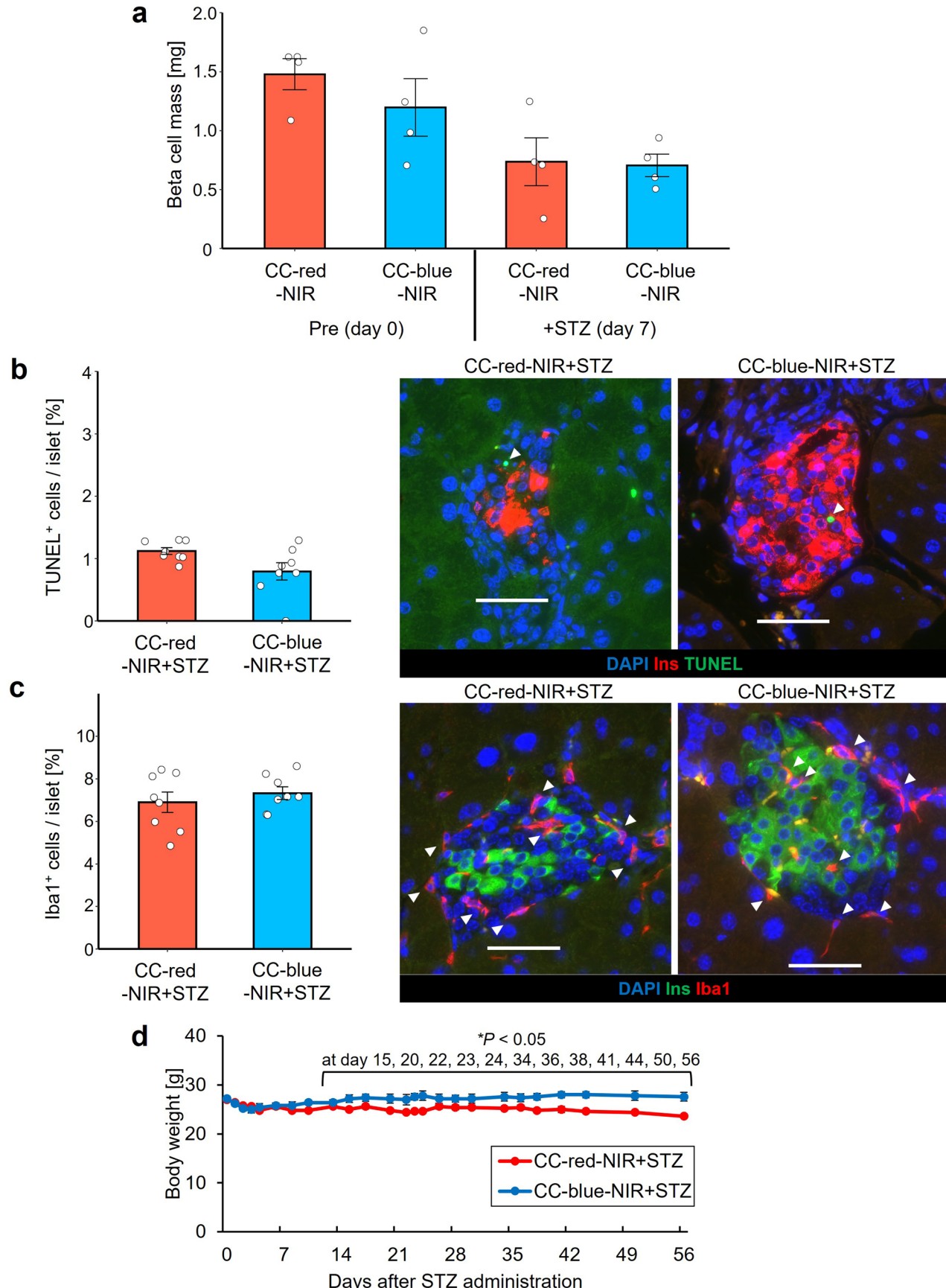

**Extended Data Fig. 10 | See next page for caption.**

**Extended Data Fig. 10 | Effects of NIR-oVNS on STZ-treated ChAT-ChR2 mice.**
**a**, beta-cell masses of ChAT-ChR2 mice before ('Pre (day 0)') and short after
('+STZ (day 7)') starting STZ administration (50 mg/kg i.p., 5 consecutive days)
accompanied by chronic NIR-oVNS (n = 4). **b, c**, The ratios of TUNEL (b) and
Iba-1 (c) positive cells to all cells in the islets of ChAT-ChR2 mice after starting
STZ administration (as above stated) accompanied by 3 weeks of chronic
NIR-oVNS (n = 8). Respective representative images are shown in the right two
panels. Each arrowhead denotes a TUNEL-positive (b) or an Iba-1-positive (c)
beta-cell. **d**, Body weights of ChAT-ChR2 mice after starting STZ administration
(as stated above) accompanied by 2 months of chronic NIR-oVNS (two-way
repeated measures ANOVA, CC-red-NIR+STZ vs CC-blue-NIR+STZ, *$P$ = 0.02219,
0.01415, 0.008994, 0.003630, 0.002314, 0.01415, 0.03460, 0.005711, 0.003630,
0.001480, 0.001480, 0.0004012 at day 15, 20, 22, 23, 24, 34, 36, 38, 41, 44, 50, 56,
respectively, n = 5). Scale bars denote 50 μm. Data are presented as means ± s.e.m.

# Reporting Summary

## Statistics

For all statistical analyses, confirm that the following items are present in the figure legend, table legend, main text, or Methods section.

| n/a | Confirmed | |
|---|---|---|
| ☐ | ☒ | The exact sample size (*n*) for each experimental group/condition, given as a discrete number and unit of measurement |
| ☐ | ☒ | A statement on whether measurements were taken from distinct samples or whether the same sample was measured repeatedly |
| ☐ | ☒ | The statistical test(s) used AND whether they are one- or two-sided<br>*Only common tests should be described solely by name; describe more complex techniques in the Methods section.* |
| ☒ | ☐ | A description of all covariates tested |
| ☐ | ☒ | A description of any assumptions or corrections, such as tests of normality and adjustment for multiple comparisons |
| ☐ | ☒ | A full description of the statistical parameters including central tendency (e.g. means) or other basic estimates (e.g. regression coefficient) AND variation (e.g. standard deviation) or associated estimates of uncertainty (e.g. confidence intervals) |
| ☐ | ☒ | For null hypothesis testing, the test statistic (e.g. *F*, *t*, *r*) with confidence intervals, effect sizes, degrees of freedom and *P* value noted<br>*Give P values as exact values whenever suitable.* |
| ☒ | ☐ | For Bayesian analysis, information on the choice of priors and Markov chain Monte Carlo settings |
| ☒ | ☐ | For hierarchical and complex designs, identification of the appropriate level for tests and full reporting of outcomes |
| ☒ | ☐ | Estimates of effect sizes (e.g. Cohen's *d*, Pearson's *r*), indicating how they were calculated |

*Our web collection on statistics for biologists contains articles on many of the points above.*

## Software and code

Policy information about availability of computer code

| | |
|---|---|
| Data collection | BIOREVO BZ-X710 Viewer (version 1.4.0.1) and BZ-X700 analyzer (version 1.4.1.1) (Keyence, Osaka Japan) for histological analyses. Imaris software's surface function (Bitplane, Belfast, UK) (version 9.6.0) for quantification of contacts between nerve and islet cells. Video motion analysis software Kinovea (version 0.8.15) (https://www.kinovea.org) for measurement of intestinal motility. OMEGAZONE (Omegawave, Tokyo, Japan) Laser Speckle Blood Flow Imager (version 1.03) and Laser Image Analyzer (version 1.07.3) for measurements of organ blood flow. ImageJ (version 1.53c) for measurements of islet cell size. |
| Data analysis | BellCurve for Excel (version 4.02) and R (version 3.5.1) for statistical analysis. |

For manuscripts utilizing custom algorithms or software that are central to the research but not yet described in published literature, software must be made available to editors and reviewers. We strongly encourage code deposition in a community repository (e.g. GitHub). See the Nature Portfolio guidelines for submitting code & software for further information.

## Data

Policy information about availability of data

All manuscripts must include a data availability statement. This statement should provide the following information, where applicable:
- Accession codes, unique identifiers, or web links for publicly available datasets
- A description of any restrictions on data availability
- For clinical datasets or third party data, please ensure that the statement adheres to our policy

The main data supporting the results in this study are available within the paper and its Supplementary Information. Source data are provided with this paper. The raw and analysed datasets generated during the study are available for research purposes from the corresponding author on reasonable request.

# Field-specific reporting

Please select the one below that is the best fit for your research. If you are not sure, read the appropriate sections before making your selection.

☒ Life sciences ☐ Behavioural & social sciences ☐ Ecological, evolutionary & environmental sciences

For a reference copy of the document with all sections, see nature.com/documents/nr-reporting-summary-flat.pdf

# Life sciences study design

All studies must disclose on these points even when the disclosure is negative.

| | |
|---|---|
| Sample size | Sample sizes were based on previous relevant studies (such as those with PMIDs 19023081, 29208957 and 30546054) and pilot experiments. |
| Data exclusions | No data were excluded from the analyses. |
| Replication | More than two independent experiments were performed in every experiment to ensure that the experimental results were reliable. Detailed information can be found in the figure legends. |
| Randomization | The animals were assigned randomly to experimental and control groups. |
| Blinding | Blinding was not required, as the same analysis was adopted for both the experimental and control groups in all experimental conditions, and because the data analyses were based on objectively measurable data. |

# Reporting for specific materials, systems and methods

We require information from authors about some types of materials, experimental systems and methods used in many studies. Here, indicate whether each material, system or method listed is relevant to your study. If you are not sure if a list item applies to your research, read the appropriate section before selecting a response.

## Materials & experimental systems

| n/a | Involved in the study |
|---|---|
| ☐ ☒ | Antibodies |
| ☒ ☐ | Eukaryotic cell lines |
| ☒ ☐ | Palaeontology and archaeology |
| ☐ ☒ | Animals and other organisms |
| ☒ ☐ | Human research participants |
| ☒ ☐ | Clinical data |
| ☒ ☐ | Dual use research of concern |

## Methods

| n/a | Involved in the study |
|---|---|
| ☒ ☐ | ChIP-seq |
| ☒ ☐ | Flow cytometry |
| ☒ ☐ | MRI-based neuroimaging |

## Antibodies

| | |
|---|---|
| Antibodies used | Antibodies to Insulin(I2018, Sigma), glucagon (8233, Cell Signaling Technology, MA, USA), somatostatin (MAB354, Millipore, MA, USA), amylase (A8273, Sigma) and CD31 (550274, BD Bioscience, San Jose, CA, USA), TUNEL (G3250, Promega, Madison, WI, USA), Iba1 (019-19741, Wako, Tokyo, Japan), BrdU (551321, BD Bioscience), Tuj1 (802001, BioLegend, CA, USA), c-Fos (ab222699, abcam), Alexa Fluor 488 goat anti-mouse IgG (ab150117, Abcam, MA, USA), Alexa Fluor 488 goat anti-rabbit IgG (4412, Cell Signaling Technology), Alexa Fluor 488 goat anti-rat IgG (A11006, Molecular Probes, OR, USA), Alexa Fluor 488 goat anti-rabbit IgG (A11008, Molecular Probes), Alexa Fluor 488 donkey anti-mouse IgG (715-545-151, Jackson ImmunoResearch, PA, USA) or Alexa Fluor 594 donkey anti-mouse IgG . (715-585-151, Jackson ImmunoResearch), Alexa Fluor 594 donkey anti-rabbit IgG (A21207, Life Technologies, CA, USA), streptavidin Alexa Fluor 594 (S32356, Invitrogen, MA, USA) and Alexa Fluor 546 goat anti-rabbit IgG (A11010, Invitrogen, MA, USA). |
| Validation | The antibodies were validated by the manufacturers.<br><br>Insulin (I2018, Sigma) https://www.sigmaaldrich.com/JP/en/product/sigma/i2018<br>glucagon (8233, Cell Signaling Technology, MA, USA) https://www.cellsignal.com/products/primary-antibodies/proglucagon-d16g10-xp-rabbit-mab/8233<br>somatostatin (MAB354, Millipore, MA, USA) https://www.merckmillipore.com/JP/en/product/Anti-Somatostatin-Antibody-clone-YC7,MM_NF-MAB354<br>amylase (A8273, Sigma) https://www.sigmaaldrich.com/JP/en/product/sigma/a8273<br>CD31 (550274, BD Bioscience, San Jose, CA, USA) https://www.bdbiosciences.com/en-us/products/reagents/flow-cytometry-reagents/research-reagents/single-color-antibodies-ruo/purified-rat-anti-mouse-cd31.550274<br>TUNEL (G3250, Promega, Madison, WI, USA) https://www.promega.jp/en/products/cell-health-assays/apoptosis-assays/deadend-fluorometric-tunel-system/?catNum=G3250&cs=y<br>Iba1 (019-19741, Wako, Tokyo, Japan) https://labchem-wako.fujifilm.com/us/product/detail/W01W0101-1974.html |

BrdU (51-75512 L, BD Bioscience)
Tuj1 (802001, BioLegend, CA, USA) https://www.biolegend.com/ja-jp/products/purified-anti-tubulin-beta-3-tubb3-antibody-11579?GroupID=GROUP686
c-Fos (ab222699, abcam) https://www.abcam.com/products/primary-antibodies/c-fos-antibody-epr21930-238-ab222699.html
Alexa Fluor 488 goat anti-mouse IgG (ab150117, Abcam, MA, USA) https://www.abcam.com/products/secondary-antibodies/goat-mouse-igg-hl-alexa-fluor-488-preadsorbed-ab150117.html
Alexa Fluor 488 goat anti-rabbit IgG (4412, Cell Signaling Technology) https://www.cellsignal.jp/products/secondary-antibodies/anti-rabbit-igg-h-l-f-ab-2-fragment-alexa-fluor-488-conjugate/4412
Alexa Fluor 488 goat anti-rat IgG (A11006, Molecular Probes, OR, USA) https://www.thermofisher.com/antibody/product/Goat-anti-Rat-IgG-H-L-Cross-Adsorbed-Secondary-Antibody-Polyclonal/A-11006
Alexa Fluor 488 goat anti-rabbit IgG (A11008, Molecular Probes) https://www.thermofisher.com/antibody/product/Goat-anti-Rabbit-IgG-H-L-Cross-Adsorbed-Secondary-Antibody-Polyclonal/A-11008
Alexa Fluor 488 donkey anti-mouse IgG (715-545-151, Jackson ImmunoResearch, PA, USA) https://www.jacksonimmuno.com/catalog/products/715-545-151
Alexa Fluor 594 donkey anti-mouse IgG (715-585-151, Jackson ImmunoResearch) https://www.jacksonimmuno.com/catalog/products/715-585-151
Alexa Fluor 594 donkey anti-rabbit IgG (A21207, Life Technologies, CA, USA) https://www.thermofisher.com/antibody/product/Donkey-anti-Rabbit-IgG-H-L-Highly-Cross-Adsorbed-Secondary-Antibody-Polyclonal/A-21207
streptavidin Alexa Fluor 594 (S32356, Invitrogen, MA, USA) https://www.thermofisher.com/order/catalog/product/jp/en/S32356
Alexa Fluor 546 goat anti-rabbit IgG (A11010, Invitrogen, MA, USA) https://www.thermofisher.com/antibody/product/Goat-anti-Rabbit-IgG-H-L-Cross-Adsorbed-Secondary-Antibody-Polyclonal/A-11010
Insulin (I2018, Sigma) https://www.sigmaaldrich.com/JP/en/product/sigma/i2018
glucagon (8233, Cell Signaling Technology, MA, USA) https://www.cellsignal.com/products/primary-antibodies/proglucagon-d16g10-xp-rabbit-mab/8233
somatostatin (MAB354, Millipore, MA, USA) https://www.merckmillipore.com/JP/en/product/Anti-Somatostatin-Antibody-clone-YC7,MM_NF-MAB354
amylase (A8273, Sigma) https://www.sigmaaldrich.com/JP/en/product/sigma/a8273
CD31 (550274, BD Bioscience, San Jose, CA, USA) https://www.bdbiosciences.com/en-us/products/reagents/flow-cytometry-reagents/research-reagents/single-color-antibodies-ruo/purified-rat-anti-mouse-cd31.550274
TUNEL (G3250, Promega, Madison, WI, USA) https://www.promega.jp/en/products/cell-health-assays/apoptosis-assays/deadend-fluorometric-tunel-system/?catNum=G3250&cs=y
Iba1 (019-19741, Wako, Tokyo, Japan) https://labchem-wako.fujifilm.com/us/product/detail/W01W0101-1974.html
BrdU (551321, BD Bioscience)  Validation reports are deleted from the manufacture's website. However, we obtained the validation report previously and  relied on information in the report.
Tuj1 (802001, BioLegend, CA, USA) https://www.biolegend.com/ja-jp/products/purified-anti-tubulin-beta-3-tubb3-antibody-11579?GroupID=GROUP686
c-Fos (ab222699, abcam) https://www.abcam.com/products/primary-antibodies/c-fos-antibody-epr21930-238-ab222699.html
Alexa Fluor 488 goat anti-mouse IgG (ab150117, Abcam, MA, USA) https://www.abcam.com/products/secondary-antibodies/goat-mouse-igg-hl-alexa-fluor-488-preadsorbed-ab150117.html
Alexa Fluor 488 goat anti-rabbit IgG (4412, Cell Signaling Technology) https://www.cellsignal.jp/products/secondary-antibodies/anti-rabbit-igg-h-l-f-ab-2-fragment-alexa-fluor-488-conjugate/4412
Alexa Fluor 488 goat anti-rat IgG (A11006, Molecular Probes, OR, USA) https://www.thermofisher.com/antibody/product/Goat-anti-Rat-IgG-H-L-Cross-Adsorbed-Secondary-Antibody-Polyclonal/A-11006
Alexa Fluor 488 goat anti-rabbit IgG (A11008, Molecular Probes) https://www.thermofisher.com/antibody/product/Goat-anti-Rabbit-IgG-H-L-Cross-Adsorbed-Secondary-Antibody-Polyclonal/A-11008
Alexa Fluor 488 donkey anti-mouse IgG (715-545-151, Jackson ImmunoResearch, PA, USA) https://www.jacksonimmuno.com/catalog/products/715-545-151
Alexa Fluor 594 donkey anti-mouse IgG (715-585-151, Jackson ImmunoResearch) https://www.jacksonimmuno.com/catalog/products/715-585-151
Alexa Fluor 594 donkey anti-rabbit IgG (A21207, Life Technologies, CA, USA) https://www.thermofisher.com/antibody/product/Donkey-anti-Rabbit-IgG-H-L-Highly-Cross-Adsorbed-Secondary-Antibody-Polyclonal/A-21207
streptavidin Alexa Fluor 594 (S32356, Invitrogen, MA, USA) https://www.thermofisher.com/order/catalog/product/jp/en/S32356
Alexa Fluor 546 goat anti-rabbit IgG (A11010, Invitrogen, MA, USA) https://www.thermofisher.com/antibody/product/Goat-anti-Rabbit-IgG-H-L-Cross-Adsorbed-Secondary-Antibody-Polyclonal/A-11010

# Animals and other organisms

Policy information about studies involving animals; ARRIVE guidelines recommended for reporting animal research

| Laboratory animals | ChAT-ChR2 mice (male, 10-20 weeks of age) were generated by crossing heterozygous ChAT-IRES-Cre mice (male and female, 10–20 weeks of age, Jax stock number 018957) and homozygous LSL-ChR2(H134R)-EYFP mice (male and female, 10–20 weeks of age, Jax stock number 024109). |
|---|---|
| Wild animals | The study did not involve wild animals. |
| Field-collected samples | The study did not involve samples collected from the field. |
| Ethics oversight | All experiments in this study were conducted in accordance with the Tohoku University institutional guidelines. Ethics approval was obtained from the Institutional Animal Care and Use Committee of the Tohoku University Environmental & Safety Committee. |

Note that full information on the approval of the study protocol must also be provided in the manuscript.

