## [Peer Review File · Nature Biomedical Engineering]

Optogenetic stimulation of vagal nerves for enhanced glucose-stimulated insulin secretion and beta-cell proliferation

Corresponding author: Junta Imai

Editorial note

This document includes relevant written communications between the manuscript's corresponding author and the editor and reviewers of the manuscript during peer review. It includes decision letters relaying any editorial points and peer-review reports, and the authors' replies to these (under 'Rebuttal' headings). The editorial decisions are signed by the manuscript's handling editor, yet the editorial team and ultimately the journal's Chief Editor share responsibility for all decisions.

Any relevant documents attached to the decision letters are referred to as **Appendix #**, and can be found appended to this document. Any information deemed confidential has been redacted or removed. Earlier versions of the manuscript are not published, yet the originally submitted version may be available as a preprint. Because of editorial edits and changes during peer review, the published title of the paper and the title mentioned in below correspondence may differ.

This manuscript was originally peer-reviewed at *Nature* before being transferred to *Nature Biomedical Engineering*.

Correspondence

Fri 04 Mar 2022

Decision on Article nBME-22-0044-T

Dear Prof Imai,

Thank you again for submitting to *Nature Biomedical Engineering* your manuscript, "Optogenetic vagal nerve stimulation enhances glucose-stimulated insulin secretion and promotes pancreatic β cell proliferation". The manuscript has been seen by 3 experts, whose reports you will find at the end of this message. One other reviewer is yet to provide their feedback (at this point it is unlikely that they will, but we will send you their report if we receive it).

You will see that although the reviewers have some good words for the work, they articulate concerns about the degree of support for the claims and the translatability of the system, and in this regard provide useful suggestions for improvement. We hope that with significant further work you can address the criticisms and convince the reviewers of the merits of the study. Given the translational relevance of the manuscript, eventual publication of the work would hinge on the safety of the lanthanide microparticles, the long-term performance of the system and the degree to which higher insulin levels impact overall blood-glucose control, and therefore the robust validation of safety and blood glucose control raised by both reviewers being appropriately addressed (with compelling arguments and/or additional evidence). In particular, we would expect that a revised version of the manuscript provides:

* Inclusion of appropriate controls for mouse experiments.

* Clarification of the reason for the use of blue light-emitting lanthanide microparticles (LMPs) in the pancreatic ducts (with compelling arguments and/or additional evidence).- * Demonstration of blood-glucose control.
- * Demonstration of long-term performance.
- * Demonstration of the long-term safety of the system.

When you are ready to resubmit your manuscript, please upload the revised files, a point-by-point rebuttal to the comments from all reviewers, the reporting summary, and a cover letter that explains the main improvements included in the revision and responds to any points highlighted in this decision.

Please follow the following recommendations:

- * Clearly highlight any amendments to the text and figures to help the reviewers and editors find and understand the changes (yet keep in mind that excessive marking can hinder readability).
- * If you and your co-authors disagree with a criticism, provide the arguments to the reviewer (optionally, indicate the relevant points in the cover letter).
- * If a criticism or suggestion is not addressed, please indicate so in the rebuttal to the reviewer comments and explain the reason(s).
- * Consider including responses to any criticisms raised by more than one reviewer at the beginning of the rebuttal, in a section addressed to all reviewers.
- * The rebuttal should include the reviewer comments in point-by-point format (please note that we provide all reviewers will the reports as they appear at the end of this message).
- * Provide the rebuttal to the reviewer comments and the cover letter as separate files.

We hope that you will be able to resubmit the manuscript within 25 weeks from the receipt of this message. If this is the case, you will be protected against potential scooping. Otherwise, we will be happy to consider a revised manuscript as long as the significance of the work is not compromised by work published elsewhere or accepted for publication at *Nature Biomedical Engineering*.

We hope that you will find the referee reports helpful when revising the work. Please do not hesitate to contact me should you have any questions.

Best wishes,

Michelle

Dr Michelle Korda
Senior Editor, Nature Biomedical Engineering

Reviewer #1 (Report for the authors (Required)):

In their revision to the manuscript, Kawana and collaborators satisfactorily addressed the concerns raised by this reviewer. They provide extensive new relevant data and have included important control experiments. As a result, the results support the main conclusions drawn by the authors, namely that vagal innervation stimulates insulin secretion and promotes beta cell proliferation. It is still a mystery why these effects are so specific for beta cells given that cholinergic agonists stimulate many other tissue compartments within the pancreas. Nevertheless, this work can be considered seminal for the field of autonomic control of pancreas endocrine function and glucose homeostasis. It is further expected that the technologies developed here will be widely adopted by other investigators, including yours truly.

Reviewer #2 (Report for the authors (Required)):

In this work, the authors deploy an optogenetic tool (ChR2) to activate vagal nerves which in turn innervate the pancreas. Briefly, first of all, they developed the optical fiber-implanting surgical procedure to stimulate vagal nerve activation and found that the optogenetic stimulation leads to functional modulation of beta cells. To avoid fiber implantation and achieve more selective vagal nerve activation, the authors used near-infrared (NIR) light to activate pancreas by injection of blue light-emitting lanthanide microparticles (LMPs) in the pancreatic ducts, and they found that selective activation of vagal nerves innervating the pancreases is sufficient to enhance glucose-stimulated insulin secretion and marked beta cell proliferation. While the study is interesting and further showed the advantages of optogenetic tools enable to selectively activate vagal nerves of specific organs, the experimental design and the present data in the manuscript can not support their claims. The lack of substantial long-term performance of the system to demonstrate biological functions and the need of injection of blue light-emitting lanthanide microparticles in the target organ hindered the translatability of their method. It seems this work is too preliminary and unlikely to provide the high technical quality and translational relevance for biomedical applications. However, I have the following the comments that might be helpful for the authors to further improve the quality of this manuscript.

1: The authors should further improve the writing. For example, in the introduction part of this manuscript, the author should simply explain or introduce which optogenetic tool they will use for the following studies. They should explain the reason and the working mode to use ChR2 for insulin secretion and β cell proliferation. They should explain why they use the mutation version of ChR2 (H134R)? All the information needs to be explained in the appropriated part. ChR2 is activated by blue light. Why they don't use the red light activated opsin for in vivo study?

2: The author should provide a graphic abstract as figure 1 or figure 1a, showing the general design principle or working mechanism of this study to help readers easily understand their story. Moreover, in the results, the authors mentioned a lot of controls, but these controls are different. They should clearly explain what are these controls.

3 : The whole manuscript lack of appropriated controls in the mouse experiments. There is no dark control group. From Figure 3-5, wildtype control mice should be included. Dark control mice should also be included.

4: The huge concern is the injection of blue light-emitting lanthanide microparticles (LMPs) in the pancreatic ducts. This will cause toxicity and cause inflammation and immune response. Moreover, whether the light intensity is consistent throughout pancreas? What is the light intensity in the organ? 35 W of NIR illumination is very strong and can cause heat. There is red light responsive opsin which has been identified. If the authors use red light responsive light sensors instead of ChR2, they do not need to inject the LMPs to make the system so complicate.

5: The author should explain why plasma insulin levels after glucose loading were significantly higher in LED-oVNS mice compared with control mice, while blood glucose levels were similar between the two groups in Figure 3f and g.

7: Longer-term performance (functionality and safety) of their system should be presented in T1D mice.

8: In figure 6, some of the data obtained in one week (eg. Brdu) and some in three weeks (blood glucose). Actually, they should show the BrdU data after three weeks of illumination.

9: The time schedule of STZ injection, LMPs injection and light illumination was not specified in the text. Please clarify the specific timeline in Figures.

10: The authors only present the light frequency but they did not present the light intensity which should be present in the results.

Reviewer #3 (Report for the authors (Required)):

In their manuscript, Kawana et al., are combining two standard optogenetic strategies such as optical fibers and upconversion to investigate the role of vagal nerves in glucose-stimulated insulin secretion and induction of beta-cell proliferation. Such optogenetic stimulation provided stable, prolonged and selective in vivo activation of the vagal nerves in awake animals. Indeed, light-induced vagal nerve activation increased beta-cell mass and eventually attenuated hyperglycemia in diabetic mice. Altogether, this work shows that optogenetic remote stimulation of vagal nerves triggers beta cell proliferation and could as such be useful for diabetes treatment. We have several suggestions and comments for the authors to drive their work to a higher level and expose some critical points question whether the strategy will work as shown.

Major points:

1. Data shown in Figures 3 f and g are not consistent. Although Fig. 3f shows higher insulin levels in response to LED-oVNS stimulation, the resulting blood-glucose levels shown in Fig. 3f remain unchanged compared to controls. Since blood-glucose control is the ultimate therapeutic goal of any diabetes therapy, the concept kind of misses the major therapeutic point. It is imperative to see a glycemic response as a consequence of increased insulin and it will be also important to show that these insulin levels will not lead to any hypoglycemia.
2. It seems like an important negative control is missing in Figure 4C. The authors only show NIR-induced vagal nerve activation in ChAT-ChR2. Essential control would show animals without ChAT-ChR2 and ChAT-ChR2 animals illuminated with red light (660 nm) instead of NIR.
3. Although this standard optogenetic stimulation strategy resulted in the activation of the vagal nerves, similar to previous optogenetic reports on stimulation of nerve cells, the authors must discuss and show by additional experimentation beyond doubt the absence of long-term cytotoxicity, the long-term efficiency of upconversion and the impact of high energy requirement on every-day and real-world therapies.

Minor points:

1. There are several conceptionally and technically similar publications out there and the authors should carefully compare their work with this previous work.
2. bar graphs in the main figures are not aligned (e.g., Fig 4) and are using different style and fonts (e.g., Fig3h and i).
3. Figure 1; The immunostainings of beta-cells and vagal nerve are very poor and should be replaced with images of much higher quality.

Wed 21 Sep 2022

Decision on Article NBME-22-0044A

Dear Prof Imai,

Thank you for your revised manuscript, "Optogenetic vagal nerve stimulation enhances glucose-stimulated insulin secretion and promotes pancreatic β cell proliferation", which has been seen by Reviewer #2 and new Reviewer #4. In their reports, which you will find at the end of this message, you will see that the reviewers acknowledge the improvements to the work and raise a few additional technical criticisms that we hope you will be able to address. Given the translational relevance of the manuscript, eventual publication of the work would hinge on the degree to which the vagal stimulation for pancreatic beta cell proliferation as a therapy for insulin-deficient diabetes constitutes an advance, and therefore the robust validation of the control of blood-glucose levels being appropriately addressed.

In particular, we would expect that a revised version of the manuscript provides:

- * Demonstration of the insulin tolerance test during acute and chronic LED and LMP stimulation.
- * Demonstration that the intensity of NIR light used is sufficient to activate cholinergic neurons.
- * Analysis of insulin production in the NIR treated-STZ studies.
- * Analysis of control mice injected with vehicle alone.
- * Assessment of statistical significance using 2-way ANOVA.

As before, when you are ready to resubmit your manuscript, please upload the revised files, a point-by-point rebuttal to the comments from all reviewers, the reporting summary, and a cover letter that explains the main improvements included in the revision and responds to any points highlighted in this decision.

As a reminder, please follow the following recommendations:

- * Clearly highlight any amendments to the text and figures to help the reviewers and editors find and understand the changes (yet keep in mind that excessive marking can hinder readability).
- * If you and your co-authors disagree with a criticism, provide the arguments to the reviewer (optionally, indicate the relevant points in the cover letter).
- * If a criticism or suggestion is not addressed, please indicate so in the rebuttal to the reviewer comments and explain the reason(s).
- * Consider including responses to any criticisms raised by more than one reviewer at the beginning of the rebuttal, in a section addressed to all reviewers.
- * The rebuttal should include the reviewer comments in point-by-point format (please note that we provide all reviewers will the reports as they appear at the end of this message).
- * Provide the rebuttal to the reviewer comments and the cover letter as separate files.

We hope that you will be able to resubmit the manuscript within 12 weeks from the receipt of this message. If this is the case, you will be protected against potential scooping. Otherwise, we will be happy to consider a revised manuscript as long as the significance of the work is not compromised by work published elsewhere or accepted for publication at *Nature Biomedical Engineering*.

We look forward to receive a further revised version of the work. Please do not hesitate to contact me should you have any questions.

Best wishes,

Michelle

Dr Michelle Korda
Senior Editor, Nature Biomedical Engineering

Reviewer #2 (Report for the authors (Required)):

The authors have addressed most of my concerns.

But, I have one more question. In the schematic diagram of Fig. 6a, why did the authors start the lighting before creating the diabetes model? This doesn't make sense. The light treating should be applied in a created diabetic mouse model.

Reviewer #4 (Report for the authors (Required)):

This manuscript examines the role of optogenetic stimulation of cholinergic innervation in modulating glucose-stimulated insulin release and beta cell proliferation in vivo in mice. Chronic optogenetic activation of subdiaphragmatic vagal fibers using an implanted optical fiber increased beta cell replication and mass. Optogenetic activation of pancreatic cholinergic fibers using blue-light emitting lanthanide micro-particles improved glucose tolerance and increased beta cell mass in WT and STZ-treated mice.

These are interesting studies with a technological advance in the use of LMP to modulate peripheral innervation in an organ-specific manner. This approach could be applied to investigate the role of neural pathways in other organs so has implications for several fields. The manuscript may also advance our understanding of neural regulation of beta cell proliferation by demonstrating activation of targeted pancreatic cholinergic innervation increases beta cell mass which could have translational applications in type 1 and type 2 diabetes. However, there are several limitations to the current manuscript.

Major

1. It would be helpful to know which of the organs adjacent to the pancreas also express ChR2 in cholinergic fibers and may be modulated by chronic light delivery to the abdomen. Cholinergic innervation of the stomach, duodenum, jejunum, and ileum have been reported. Reports regarding cholinergic innervation of the liver and portal system are conflicting and examining ChR2 expression may help clarify this field.
2. The authors show optogenetic vagal stimulation increases plasma insulin without an effect on blood glucose during ipGTT. These results are in keeping with decreased insulin sensitivity and perhaps not enhanced insulin secretion. There are no significant differences in glucagon between control and treated groups but these studies have a relatively small n and only examine a single time point. The authors performed an ITT but according to the methods, this was not conducted with nerve stimulation so does not test whether vagal nerve stimulation alters insulin sensitivity. It would be helpful to perform an ITT during vagal stimulation to determine insulin sensitivity and perhaps examine glucagon release at additional time points.
3. Similarly, it would be helpful to perform insulin tolerance testing with chronic NIR stimulation to ensure the increase in beta cell mass is not secondary to insulin resistance during stimulation.
4. Please could the authors clarify the intensity of NIR light used during measurement of electrophysiological responses of vagal nerves (Fig 4c)? The NIR intensity reported in the abdominal cavity (0.5mW/mm²) appears to be significantly less than used in published CNS studies so confirming this is sufficient to activate cholinergic neurons is important, eg. by demonstrating increased fos expression in intrapancreatic ganglia. If the NIR light is producing recordable responses in branches of the vagus, this suggests that activation is not limited to intrapancreatic terminals and maybe back-propagating, potentially with effects on other organs.

5. The authors do not show increased insulin in the NIR treated-STZ studies which would be helpful to demonstrate the increase in beta cell mass is functional.

6. The authors should assess significance using 2-way ANOVA for their longitudinal studies examining insulin, glucose and body weight rather than multiple t-tests or 1-way ANOVA (Fig 3 f, g, Fig 4 e – h, Fig 6 b, Extended data: fig 3a, Fig 4a, Fig 6b-e, Fig 7, Fig 11 c, fig 12 e).

7. The volume of LMP (500ul) injected into the pancreatic duct for LMP delivery is quite large compared to published studies and both groups of LMP treated mice fail to gain weight over two months of the study. While blue LMP injected mice may not differ from red LMP injected mice, there may be toxicity from LMP per se and comparison against mice injected with vehicle alone would be helpful to determine this.

Minor

1. ChR2 expression is shown in cells adjacent to islets but there is no neural marker used to confirm expression in intrapancreatic ganglia rather than off-target expression in acinar cells.

2. Some experimental details are missing that are needed to interpret the data. When were the electrophysiological recordings from the vagus made in relation to the chronic stimulation with LED implants? It would be helpful to show that light-evoked activation is present at the end of the 2 week stimulation schedule.

3. It would be helpful for the authors to indicate in the text that the LMP approach likely modulates activity at both the preganglionic cholinergic terminals onto the intrapancreatic ganglia and the post-ganglionic cholinergic neurons in the intrapancreatic ganglia.

4. Please could the authors indicate when the vagotomy was performed in relation to the GTT studies performed with LMP activation. Does the loss of insulin response with vagotomy indicate the site of the dominant effect of LMP activation (preganglionic cholinergic terminals vs. postganglionic neurons)?

5. Could the authors also clarify the NIR intensity used to demonstrate blue light emission at the surface of the pancreas (fig 4b)?

Fri 06 Jan 2023

Decision on Article NBME-22-0044A

Dear Prof Imai,

Thank you for your revised manuscript, "Optogenetic vagal nerve stimulation enhances glucose-stimulated insulin secretion and promotes pancreatic β cell proliferation". Having consulted with Reviewers #6 and #8 (whose comments you will find at the end of this message), I am pleased to write that we shall be happy to publish the manuscript in *Nature Biomedical Engineering*, provided that the points specified below by the Reviewers are addressed.

We will be performing detailed checks on your manuscript, and in due course will send you a checklist detailing our editorial and formatting requirements, as well as a request for the aforementioned minor points to be addressed. You will need to follow these instructions before you upload the final manuscript files.

Best wishes,

Michelle

Dr Michelle Korda
Senior Editor, Nature Biomedical Engineering

Reviewer #2 (Report for the authors (Required)):

They have addressed my concerns.

One more suggestion, to better show the effectiveness of NIR-oVNS-mediated blood control compared with healthy mice, it is better to include wild-type control mice in Figure 6. b and c.

Reviewer #4 (Report for the authors (Required)):

The authors have added a significant amount of new data and addressed most of my concerns.

Specifically, they have confirmed that there are no ChR2-expressing fibers in the liver and portal system in their model, that there are no significant differences in glucagon secretion or insulin sensitivity with acute optogenetic vagal stimulation or in insulin sensitivity with chronic NIR stimulation, confirmed activation of parasympathetic ganglionic cells with NIR stimulation, and demonstrated increased insulin in NIR-treated STZ-treated mice. In addition, they have reanalyzed their results using 2-way ANOVA.

The manuscript describes elegant technological advances in the use of LMP for targeted modulation of pancreatic innervation and provides important new information about the contribution of pancreatic parasympathetic innervation to the regulation of beta cell replication and mass.

I have one minor comment.

-On page 4, lines 37-38, the authors comment that enhanced gastrointestinal motility might minimize suppression of post-load blood glucose despite increased insulin. According to their methods, the glucose load was delivered by IP injection rather than orally and so the glucose load would bypass the GI tract and not be affected by changes in GI motility.

Perhaps the authors could comment on why the significant increase in plasma insulin during optogenetic stimulation does not alter blood glucose during ipGTT when insulin sensitivity is unchanged and effects on GI motility are unlikely to be contributing.

Rebuttal 1

Responses to reviewer 2

1: The authors should further improve the writing. For example, in the introduction part of this manuscript, the author should simply explain or introduce which optogenetic tool they will use for the following studies. They should explain the reason and the working mode to use ChR2 for insulin secretion and β cell proliferation. They should explain why they use the mutation version of ChR2 (H134R)? All the information needs to be explained in the appropriated part. ChR2 is activated by blue light. Why they don't use the red light activated opsin for in vivo study?

As suggested, we included explanations and reasons for employing the optogenetic tools which we used in the present study in the Introduction section (page 2, lines 23 to 32).

We used Rosa-CAG-LSL-ChR2 (H134R)-EYFP mice, because Rosa-CAG-LSL-ChR2 (H134R)-EYFP mice are widely used in optogenetic research (van der Zouwen et al, 2021, *Front Neural Circuits*), and substantial expressions of ChR2 in cholinergic neurons, achieved by crossing with ChAT-IRES-Cre mice, were previously reported (Hedrick et al, 2016, *PLoS ONE*). Therefore, to achieve stable vagal nerve activation *in vivo*, we selected this system to generate ChAT-ChR2 mice. We added these descriptions to the Results section (page 2, lines 39 to 41) with citing new references as #22 and 24.

Indeed, strong ChR2 expressions were successfully observed in vagal nerves, including those in the pancreas in ChAT-ChR2 mice, and employing these mice allowed us to examine the effects of vagal nerve activation on β cells.

Reasons for not using red light-responsive opsin in the present study are given below in the response to comment #4.

2: The author should provide a graphic abstract as figure 1 or figure 1a, showing the general design principle or working mechanism of this study to help readers easily understand their story. Moreover, in the results, the authors mentioned a lot of controls, but these controls are different. They should clearly explain what are these controls.

As suggested, to improve readability, we included a graphic abstract which shows the experimental design principle of the present study. The graphic abstract is presented in Figure 1a of the revised manuscript.

In addition, we explained the controls used in each experiment in appropriate parts of the Results section (page 3, lines 39 to 41, page 4, lines 12 to 13, page 4, lines 39 to 41, page 5, lines 4 to 5, page 5 and lines 28 to 29). To further improve readability, we have now named all control mice in the LED-oVNS and NIR-oVNS experiments, such as CC-ctrl-mice, WT-LED-mice, CC-red-NIR-mice, WT-blue-NIR-mice and CC-blue-dark-mice.

For optimal clarity, we have changed the term “control” to these specific control names throughout the manuscript.

3 : The whole manuscript lack of appropriated controls in the mouse experiments. There is no dark control group. From Figure 3-5, wildtype control mice should be included. Dark control mice should also be included.

As suggested, we performed LED-oVNS experiments with additional controls in which optical fibers were implanted in wild-type mice followed by loading of LED blue light stimuli (WT-LED-mice). We evaluated BrdU-positive β -cells and β -cell mass after 2 weeks of stimulation, and found that β -cell proliferation and mass were both significantly increased in CC-LED-mice alone as compared with either of the controls, i.e., CC-ctrl-mice and WT-LED-mice. We replaced Figures 3h and 3i and provided appropriate description in the Results section of the revised manuscript (page 4, lines 11 to 14).

Next, we performed NIR-oVNS experiments with two additional types of controls; 1) wild-type mice in which blue light-emitting lanthanide micro-particles (LMPs) had been placed in the pancreatic ducts, followed by NIR light illumination (WT-blue-NIR-mice) and 2) ChAT-ChR2 mice in which blue light-emitting LMPs had been placed in the pancreatic ducts, followed by being maintained in the dark (CC-blue-dark-mice). We performed glucose tolerance tests, and found that blood insulin levels after glucose loading were markedly elevated in ChAT-ChR2 mice in which blue light-emitting LMPs had been placed in the pancreatic ducts, followed by NIR light illumination (CC-blue-NIR-mice) as compared with ChAT-ChR2 mice in which red light-emitting LMPs had been placed in the pancreatic ducts, followed by NIR light illumination (CC-red-NIR-mice), WT-blue-NIR- and CC-blue-dark-mice. In addition, blood glucose levels after glucose loading were lower in CC-blue-NIR-mice than those in the three types of controls, i.e., WT-blue-NIR-, CC-red-NIR- and CC-blue-dark-mice. Then, we estimated BrdU-positive β -cells and β -cell mass after 2 weeks of stimulation, and found that β -cell proliferation and masses were both significantly increased in CC-blue-NIR-mice as compared to all types of controls, i.e., WT-blue-NIR-, CC-red-NIR- and CC-blue-dark-mice.

Thus, these results strongly support our conclusion that activation of vagal nerves innervating the pancreas enhances GSIS and promotes β -cell proliferation. These findings are now presented in Extended Data Figures 7a, 7b, 9a and b and are described in the Results section of the revised manuscript (page 5, lines 25 to 32 and page 5, line 51 to

page 6, line 2).

4: The huge concern is the injection of blue light-emitting lanthanide microparticles (LMPs) in the pancreatic ducts. This will cause toxicity and cause inflammation and immune response. Moreover, whether the light intensity is consistent throughout pancreas? What is the light intensity in the organ? 35 W of NIR illumination is very strong and can cause heat. There is red light responsive opsin which has been identified. If the authors use red light responsive light sensors instead of ChR2, they do not need to inject the LMPs to make the system so complicate.

Wavelength of lights which activate reported red light-responsive opsin, such as ReaChR, ChrimsonR and ChRmine, are around 600 nm (Lehtinen et al, 2022, *Front Cell Neurosci*), which is far lower than NIR. This wavelength of red light can reportedly reach portions of the brain at as deep as 7 mm from the skull surface and activate ChRmine (Chen et al, 2021, *Nat Biotechnol*). However, the pancreas is located in a much deeper portion of the abdominal cavity. In addition, unlike the brain, lights must pass through several structures to reach the pancreas in the abdominal cavity. Therefore, red light wavelengths may not reach the pancreas efficiently and we thus elected not to use red lights in the present study. Additionally, and more importantly, we aimed at selectively activating vagal nerves innervating the pancreas. Therefore, to achieve our purpose, we opted for the strategy of placing LMPs within the pancreas.

As the reviewer suggested, LMP implantation may exert toxicity and induce inflammation in the pancreas. However, circulating amylase levels were not elevated either before or after NIR illumination in CC-red-mice (Extended Data Figure 6a of the original and revised manuscripts), although LMP particles were mainly localized in the exocrine tissues of the pancreas (Extended Data Figure 5a of the original and revised manuscripts), suggesting LMP-induced inflammation in the pancreas to be quite mild. In addition, body weights and food intakes at 2 months after the start of NIR illumination were comparable to those observed before or soon after starting NIR illumination. Taken together, these results indicate that the toxic effects of LMP implantation on the general conditions of these mice are apparently too small to impact metabolic phenotypes even after long-term placement of LMP. These findings are now presented in Extended Data Figures 6d and e, and are described in the Results section of the revised manuscript (page 5 lines 12 to 16). Light intensity sensors were inserted into the abdominal cavities of anaesthetised mice, allowing us to directly measure the NIR light intensities in many portions of the pancreas, from head to tail, under the same NIR illumination conditions. With delivery of NIR from

the outside of the body, the NIR light intensities were actually detectable. As shown in Extended Data Figure 5b of the original and revised manuscripts, the NIR light intensities were approximately 0.5 mW/mm² on average and the SEM value was very small, indicating that NIR reached the pancreas efficiently and uniformly. We have now described these results and methods in detail in the revised manuscript (page 4, lines 51 to 53 and page 11, lines 10 to 14).

We agree with the reviewer's comment that NIR illumination might well generate heat. We therefore measured surface and core temperatures of the mice after NIR illumination for 3 seconds/cage every 36 seconds for 2 days and obtained results indicating no significant elevations of either surface or core body temperature. Thus, this intermittent NIR illumination raised neither surface nor core body temperature. These findings are now presented in Extended Data Figure 6h, and are described in the Results section of the revised manuscript (page 5, lines 24 to 25).

Importantly, as noted in the responses to Criticism#3, we employed several controls, e.g., WT-blue-NIR-, CC-red-NIR- and CC-blue-dark-mice, and compared glucose tolerance, amounts of BrdU-positive β -cells and β -cell mass with those of CC-blue-NIR-mice, and found that only the CC-blue-NIR-mice exhibited better glucose tolerance and increases in both β -cell proliferation and masses. These results clearly demonstrate that non-specific effects exerted solely by LMP implantation and/or NIR illumination, if any, were minimal in terms of enhancement of β -cell function and proliferation observed in CC-blue-mice. We appreciate the reviewer's insightful suggestions.

5: The author should explain why plasma insulin levels after glucose loading were significantly higher in LED-oVNS mice compared with control mice, while blood glucose levels were similar between the two groups in Figure 3f and g.

As the reviewer commented, plasma insulin levels after glucose loading were significantly higher in LED-oVNS mice than in control mice, while blood glucose levels differed minimally between the two groups. Subdiaphragmatic vagal nerves innervate not only the pancreas but also other organs in the abdominal cavity, including the gastrointestinal tract. As LED-oVNS activates subdiaphragmatic vagal nerves, this system may affect several organs, involved in glucose metabolism, including the GI tract. In fact, we observed that LED-oVNS significantly enhanced duodenal blood flow and gastrointestinal tract motility as compared to those in control mice, as shown in Extended Data Figs. 1d and 4b of the original and revised manuscripts. These mechanisms may accelerate the influx of ingested glucose into the peripheral circulation, leading to post-

load blood glucose elevation. Enhanced GSIS by LED-oVNS may have offset the enhanced post-load blood glucose elevation.

Therefore, to exclude these extra-pancreatic effects and to examine purely the role of vagal nerves innervating the pancreas, we developed NIR-oVNS, a more selective system for activating vagal nerves innervating the pancreas. Accordingly, we observed significantly lower glucose levels with enhanced GSIS after glucose loading by NIR-oVNS. Notably, in both the LED- and the NIR- oVNS experiments, no hypoglycaemia was observed during GTT. Thus, the NIR-oVNS system allowed us to demonstrate that selective activation of vagal nerves innervating the pancreas enhances GSIS, thereby reducing blood glucose levels after glucose loading. We have now added an explanation why LED-oVNS exerted minimal effects on post-load blood glucose elevation, despite promoting GSIS, to the revised manuscript (page 4, lines 28 to 29).

#6 is missing

7: Longer-term performance (functionality and safety) of their system should be presented in T1D mice.

As suggested, to evaluate the long-term functionality and safety of this optogenetic system in T1D-model mice, we administered STZ to ChAT-ChR2 mice injected with blue or red light-emitting LMPs, accompanied by NIR illumination, and then monitored the blood glucose levels of these mice for 2 months. Blood glucose elevations were markedly suppressed in STZ-treated CC-blue-NIR-mice as compared to CC-red-NIR-mice throughout the 2-month period of NIR stimulation. Interestingly, significant increases in BrdU-positive β -cells were similarly observed in STZ-treated CC-blue-NIR-mice 2 months after starting oVNS, suggesting that β -cell proliferation continued to be enhanced. As often observed in STZ-treated mice, β -cell masses of STZ-treated CC-red-mice 2 months after receiving STZ treatment and starting oVNS were markedly decreased as compared with those 3 weeks after administration of STZ and starting oVNS. In contrast, β -cell masses of STZ-treated CC-NIR-blue-mice were still increased 2 months after starting oVNS, as compared with those of STZ-treated CC-NIR-red-mice. Thus, β -cell proliferative effects of NIR-oVNS were maintained, at least for 2 months.

In addition, body weights of STZ-treated CC-blue-NIR-mice were maintained at initial levels for 2 months, whereas those of STZ-treated CC-red-mice gradually decreased likely due to insulin deficiency. Therefore, it is unlikely that harmful effects were elicited in CC-blue-NIR-mice even after long-term NIR-oVNS.

We replaced Figures 6a, b and c of the original manuscript with new figures including the

results of long-term experiments, as now presented in Figures 6b, c and d of the revised manuscript. We further included long-term body weight results as Extended Data Fig. 12e. In addition, we have now described these findings in the Results section of the revised manuscript (page 6, lines 44 to 53).

8: In figure 6, some of the data obtained in one week (eg. Brdu) and some in three weeks (blood glucose). Actually, they should show the BrdU data after three weeks of illumination.

β -cell mass increases after β -cell proliferation occurs. Therefore, we showed β -cell mass data at three weeks after starting oVNS, which is two weeks beyond the time point at which we observed enhancement of β -cell proliferation. Since the reviewer requested BrdU-positive β -cells three weeks after starting oVNS, we examined BrdU-positive β -cell ratios after three weeks of illumination and obtained results indicating that BrdU-positive β -cells were consistently increased in STZ-treated CC-blue-NIR-mice as compared with STZ-treated CC-red-mice three weeks after starting oVNS. These findings are now presented in Figure 6c, and are described in the Results section of the revised manuscript (page 6, lines 46 to 47).

9: The time schedule of STZ injection, LMPs injection and light illumination was not specified in the text. Please clarify the specific timeline in Figures.

As suggested, we added a figure (Figure 6a) showing the time course of STZ experiments in which time points of LMP injection, STZ injection, NIR illumination and data samplings are specified. The time schedule is now clearly described in the Results section of the revised manuscript (page 6, lines 34 to 36).

10: The authors only present the light frequency but they did not present the light intensity which should be present in the results.

Regarding LED-oVNS, the light intensity of LED was in the range of 5 to 10 mW. Regarding NIR-oVNS, as shown in Extended data Figure 5b of the original and revised manuscripts, NIR intensity in the abdominal cavity was 0.5 mW/mm². We now give these values in the Results section of the revised manuscript (page 4, lines 51 to 53).

Responses to reviewer 3

1. Data shown in Figures 3 f and g are not consistent. Although Fig. 3f shows higher

insulin levels in response to LED-oVNS stimulation, the resulting. Since blood-glucose control is the ultimate therapeutic goal of any diabetes therapy, the concept kind of misses the major therapeutic point. It is imperative to see a glycemic response as a consequence of increased insulin and it will be also important to show that these insulin levels will not lead to any hypoglycemia.

As the reviewer commented, in the LED-oVNS experiment, blood glucose levels remained unchanged as compared to those of the controls, despite higher insulin levels. Subdiaphragmatic vagal nerves innervate not only the pancreas but also other organs in the abdominal cavity, including those of the gastrointestinal tract. As LED-oVNS activates subdiaphragmatic vagal nerves, this system may affect several organs, involved in glucose metabolism, including the GI tract. In fact, we observed that LED-oVNS significantly enhanced duodenal blood flow and gastrointestinal tract motility as compared to those in control mice, as shown in Extended Data Figs. 1d and 4b of the original and revised manuscripts. These mechanisms may accelerate the influx of ingested glucose into the peripheral circulation, leading to post-load blood glucose elevation. Enhanced GSIS by LED-oVNS may have offset the enhanced post-load blood glucose elevation.

Therefore, to exclude these extra-pancreatic effects and to examine purely the role of vagal nerves innervating the pancreas, we developed NIR-oVNS, a more selective system for activating vagal nerves innervating the pancreas. Accordingly, we observed significantly lower glucose levels with enhanced GSIS after glucose loading by NIR-oVNS. Notably, in both of the LED- and the NIR- oVNS experiments, no hypoglycaemia was observed during GTT. Thus, the NIR-oVNS system allowed us to demonstrate that selective activation of vagal nerves innervating the pancreas enhances GSIS, thereby reducing blood glucose levels after glucose loading. We have now added an explanation why LED-oVNS exerted minimal effects on post-load blood glucose elevation, despite promoting GSIS, to the revised manuscript (page 4, lines 28 to 29).

Aiming at the ultimate therapeutic goal of diabetes management, we further evaluated the long-term functionality of NIR-oVNS employing T1D-model mice, and observed that blood glucose elevations were markedly suppressed in STZ-treated CC-blue-NIR-mice as compared to CC-red-NIR-mice throughout the 2-month period of NIR stimulation. In addition, significant increases in BrdU-positive β -cells and β -cell masses, as compared with those of STZ-treated CC-red-NIR-mice, were observed in STZ-treated CC-blue-NIR-mice 2 months after starting oVNS, indicating β -cell proliferative effects of NIR-oVNS to be maintained, at least for 2 months. These results demonstrate the therapeutic

potential of activating vagal nerves innervating the pancreas for insulin-deficient diabetes. These findings are now presented in Figures 6b, 6c and 6d (Figures 6a, 6b and 6c in the original manuscript) of the revised manuscript, and are described in the Results section of the revised manuscript (page 6, lines 44 to 52).

2. It seems like an important negative control is missing in Figure 4C. The authors only show NIR-induced vagal nerve activation in ChAT-ChR2. Essential control would show animals without ChAT-ChR2 and ChAT-ChR2 animals illuminated with red light (660 nm) instead of NIR.

We agree with the reviewer's comment that control mice, e.g., mice without ChAT-ChR2 and ChAT-ChR2 mice illuminated with light with another wavelength, are essential. First, we measured responses of vagal nerves innervating the pancreas in wild-type mice, in which blue light-emitting LMPs had been implanted, followed by illumination with NIR light. Electrophysiological responses were not observed under these experimental conditions.

As suggested, we additionally measured electrophysiological responses of vagal nerves innervating the ChAT-ChR2 mouse pancreas, in which blue light-emitting LMPs had been implanted, followed by illumination with light of another wavelength near the 973nm of the NIR light used to up-convert LMPs in the present study. We employed light with a wavelength of 850 nm which is closer to 973 nm than the suggested red visible (600 nm) light. We detected no electrophysiological responses under this experimental condition. These findings are now presented in Extended Data Fig. 5c and are described in the Results section of the revised manuscript (page 5, lines 3 to 6).

3. Although this standard optogenetic stimulation strategy resulted in the activation of the vagal nerves, similar to previous optogenetic reports on stimulation of nerve cells, the authors must discuss and show by additional experimentation beyond doubt the absence of long-term cytotoxicity, the long-term efficiency of upconversion and the impact of high energy requirement on every-day and real-world therapies.

As the reviewer suggested, we examined the long-term efficacy and toxicity of the NIR-oVNS system, by applying NIR illumination for 2 months. First, we examined β -cell proliferation and masses of CC-red-NIR-mice (referred to as control-mice for the NIR-oVNS experiments in the original manuscript) and CC-blue-NIR-mice (referred to as NIR-oVNS-mice in the original manuscript) 2 months after the start of NIR illumination.

β -cell proliferative effects of NIR-oVNS were still present after 2 months of NIR illumination, and the β -cell mass of CC-blue-NIR-mice was markedly larger than that of CC-red-mice even 2 months after starting oVNS. These findings indicate the long-term efficacy of NIR-oVNS in terms of β -cell proliferative effects. These findings are now presented in Extended Data Figures 10a and b, and are described in the Results section of the revised manuscript (page 6, lines 2 to 4).

Next, we examined body weights and food intakes of CC-blue-NIR- and CC-red-NIR-mice 2 months after the start of NIR illumination. Body weights and food intakes at 2 months after starting NIR illumination were comparable to those measured before or soon after the start of NIR illumination. Therefore, toxic effects of LMP implantation on the general conditions of these mice were apparently low. These findings are now presented in Extended Data Figure 6b-e, and are described in the Results section of the revised manuscript (page 5, lines 9 to 15).

As the reviewer suggested, the clinical relevance of our present results is very important. Optogenetic approaches are difficult to apply in humans, because these approaches involve expressing opsins in neurons. However, the present study showing that activation of vagal nerves innervating the pancreas sufficiently promotes β -cell proliferation and increases β -cell mass may open novel avenues to the development of diabetes therapies based on increasing β -cells. We are now planning to stimulate vagal nerves innervating the pancreas by employing electric vagal nerve stimulation devices in large animals, such as rhesus monkeys. These devices have recently been applied in clinical situations, such as treatments for epilepsy, depression, rheumatoid arthritis, inflammatory bowel disease and arrhythmias (Bonaz et al, 2016, *J Physiol*, Zhang et al, 2011, *Heart Failure reviews*). We now elaborate on this important issue in Discussion section of the revised manuscript (page 8, lines 16 to 23).

Minor points:

1. There are several conceptionally and technically similar publications out there and the authors should carefully compare their work with this previous work.

As suggested, we cited recent reports describing optogenetic vagal nerve activation as enhancing GSIS in anesthetised mice (Fontaine et al, 2021, *Sci Rep*) and selective activation of vagal nerves innervating the pancreas by a chemogenetic approach enhanced GSIS in freely-moving mice (Jimenez-Gonzalez et al, 2022, *Nat Biomed Eng.*). In the present study, we showed that selective activation of the vagal nerves innervating the pancreas enhanced GSIS, an observation consistent with those described in previous

reports. Importantly, we demonstrated for the first time that selective vagal nerve activation alone is sufficient to induce marked pancreatic β -cell proliferation, thereby substantially increasing the mass of functional β -cells. We added these discussions, with citations of relevant new references as #40 and 41, in the Discussion section (page 7, lines 2 to 5).

2. bar graphs in the main figures are not aligned (e.g., Fig 4) and are using different style and fonts (e.g., Fig3h and i).

As suggested, we carefully aligned the graphs and corrected the fonts in all figures.

3. Figure 1; The immunostainings of beta-cells and vagal nerve are very poor and should be replaced with images of much higher quality.

We apologise for presenting figures of poor quality. We have replaced the pictures of Figure 1 with images of high quality.

Rebuttal 2

Responses to Reviewer 2

The authors have addressed most of my concerns.

But, I have one more question. In the schematic diagram of Fig. 6a, why did the authors start the lighting before creating the diabetes model? This doesn't make sense. The light treating should be applied in a created diabetic mouse model.

As noted in the original manuscript, the reason we performed STZ experiments was to explore whether the β -cells augmented by NIR-oVNS are functional. As shown in Fig. 5b, 2-week NIR-oVNS almost doubled the β -cell mass. If NIR-oVNS applied after STZ treatment markedly decreased pancreatic β -cells, we speculated that doubling of a very few beta cells, i.e., a very small increase in β -cells, would not be sufficient to reverse hyperglycaemia. Therefore, we applied NIR-oVNS to mice in which β -cell mass was still maintained, and obtained results indicating that NIR-oVNS suppressed STZ-induced hyperglycaemia. In addition, during this revision, we examined plasma insulin levels of STZ-treated CC-blue-NIR-mice at several time-points, such as 1, 2 and 3 weeks after starting NIR-oVNS. Plasma insulin levels of STZ-treated CC-blue-NIR-mice after STZ administration were significantly higher than those of STZ-treated CC-red-NIR-mice both 2 and 3 weeks after starting NIR-oVNS. These results clearly demonstrate that NIR-oVNS increases functional β -cells which can produce sufficient insulin to suppress blood glucose elevation. We replaced Extended Data Figure 12d of the original manuscript, now presented in Figure 6b, and these results are described in the Results section of the revised manuscript (page 6, line 55 to page 7, line 2).

Responses to Reviewer 4

Major

- 1. It would be helpful to know which of the organs adjacent to the pancreas also express ChR2 in cholinergic fibers and may be modulated by chronic light delivery to the abdomen. Cholinergic innervation of the stomach, duodenum, jejunum, and ileum have been reported. Reports regarding cholinergic innervation of the liver and portal system are conflicting and examining ChR2 expression may help clarify this field.*

As suggested, using ChAT-ChR2 mice, we evaluated YFP expression in the stomach, the duodenum, the jejunum, the ileum, the liver and around the portal vein. YFP-positive cells and fibers were detected in the stomach, the duodenum, the jejunum and the ileum, but

not in the liver or around the portal vein. These observations are consistent with results showing that blood flows in the pancreas and the duodenum were increased but those in the liver were not increased, while flow in the liver was not, by activation of subdiaphragmatic vagal nerves in ChAT-ChR2 mice with LED-oVNS (CC-LED-mice). These findings are now presented in Extended Data Figure 1a and are described in the Results section of the revised manuscript (page 2, lines 47 to 48 and page 3, lines 49 to 50).

2. The authors show optogenetic vagal stimulation increases plasma insulin without an effect on blood glucose during ipGTT. These results are in keeping with decreased insulin sensitivity and perhaps not enhanced insulin secretion. There are no significant differences in glucagon between control and treated groups but these studies have a relatively small n and only examine a single time point. The authors performed an ITT but according to the methods, this was not conducted with nerve stimulation so does not test whether vagal nerve stimulation alters insulin sensitivity. It would be helpful to perform an ITT during vagal stimulation to determine insulin sensitivity and perhaps examine glucagon release at additional time points.

We appreciate the reviewer's important comments. As suggested by the reviewer, we performed insulin tolerance tests concomitantly with vagal nerve stimulation in LED-oVNS experiments both before and 2 weeks after starting LED-oVNS. Under these experimental conditions, decreases in glucose levels after insulin administration were similar in CC-ctrl-mice and CC-LED-mice at both time-points, indicating similar insulin sensitivities irrespective of LED stimulation. Subdiaphragmatic vagal nerves innervate not only the pancreas but also other organs in the abdominal cavity, including the gastrointestinal tract. As LED-oVNS activates subdiaphragmatic vagal nerves, this system may affect several organs involved in glucose metabolism, including the GI tract. In fact, we observed that LED-oVNS significantly enhanced duodenal blood flow and gastrointestinal tract motility as compared to those in control mice, as shown in Extended Data Figs. 1d and 4b of the original and revised manuscripts. These mechanisms, rather than increased insulin resistance, may accelerate the influx of ingested glucose into the peripheral circulation, leading to post-load blood glucose elevation. Enhanced GSIS in response to LED-oVNS may have offset the enhanced post-load blood glucose elevation. Therefore, to exclude these extra-pancreatic effects and to examine purely the role of vagal nerves innervating the pancreas, we developed NIR-oVNS, a more selective system for activating vagal nerves innervating the pancreas. Accordingly, we observed

significantly lower glucose levels with enhanced GSIS, after glucose loading by NIR-oVNS.

Next, we examined plasma glucagon levels at several time-points during glucose tolerance tests at 0, 15, 30 and 60 minutes after glucose loading concomitantly with vagal nerve stimulation in LED-oVNS experiments. Plasma glucagon levels of CC-LED-mice during glucose tolerance tests were similar to those of CC-ctrl-mice. Therefore, enhancement of glucose stimulated insulin secretion of CC-LED-mice is caused by vagal nerve activation rather than by secondary effects related to decreased insulin sensitivity or increased plasma glucose levels. These findings are now presented in Extended Data Figures 2a, 2d and 3b, and are described in the Results section of the revised manuscript (page 4, lines 3 to 6, page 4, lines 10 to 12 and page 4, lines 24 to 26).

3. Similarly, it would be helpful to perform insulin tolerance testing with chronic NIR stimulation to ensure the increase in beta cell mass is not secondary to insulin resistance during stimulation.

As requested, we performed insulin tolerance tests concomitantly with vagal nerve stimulation after chronic NIR-oVNS for 2 weeks. Under this experimental condition, decreases in glucose levels after insulin administration were similar in CC-blue-NIR-mice and CC-red-NIR-mice, indicating similar insulin sensitivities in these two groups of mice. Therefore, increases in β -cell masses of CC-blue-NIR-mice are caused by vagal nerve activation rather than by secondary effects related to decreased insulin sensitivity. These findings are now presented in Extended Data Figure 11c and are described in the Results section of the revised manuscript (page 6, lines 22 to 24).

4. Please could the authors clarify the intensity of NIR light used during measurement of electrophysiological responses of vagal nerves (Fig 4c)? The NIR intensity reported in the abdominal cavity (0.5mW/mm²) appears to be significantly less than used in published CNS studie so confirming this is sufficient to activate cholinergic neurons is important, eg. by demonstrating increased fos expression in intrapancreatic ganglia. If the NIR light is producing recordable responses in branches of the vagus, this suggests that activation is not limited to intrapancreatic terminals and maybe back-propagating, potentially with effects on other organs.

Electrophysiological responses were recorded under the same NIR illumination conditions; the final output power at the tip was ~35 W and the tip was set downward at

a height of 20 cm above the animals. Under these conditions, as described in the original manuscript, the measured NIR light intensities were approximately 0.5 mW/mm^2 on average and the standard error value was very small (Extended Data Fig. 5b), indicating that NIR reached the pancreas efficiently and uniformly. As noted by the reviewer, this intensity seems to be lower than that used in the CNS study (Ref 36 in the revised manuscript). However, in that previous study, the authors provided NIR transmission rates (approximately 6%) and only estimated its intensity at a depth of 4.5 mm in the brain. In any case, we agree with the reviewer's comment that we should confirm that our NIR procedure was indeed sufficient to activate cholinergic neurons.

First, we showed electrophysiological responses of vagal nerves in the original manuscript, which indicates that vagal nerves were activated during NIR-oVNS *in vivo*. In addition, as suggested by the reviewer, we histologically examined c-Fos expression in pancreatic ganglia of CC-blue-NIR-mice and evaluated the ratios of c-Fos positive cells in YFP-positive cells in parasympathetic ganglionic cells after 1 week of NIR-oVNS, and showed the ratios of c-Fos+/YFP+ cells in CC-blue-NIR-mice to be significantly higher than those in those of CC-red-NIR-mice. Thus, we clearly demonstrated that vagal nerves innervating the pancreas were indeed activated by the NIR-oVNS system *in vivo*. We appreciate the reviewer's useful suggestion. These findings are now presented in Figure 4d and are described in the Results section of the revised manuscript (page 5, lines 17 to 19).

With regard to the second comment, we measured electrophysiological responses of vagal nerves innervating the pancreas during NIR illumination by applying hooked paired bipolar electrodes to the pancreas. Therefore, it is likely that electrophysiological responses were evoked in the intrapancreatic vagal nerves but not in vagal nerves distant from the pancreas. However, we agree that examining the possibility of back-propagating responses by NIR-oVNS is important to further confirm the specificity of the NIR-oVNS system as it pertains to the pancreas. Therefore, we examined electrophysiological responses of vagal nerves running along the oesophageal wall, which is more central and distant from the intrapancreatic vagus nerves, during NIR-oVNS, but no electrophysiological responses were detected at the vagal nerve trunk running along the oesophageal wall. These findings suggest that back-propagating responses by NIR-oVNS are unlikely under our experimental conditions. These findings are now presented in Figure 4d and Extended Data Figure 5d, and are described in the Results section of the

revised manuscript (page 5, lines 15 to 17).

5. *The authors do not show increased insulin in the NIR treated-STZ studies which would be helpful to demonstrate the increase in beta cell mass is functional.*

We appreciate the reviewer's important comment. In the original manuscript, we only showed a tendency for increases in plasma insulin levels in STZ-treated CC-blue-NIR-mice by evaluating at single time-point, one week after starting NIR-oVNS. We further examined plasma insulin levels at several time-points, such as 1, 2 and 3 weeks after starting NIR-oVNS. Plasma insulin levels of STZ-treated CC-blue-NIR-mice after STZ administration were significantly higher than those of STZ-treated CC-red-NIR-mice both 2 and 3 weeks after starting NIR-oVNS. These results indicate that NIR-oVNS increases functional β -cells which can produce sufficient insulin to suppress blood glucose elevation. We replaced Extended Data Figure 12d of the original manuscript and now presented the findings in Figure 6b, and these results are also described in the Results section of the revised manuscript (page 6, line 55 to page 7, line 2).

6. *The authors should assess significance using 2-way ANOVA for their longitudinal studies examining insulin, glucose and body weight rather than multiple t-tests or 1-way ANOVA (Fig 3 f, g, Fig 4 e – h, Fig 6 b, Extended data: fig 3a, Fig 4a, Fig 6b-e, Fig 7, Fig 11 c, fig 12 e).*

As suggested, we statistically re-analyzed our results using 2-way repeated measures ANOVA for all longitudinal studies. In some of the studies, levels of statistical significance partially changed, but these alterations did not affect the conclusions drawn based on each of the experimental results.

7. *The volume of LMP (500ul) injected into the pancreatic duct for LMP delivery is quite large compared to published studies and both groups of LMP treated mice fail to gain weight over two months of the study. While blue LMP injected mice may not differ from red LMP injected mice, there may be toxicity from LMP per se and comparison against mice injected with vehicle alone would be helpful to determine this.*

As the reviewer suggested, LMP implantation may exert toxic effects in the pancreas. Therefore, as suggested by the reviewer, we examined body weights and food intakes of

ChAT-ChR2 mice in which the same amount of vehicle (saline) had been injected into the pancreatic ducts (CC-veh-NIR), for comparison with the same parameters in CC-blue-NIR- and CC-red-NIR- mice with 2 weeks of NIR illumination. Neither body weights nor food intakes differed among the three groups of mice. These results further suggest that LMP-induced chronic blue light emission had minimal adverse effects on the general conditions of ChAT-ChR2 mice. We replaced Extended Data Figures 6b and 6c of the original manuscript with figures including the results of CC-veh-NIR-mice and these findings are now presented in Extended Data Figures 6b and 6c, as well as being described in the Results section of the revised manuscript (page 5, lines 23 to 27).

Minor

- 1. Chr2 expression is shown in cells adjacent to islets but there is no neural marker used to confirm expression in intrapancreatic ganglia rather than off-target expression in acinar cells.*

As suggested, we performed immunohistochemistry of the pancreas from ChAT-ChR2-YFP mice using antibodies against beta-Tubulin 3, a neuron cell specific marker. We observed that YFP-positive areas adjacent to islets were also positive for beta-tubulin 3, indicating these areas to be parasympathetic ganglia. These findings are now presented in Figure 1c and are described in the Results section of the revised manuscript (page 2, lines 50 to 51).

- 2. Some experimental details are missing that are needed to interpret the data. When were the electrophysiological recordings from the vagus made in relation to the chronic stimulation with LED implants? It would be helpful to show that light-evoked activation is present at the end of the 2 week stimulation schedule.*

In the original manuscript, we recorded electrophysiological responses of the intrapancreatic vagal nerves before starting LED-oVNS. Therefore, as suggested by the reviewer, we recorded electrophysiological responses of vagal nerves 2 weeks after the starting LED-oVNS. Electrophysiological responses were clearly detected according to LED stimulation in terms of timings and amplitude. These results indicate that oVNS does, in fact, activate vagal nerves even during chronic LED-oVNS. These findings are now presented in Extended Data Figure 3a and are described in the Results section of the revised manuscript (page 4, lines 19 to 20).

3. It would be helpful for the authors to indicate in the text that the LMP approach likely modulates activity at both the preganglionic cholinergic terminals onto the intrapancreatic ganglia and the post-ganglionic cholinergic neurons in the intrapancreatic ganglia.

As requested, we described NIR-oVNS as being likely to activate both pre- and post-ganglionic vagal nerves thereby improving the readability of the Discussion section in the revised manuscript (page 7, lines 31 to 33).

4. Please could the authors indicate when the vagotomy was performed in relation to the GTT studies performed with LMP activation. Does the loss of insulin response with vagotomy indicate the site of the dominant effect of LMP activation (preganglionic cholinergic terminals vs. postganglionic neurons)?

We performed the vagotomy concomitantly with LMP implantation, followed by GTT one week after the vagotomy. For the vagotomy procedure, we dissected subdiaphragmatic vagal nerves. Therefore, the primary site of blockade was the preganglionic vagal nerve fibers. In addition, as neurons injected from the parasympathetic ganglia in the pancreas are reportedly connected to preganglionic nerve fibers at the ganglia (Jimenez-Gonzalez et al, 2022, *Nat Biomed Eng.*), vagotomy affects the activities of both pre- and post-ganglionic neuronal fibers. We described this procedure in the Methods section (page 10, lines 24 to 26).

5. Could the authors also clarify the NIR intensity used to demonstrate blue light emission at the surface of the pancreas (fig 4b)?

As described in the original and revised manuscripts, we measured NIR intensity in many intraabdominal portions, including the surface of the pancreas, by inserting light intensity sensors into the abdominal cavities of the anaesthetised mice under the same experimental conditions. The NIR light intensities were approximately 0.5 mW/mm² on average and the standard error value was very small (Extended Data Fig. 5b), indicating that NIR reached the pancreas efficiently and uniformly.

Rebuttal 3

Responses to Reviewer 2

They have addressed my concerns.

One more suggestion, to better show the effectiveness of NIR-oVNS-mediated blood control compared with healthy mice, it is better to include wild-type control mice in Figure 6. b and c.

As shown in Figure 6c, blood glucose levels of STZ-treated CC-red-NIR-mice elevated above 500 mg/dl. These blood glucose levels are comparable to those of STZ-treated wild-type mice in previous reports including ours (Imai et al, 2008, *Science*, Lee et al, 2016 *Sci Rep*, Sidarala et al, 2020 *JCI Insight*). Therefore, experiments in Figure 6 fully demonstrated the effectiveness of NIR-oVNS in ameliorating STZ-induced hyperglycemia, and STZ-treated wild-type mice were not included as a control in these experiments.

Responses to Reviewer 4

On page 4, lines 37-38, the authors comment that enhanced gastrointestinal motility might minimize suppression of post-load blood glucose despite increased insulin. According to their methods, the glucose load was delivered by IP injection rather than orally and so the glucose load would bypass the GI tract and not be affected by changes in GI motility.

Perhaps the authors could comment on why the significant increase in plasma insulin during optogenetic stimulation does not alter blood glucose during ipGTT when insulin sensitivity is unchanged and effects on GI motility are unlikely to be contributing.

We agree with the reviewer's insightful comment. We observed that LED-oVNS significantly enhanced duodenum blood flows (Extended Data Fig. 1e in both original and revised manuscripts). Therefore, we speculate that this effect might promote the absorption of intraperitoneal glucose from the peritoneal membrane, thereby minimising the suppression of post-load blood glucose elevation despite the promotion of glucose-stimulated insulin secretion. We added these discussions in the Results section of the revised manuscript (page 4, lines 30 to 32).